# LARGE LEARNING RATE TAMES HOMOGENEITY: CONVERGENCE AND BALANCING EFFECT

**Yuqing Wang, Minshuo Chen, Tuo Zhao, Molei Tao**
Georgia Institute of Technology
{ywang3398,mchen393,tourzhao,mtao}@gatech.edu

## ABSTRACT

Recent empirical advances show that training deep models with large learning rate often improves generalization performance. However, theoretical justifications on the benefits of large learning rate are highly limited, due to challenges in analysis. In this paper, we consider using Gradient Descent (GD) with a large learning rate on a homogeneous matrix factorization problem, i.e., $\min_{X,Y} \|A - XY^\top\|_F^2$. We prove a convergence theory for constant large learning rates well beyond $2/L$, where $L$ is the largest eigenvalue of Hessian at the initialization. Moreover, we rigorously establish an implicit bias of GD induced by such a large learning rate, termed 'balancing', meaning that magnitudes of $X$ and $Y$ at the limit of GD iterations will be close even if their initialization is significantly unbalanced. Numerical experiments are provided to support our theory.

## 1 INTRODUCTION

Training machine learning models such as deep neural networks involves optimizing highly nonconvex functions. Empirical results indicate an intimate connection between training algorithms and the performance of trained models (Le et al., 2011; Bottou et al., 2018; Zhang et al., 2021; Soydaner, 2020; Zhou et al., 2020). Especially for widely used first-order training algorithms (e.g., GD and SGD), the learning rate is of essential importance and has received extensive focus from researchers (Smith, 2017; Jastrzebski et al., 2017; Smith, 2018; Gotmare et al., 2018; Liu et al., 2019; Li & Arora, 2019). A recent perspective is that large learning rates often lead to improved testing performance compared to the counterpart trained with small learning rates (Smith & Topin, 2019; Yue et al., 2020). Towards explaining the better performance, a common belief is that large learning rates encourage the algorithm to search for flat minima, which often generalize better and are more robust than sharp ones (Seong et al., 2018; Lewkowycz et al., 2020).

Despite abundant empirical observations, theoretical understandings of the benefits of large learning rate are still limited for non-convex functions, partly due to challenges in analysis. For example, the convergence (of GD or SGD) under large learning rate is not guaranteed. Even for globally smooth functions, very few general results exist if the learning rate exceeds certain threshold (Kong & Tao, 2020). Besides, popular regularity assumptions such as global smoothness for simplified analyses are often absent in homogeneous models, including commonly used ReLU neural networks.

This paper theoretically studies the benefits of large learning rate in a matrix factorization problem

$$\min_{X,Y} \frac{1}{2} \left\| A - XY^\top \right\|_F^2, \quad \text{where } A \in \mathbb{R}^{n \times n}, \ X, Y \in \mathbb{R}^{n \times d}. \tag{1}$$

We consider Gradient Descent (GD) for solving (1): at the $k$-th iteration, we have

$$X_{k+1} = X_k + h(A - X_k Y_k^\top)Y_k \quad \text{and} \quad Y_{k+1} = Y_k + h(A^\top - Y_k X_k^\top)X_k,$$

where $h$ is the learning rate. Despite its simple formula, problem (1) serves as an important foundation of a variety of problems, including matrix sensing (Chen & Wainwright, 2015; Bhojanapalli et al., 2016; Tu et al., 2016), matrix completion (Keshavan et al., 2010; Hardt, 2014), and linear neural networks (Ji & Telgarsky, 2018; Gunasekar et al., 2018).

Problem (1) possesses several intriguing properties. Firstly, the objective function is non-convex, and critical points are either global minima or saddles (see e.g., Baldi & Hornik (1989); Li et al. (2019b); Valavi et al. (2020a); Chen et al. (2018)). Secondly, problem (1) is homogeneous in $X$ and $Y$, meaning that rescaling $X, Y$ to $aX, a^{-1}Y$ for any $a \neq 0$ will not change the objective's value. This property is shared by commonly used ReLU neural networks. A direct consequence of homogeneity is that global minima of (1) are non-isolated and can be unbounded. The curvatures at these global minima are highly dependent on the magnitudes of $X, Y$. When $X, Y$ have comparable magnitudes, the largest eigenvalue of Hessian is small, and this corresponds to a flat minimum; on the contrary, unbalanced $X$ and $Y$ give a sharp minimum. Last but not the least, the homogeneity impairs smoothness conditions of (1), rendering the gradient being not Lipschitz continuous unless $X, Y$ are bounded. See a formal discussion in Section 2.

Existing approaches for solving (1) often uses explicit regularization (Ge et al., 2017; Tu et al., 2016; Cabral et al., 2013; Li et al., 2019a), or infinitesimal (or diminishing) learning rates for controlling the magnitudes of $X, Y$ (Du et al., 2018; Ye & Du, 2021). In this paper, we go beyond the scope of aforementioned works, and analyze GD with a large learning rate for solving (1). In particular, we allow the learning rate $h$ to be as large as approximately $4/L$ (see more explanation in Section 2), where $L$ denotes the largest eigenvalue of Hessian at GD initialization. In connection to empirical observations, we provide positive answers to the following two questions:

> *Does GD with large learning rate converge at least for some cases of* (1)?
> *Does larger learning rate biases toward flatter minima (i.e., $X, Y$ with comparable magnitudes)?*

We theoretically show the convergence of GD with large learning rate for the two situations $n = 1, d \in \mathbb{N}^+$ or $d = 1, n \in \mathbb{N}^+$ with isotropic $A$. We also observe a, perhaps surprising, "balancing effect" for general matrix factorization (i.e., any $d$, $n$, and $A$), meaning that when $h$ is sufficiently large, the difference between $X$ and $Y$ shrinks significantly at the convergence of GD compared to its initial, even if the initial point is close to an unbalanced global minimum. In fact, with a proper large learning rate $h$, $\|X_k - Y_k\|_{\mathsf{F}}^2$ may decrease by an arbitrary factor at its limit. The following is a simple example of our theory for $n = 1$ (i.e. scalar factorization), and more general results will be presented later with a precise bound for $h$ depending on the initial condition and $A$.

**Theorem 1.1** (Informal version of Thm.3.1 & 3.2). *Given scalar $A$ and initial condition $X_0, Y_0 \in \mathbb{R}^{1 \times d}$ chosen almost everywhere, with learning rate $h \lesssim 4/L$, GD converges to a global minimum $(X, Y)$ satisfying $\|X\|_{\mathsf{F}}^2 + \|Y\|_{\mathsf{F}}^2 \leq \frac{2}{h}$. Consequently, its extent of balancing is quantified by $\|X - Y\|_{\mathsf{F}}^2 \leq \frac{2}{h} - 2A$.*

We remark that having a learning rate $h \approx 4/L$ is far beyond the commonly analyzed regime in optimization. Even for globally $L$-smooth objective, traditional theory requires $h < 2/L$ for GD convergence and $h = 1/L$ is optimal for convex functions (Boyd et al., 2004), not to mention that our problem (1) is never globally $L$-smooth due to homogeneity. Modified equation provides a tool for probing intermediate learning rates (see Hairer et al. (2006, Chapter 9) for a general review, and Kong & Tao (2020, Appendix A) for the specific setup of GD), but the learning rate here is too large for modified equation to work (see Appendix C). In fact, besides blowing up, GD with large learning rate may have a zoology of limiting behaviors (see e.g., Appendix B for convergence to periodic orbits under our setup, and Kong & Tao (2020) for convergence to chaotic attractors).

Our analyses (of convergence and balancing) leverage various mathematical tools, including a proper partition of state space and its dynamical transition (specifically invented for this problem), stability theory of discrete time dynamical systems (Alligood et al., 1996), and geometric measure theory (Federer, 2014).

The rest of the paper is organized as: Section 2 provides the background of studying (1) and discusses related works; Section 3 presents convergence and balancing results for scalar factorization problems; Section 4 generalizes the theory to rank-1 matrix approximation; Section 5 studies problem (1) with arbitrary $A$ and its arbitrary-rank approximation; Section 6 summarizes the paper and discusses broadly related topics and future directions.

## 2    BACKGROUND AND RELATED WORK

**Notations**. $\|v\|$ is $\ell_2$ norm of a column or row vector $v$. $\|M\|_{\mathsf{F}}$ is the Frobenius norm of a matrix $M$.

**Sharp and flat minima in** (1)    We discuss the curvatures at global minima of (1). To ease the presentation, consider simplified versions of (1) with either $n = 1$ or $d = 1$. In this case, $X$ and $Y$ become vectors and we denote them as $x, y$, respectively. We show the following proposition characterizing the spectrum of Hessian at a global minimum.

**Proposition 2.1.** *When $n = 1, d \in \mathbb{N}^+$ or $d = 1, n \in \mathbb{N}^+$ in (1), the largest eigenvalue of Hessian at a global minimum $(x, y)$ is $\|x\|^2 + \|y\|^2$ and the smallest eigenvalue is $0$.*

Homogenity implies global minimizers of (1) are not isolated, which is consistent with the $0$ eigenvalue. On the other hand, if the largest eigenvalue $\|x\|^2 + \|y\|^2$ is large (or small), then the curvature at such a global minimum is sharp (or flat), in the direction of the leading eigenvector. Meanwhile, note this sharpness/flatness is an indication of the balancedness between magnitudes of $x, y$ at a global minimum. To see this, singular value decomposition (SVD) yields that at a global minimum, $(x, y)$ satisfies $\left\|xy^\top\right\|_{\mathsf{F}}^2 = \sigma_{\max}^2(A)$. Therefore, large $\|x\|^2 + \|y\|^2$ is obtained when $|\|x\| - \|y\||$ is large, i.e., $x$ and $y$ magnitudes are unbalanced, and small $\|x\|^2 + \|y\|^2$ is obtained when balanced.

**Large learning rate**    We study smoothness properties of (1) and demonstrate that our learning rate is well beyond conventional optimization theory. We first define the smoothness of a function.

**Definition 2.2** ($L$-smooth). A function $f \in \mathcal{C}^1$ defined on $\mathbb{R}^N$ is $L$-*smooth* if for all $u_1, u_2 \in \mathbb{R}^N$,

$$\|\nabla f(u_1) - \nabla f(u_2)\| \leq L\|u_1 - u_2\|. \tag{2}$$

If further $f \in \mathcal{C}^2$, then $\nabla^2 f \preceq LI$. Moreover, if we have (2) for $u_1, u_2 \in \mathcal{X} \subseteq \mathbb{R}^N$, we call it locally $L$-smooth.

In traditional optimization (Nesterov, 2003; Polyak, 1987; Nesterov, 1983; Polyak, 1964; Beck & Teboulle, 2009), most analyzed objective functions often satisfy (i) (some relaxed form of) convexity or strong convexity, and (ii) $L$-smoothness. Choosing a step size $h < 2/L$ guarantees the convergence of GD to a minimum by the existing theory (reviewed in Appendix F.3). Our choice of learning rate $h \approx 4/L$ (more precisely, $4/L_0$; see below) goes beyond the classical analyses.

Besides, in our problem (1), the regularity is very different. Even simplified versions of (1), i.e., with either $n = 1$ or $d = 1$, suffer from (i) non-convexity and (ii) unbounded eigenvalues of Hessian, i.e., no global smoothness (see Appendix F.2 for more details). As shown in Du et al. (2018); Ye & Du (2021); Ma et al. (2021), decaying or infinitesimal learning rate ensures that the GD trajectory stays in a locally smooth region. However, the gap between the magnitudes of $X, Y$ can only be maintained in that case. We show, however, that larger learning rate can shrink this gap. More precisely, if initial condition is well balanced, there is no need to use large learning rate; otherwise, we can use learning rate as large as approximately $4/L$, and within this range, larger $h$ provides smaller gap between $X$ and $Y$ at the limit (i.e. infinitely many GD iterations).

**Related work**    Matrix factorization problems in various forms have been extensively studied in the literature. The version of (1) is commonly known as the low-rank factorization, although here $d$ can arbitrary and we consider both $d \leq \text{rank}(A)$ and $d > \text{rank}(A)$ cases. Baldi & Hornik (1989); Li et al. (2019b); Valavi et al. (2020b) provide landscape analysis of (1). Ge et al. (2017); Tu et al. (2016) propose to penalize the Frobenius norm of $\left\|X^\top X - Y^\top Y\right\|_{\mathsf{F}}^2$ to mitigate the homogeneity and establish global convergence guarantees of GD for solving (1). Cabral et al. (2013); Li et al. (2019a) instead penalize individual Frobenius norms of $\|X\|_{\mathsf{F}}^2 + \|Y\|_{\mathsf{F}}^2$. We remark that penalizing $\|X\|_{\mathsf{F}}^2 + \|Y\|_{\mathsf{F}}^2$ is closely related to nuclear norm regularization, since the variational formula for nuclear norm $\|Z\|_*^2 = \min_{Z = XY^\top} \|X\|_{\mathsf{F}}^2 + \|Y\|_{\mathsf{F}}^2$. More recently, Liu et al. (2021) consider using injected noise as regularization to GD and establish a global convergence (see also Zhou et al. (2019); Liu et al. (2022). More specifically, by perturbing the GD iterate $(X_k, Y_k)$ with Gaussian noise, GD will converge to a flat global optimum. On the other hand, Du et al. (2018); Ye & Du (2021); Ma et al. (2021) show that even without explicit regularization, when the learning rate of GD is infinitesimal,

i.e., GD approximating gradient flow, $X$ and $Y$ maintain the gap in their magnitudes. Such an effect is more broadly recognized as implicit bias of learning algorithms (Neyshabur et al., 2014; Gunasekar et al., 2018; Soudry et al., 2018; Li et al., 2020; 2021). Built upon this implicit bias, Du et al. (2018) further prove that GD with diminishing learning rates converges to a bounded global minimum of (1), and this conclusion is recently extended to the case of a constant small learning rate (Ye & Du, 2021). Our work goes beyond the scopes of these milestones and considers matrix factorization with much larger learning rates.

Additional results exist that demonstrate large learning rate can improve performance in various learning problems. Most of them involve non-constant learn rates. Specifically, Li et al. (2019c) consider a two-layer neural network setting, where using learning rate annealing (initially large, followed by small ones) can improve classification accuracy compared to training with small learning rates. Nakkiran (2020) shows that the observation in Li et al. (2019c) even exists in convex problems. Lewkowycz et al. (2020) study constant large learning rates, and demonstrate distinct algorithmic behaviors of large and small learning rates, as well as empirically illustrate large learning rate yields better testing performance on neural networks. Their analysis is built upon the neural tangent kernel perspective (Jacot et al., 2018), with a focus on the kernel spectrum evolution under large learning rate. Worth noting is, Kong & Tao (2020) also study constant large learning rates, and show that large learning rate provides a mechanism for GD to escape local minima, alternative to noisy escapes due to stochastic gradients.

## 3 OVERPARAMETERIZED SCALAR FACTORIZATION

In order to provide intuition before directly studying the most general problem, we begin with a simple special case, namely factorizing a scalar by two vectors. It corresponds to (1) with $n = 1$ and $d \in \mathbb{N}^+$, and this overparameterized problem is written as

$$\min_{x,y \in \mathbb{R}^{1 \times d}} \frac{1}{2}(\mu - xy^\top)^2, \tag{3}$$

where $\mu$ is assumed without loss of generality to be a positive scalar. Problem (3) can be viewed as univariate regression using a linear two-layer neural network with the quadratic loss, which is studied in Lewkowycz et al. (2020) with atomic data distribution. Yet our analysis in the sequel can be used to study arbitrary univariate data distributions; see details in Section 6.

Although simplified, problem (3) is still nonconvex and exhibits the same homogeneity as (1). The convergence of its large learning rate GD optimization was previously not understood, let alone balancing. Many results that we will obtain for (3) will remain true for more general problems.

We first prove that GD converges despite of $> 2/L$ learning rate and for almost all initial conditions:

**Theorem 3.1** (Convergence). *Given* $(x_0^\top, y_0^\top) \in (\mathbb{R}^d \times \mathbb{R}^d) \backslash \mathcal{B}$ *where* $\mathcal{B}$ *is some Lebesgue measure-0 set, when the learning rate* $h$ *satisfies*

$$h \leq \min \left\{ \frac{4}{\|x_0\|^2 + \|y_0\|^2 + 4\mu}, \frac{1}{3\mu} \right\},$$

*GD converges to a global minimum.*

Theorem 3.1 says that choosing a constant learning rate depending on GD initialization guarantees the convergence for almost every starting point in the whole space. This result is even stronger than the already nontrivial convergence under small learning rate with high probability over random initialization (Ye & Du, 2021). Furthermore, the upper bound on $h$ is sufficiently large: on the one hand, suppose GD initialization $(x_0, y_0)$ is close to an unbalanced global minimum. By Proposition 2.1, we can check that the largest eigenvalue $L(x_0, y_0)$ of Hessian $\nabla^2 f(x_0, y_0)$ is approximately $\|x_0\|^2 + \|y_0\|^2$. Consequently, our upper bound of $h$ is almost $4/L$, which is beyond $2/L$ (see Section 2 for more details). On the other hand, we observe numerically that the $4/L$ upper bound is actually very close to the stability limit of GD when initialized away from the origin (see more details in Appendix E).

The convergence in Theorem 3.1 has an interesting searching-to-converging transition as depicted in Figure 1, where we observe two phases. In Phase 1, large learning rate drives GD to search for flat

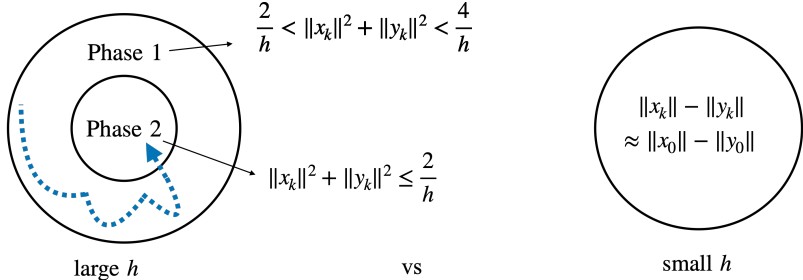

Figure 1: The dynamics of GD under different learning rate $h$

regions, escaping from the attraction of sharp minima. After some iterations, the algorithm enters the vicinity of a global minimum with more balanced magnitudes in $x, y$. Then in Phase 2, GD converges to the found balanced global minimum. We remark that the searching-to-converging transition also appears in Lewkowycz et al. (2020). However, the algorithmic behaviors are not the same. In fact, in our searching phase (phase 1), the objective function does not exhibit the blow-up phenomenon. In our convergence phase (phase 2), the analysis relies on a detailed state space partition (see Line 192) due to nonconvex nature of (3), while the analysis in Lewkowycz et al. (2020) is akin to monotone convergence in a convex problem.

In comparison with the dynamics of small learning rate, we note that the searching phase (Phase 1) is vital to the convergence analysis. Meanwhile, the searching phase induces a balancing effect of large learning rate. The following theorem explicitly quantifies the extent of balancing.

**Theorem 3.2** (Balancing). *Under the same initial condition and learning rate $h$ as Theorem 3.1, GD for (3) converges to a global minimizer $(x, y)$ satisfying*

$$\|x\|^2 + \|y\|^2 \leq \frac{2}{h}.$$

*Consequently, its extent of balancing is quantified by*

$$\|x - y\|^2 \leq \frac{2}{h} - 2\mu.$$

One special case of Theorem 3.2 is the following theorem, which states that no matter how close to a global minimum does GD start, if this minimum does not correspond to well-balanced norms, a large learning rate will take the iteration to a more balanced limit. We also demonstrate a sharp shrinkage in the distance between $x$ and $y$.

**Corollary 3.3** (From 'unbalanced' to 'balanced'). *For any $\delta \in (0, \mu)$, let the GD initialization satisfy*

$$(x_0, y_0) \in \left\{ (u, v) : |uv^\top - \mu| < \delta, \ \|u\|^2 + \|v\|^2 > 8\mu \right\} \backslash \mathcal{B},$$

*where $\mathcal{B}$ is some Lebesgue measure-0 set. When the learning rate $h$ satisfies $h = \frac{4}{\|x_0\|^2 + \|y_0\|^2 + 4\mu}$, the extent of balancing at the limiting point $(x, y)$ of GD obeys*

$$\|x - y\|^2 < \frac{1}{2}\|x_0 - y_0\|^2 + 2\mu.$$

Both Theorem 3.2 and Corollary 3.3 suggest that larger learning rate yields better balancing effect, as $\|x - y\|^2$ at the limit of GD may decrease a lot and is controlled by the learning rate. We remark that the balancing effect is a consequence of large learning rate, as small learning rate can only maintain the difference in magnitudes of $x, y$ (Du et al., 2018).

In addition, the actual balancing effect can be quite strong with $\|x_k - y_k\|^2$ decreasing to be almost 0 at its limit under a proper choice of large learning rate. Figure 2 illustrates an almost perfect balancing case when $h = \frac{4}{\|x_0\|^2 + \|y_0\|^2 + 4} \approx 0.0122$ is chosen as the upper bound. The difference $\|x_k - y_k\|$ decreases from approximately 17.9986 to 0.0154 at its limit. Additional experiments with various learning rates and initializations can be found in Appendix A.

**Technical Overview** We sketch the main ideas behind Theorem 3.1, which lead to the balancing effect in Theorem 3.2. Full proof is deferred to Appendix G.

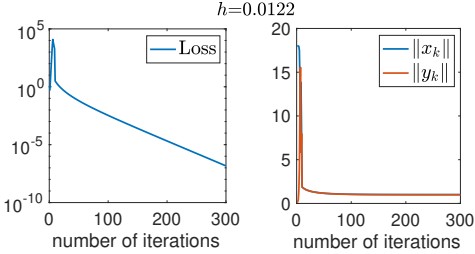

Figure 2: The objective function is $(1 - xy^\top)^2/2$, where $x^\top, y^\top \in \mathbb{R}^{10}$. Highly unbalanced initial condition is uniformly randomized, with the norms to be $\|x_0\| = 18, \|y_0\| = 0.09$.

The convergence is proved by handling Phase 1 and 2 separately (see Fig.1). In Phase 1, we prove that $\|x_k\|^2 + \|y_k\|^2$ has a decreasing trend as GD searches for flat minimum. We show that $\|x_k\|^2 + \|y_k\|^2$ may not be monotone, i.e., it either decreases every iteration or decreases every other iteration.

In Phase 2, we carefully partition the state space and show GD at each partition will eventually enter a monotone convergence region. Note that the partition is based on detailed understanding of the dynamics and is highly nontrivial. Attentive readers may refer to Appendix G for more details. The combination of Phase 1 & 2 is briefly summarized as a proof flow chart in Figure 3.

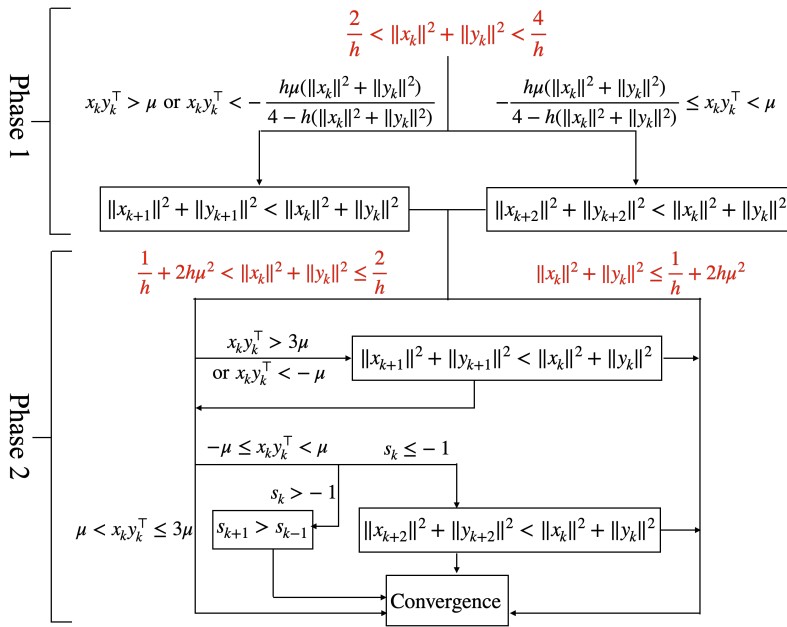

Figure 3: Proof overview of Theorem 3.1. At the $k$-th iteration, we denote $(x_k, y_k)$ as the iterate and $s_k$ is defined as $x_{k+1}y_{k+1}^\top - \mu = s_k(x_k y_k^\top - \mu)$.

# 4  RANK-1 APPROX. OF ISOTROPIC $A$ (AN UNDER-PARAMETERIZED CASE)

Given insights from scalar factorization, we consider rank-1 factorization of an isotropic matrix $A$, i.e., $A = \mu I_{n\times n}$ with $\mu > 0$, $d = 1$, and $n \in \mathbb{N}^+$. The corresponding optimization problem is

$$\min_{x,y\in\mathbb{R}^{n\times 1}} \frac{1}{2} \left\| \mu I_{n\times n} - xy^\top \right\|_F^2. \tag{4}$$

Although similar at an uncareful glance, Problems (4) and (3) are rather different unless $n = d = 1$. First of all, Problem (4) is under-parameterized for $n > 1$, while (3) is overparameterized. More importantly, we'll show that, when $(x, y)$ is a global minimum of (4), $x, y$ must be aligned, i.e., $x = \ell y$ for some $\ell > 0$. In the scalar factorization problem, however, no such alignment is required. As a result, the set of global minima of (4) is an $n$-dimensional submanifold embedded in a $2n$-dimensional space, while in the scalar factorization problem the set of global minimum is a $(2d - 1)$-dimensional

submanifold — one rank deficient — in a $2d$-dimensional space. We expect the convergence in (4) is more complicated than that in (3), since searching in (4) is demanding.

To prove the convergence of large learning rate GD for (4), our theory consists of two steps: (i) show the convergence of the alignment between $x$ and $y$ (this is new); (ii) use that to prove the convergence of the full iterates (i.e., $x$ & $y$). Step (i) first:

**Theorem 4.1** (Alignment). *Given $(x_0, y_0) \in (\mathbb{R}^n \times \mathbb{R}^n)\backslash\mathcal{B}$, where $\mathcal{B}$ is some Lebesgue measure-0 set, when learning rate $h \leq \min\left\{ \frac{4}{\|x_0\|^2 + \|y_0\|^2 + 4\sqrt{7}\mu}, \frac{1}{2\sqrt{7}\mu} \right\}$, the iterator $(x_k, y_k)$ of GD at the $k$-th iteration satisfies $|\cos(\angle(x_k, y_k))| \to 1$ as $k \to \infty$.*

*Proof sketch.* A sufficient condition for the convergence of $|\cos(\angle(x_k, y_k))|$ is $\|x_k\|^2 \|y_k\|^2 - (x_k^\top y_k)^2 \to 0$. To ease the presentation, let $U_k = x_k^\top y_k$, $V_k = x_k^\top x_k$, and $W_k = y_k^\top y_k$. By the GD update and some algebraic manipulation, we derive

$$V_{k+1}W_{k+1} - U_{k+1}^2 = r_k \cdot (V_k W_k - U_k^2),$$

where $r_k = (1 - h(V_k + W_k) + h^2(V_k W_k - \mu^2))^2$. When $k$ is sufficiently large, we can show a uniform upper bound on $r_k < 1 - c$ for some constant $c > 0$. In this way, we deduce that $V_k W_k - U_k^2$ will exponentially decay and converge to 0. More details are provided in Appendix H. $\square$

Theorem 4.1 indicates that GD iterations will converge to the neighbourhood of $\{(x, y) : x = \ell y, \text{for some } \ell \in \mathbb{R}\backslash\{0\}\}$. This helps establish the global convergence as stated in Step (ii).

**Theorem 4.2** (Convergence). *Under the same initial conditions and learning rate $h$ as Theorem 4.1, GD for (4) converges to a global minimum.*

Similar to the over-parametrized scalar case, this convergence can also be split into two phases where phase 1 motivates the balancing behavior with the decrease of $\|x_k\|^2 + \|y_k\|^2$, and phase 2 ensures the convergence. The following balancing theorem is thus obtained.

**Theorem 4.3** (Balancing). *Under the same initial conditions and learning rate $h$ as Theorem 4.1, GD for (4) converges to a global minimizer that obeys*

$$\|x\|^2 + \|y\|^2 \leq \frac{2}{h},$$

*and its extent of balancing is quantified by*

$$\|x - y\|^2 \leq \frac{2}{h} - 2\mu.$$

This conclusion is the same as the one in Section 3. A quantitatively similar corollary like Corollary 3.3 can also be obtained from the above theorem, namely, if $x_0$ and $y_0$ start from an unbalanced point near a minimum, the limit will be a more balanced one.

## 5 GENERAL MATRIX FACTORIZATION

In this section, we consider problem (1) with an arbitrary matrix $A \in \mathbb{R}^{n \times n}$. We replicate the problem formulation here for convenience,

$$\min_{X, Y \in \mathbb{R}^{n \times d}} \frac{1}{2} \left\| A - XY^\top \right\|_\mathsf{F}^2. \tag{5}$$

Note this is the most general case with $n, d \in \mathbb{N}^+$ and any square matrix $A$. Due to this generalization, we no longer utilize the convergence analysis and instead, establish the balancing theory via stability analysis of GD as a discrete time dynamical system.

Let $\mu_1 \geq \mu_2 \geq \cdots \geq \mu_n \geq 0$ be the singular values of $A$. Assume for technical convenience $\|A\|_\mathsf{F}^2 = \sum_{i=1}^n \mu_i^2$ being independent of $d$ and $n$. We denote the singular value decomposition of $A$ as $A = UDV^\top$, where $U, D, V \in \mathbb{R}^{n \times n}$, $U, V$ are orthogonal matrices and $D$ is diagonal. Then we establish the following balancing effect.

**Theorem 5.1.** *Given almost all the initial conditions, for any learning rate $h$ such that GD for (5) converges to a point $(X, Y)$, there exists $c = c(n, d) > c_0$ with constant $c_0 > 0$ independent of $h$, $n$, and $d$, such that $(X, Y)$ satisfies*

$$c(\|X\|_{\mathsf{F}}^2 + \|Y\|_{\mathsf{F}}^2) < \frac{2}{h},$$

*and the extent of balancing is quantified by $\left\| X - (UV^\top)Y \right\|_{\mathsf{F}}^2 < \frac{2}{ch} - 2\sum_{i=1}^{\min\{d,n\}} \mu_i$, which means*

$$\left| \|X\|_{\mathsf{F}} - \|Y\|_{\mathsf{F}} \right|^2 < \frac{2}{ch} - 2\sqrt{\sum_{i=1}^{\min\{d,n\}} \mu_i^2}.$$

*In particular, when $d = 1$, i.e., rank-1 factorization of an arbitrary $A$, the constant $c$ equals $1$.*

We observe that the extent of balancing can be quantified under some rotation of $Y$. This is necessary, since for factorizing a general matrix $A$ (which can be asymmetric), at a global minimum, $X, Y$ may only align after a rotation (which is however fixed by $A$, independent of initial or final conditions). Figure 4 illustrates an example of the balancing effect under different learning rates. Evidently, larger learning rate leads to a more balanced global minimizer. Additional experiments with various dimensions, learning rates, and initializations can be found in Appendix A; a similar balancing effect is also shown for additional problems including matrix sensing and matrix completion there.

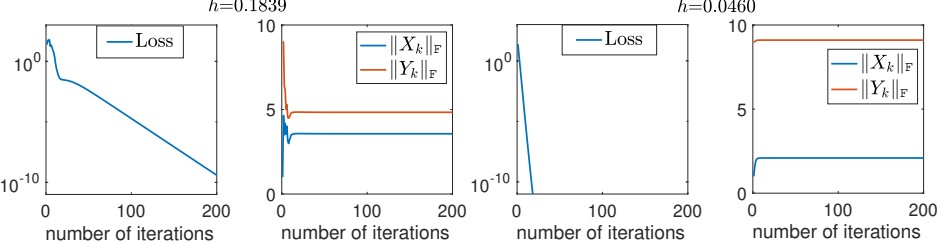

Figure 4: Balancing effect of general matrix factorization. We independently generate elements in $A \in \mathbb{R}^{6 \times 6}$ from a Gaussian distribution. We choose $X, Y \in \mathbb{R}^{6 \times 100}$ and randomly pick a pair of initial point $(X_0, Y_0)$ with $\|X_0\|_{\mathsf{F}} = 1$ and $\|Y_0\|_{\mathsf{F}} = 9$.

Different from previous sections, Theorem 5.1 builds on stability analysis by viewing GD as a dynamical system in discrete time. More precisely, the proof of Theorem 5.1 (see Appendix I for details) consists of two parts: (i) the establishment of an easier but equivalent problem via the rotation of $X$ and $Y$, (ii) stability analysis of the equivalent problem.

For (i), by singular value decomposition (SVD), $A = UDV^\top$, where $U, D, V \in \mathbb{R}^{n \times n}$, $U$ and $V$ are orthogonal matrices, and $D$ is a non-negative diagonal matrix. Let $X_k = UR_k$, $Y_k = VS_k$. Then

$$\begin{cases} X_{k+1} = X_k + h(A - X_k Y_k^\top)Y_k \\ Y_{k+1} = Y_k + h(A - X_k Y_k^\top)^\top X_k \end{cases}$$

$$\Leftrightarrow \begin{cases} UR_{k+1} = UR_k + h(UDV^\top - UR_k S_k^\top V^\top)VS_k \\ VS_{k+1} = VS_k + h(UDV^\top - UR_k S_k^\top V^\top)^\top UR_k \end{cases}$$

$$\Leftrightarrow \begin{cases} R_{k+1} = R_k + h(D - R_k S_k^\top)S_k \\ S_{k+1} = S_k + h(D - R_k S_k^\top)^\top R_k \end{cases}.$$

Therefore, GD for problem (5) is equivalent to GD for the following problem

$$\min_{R,S \in \mathbb{R}^{n \times d}} \frac{1}{2}\left\| D - RS^\top \right\|_{\mathsf{F}}^2$$

and it thus suffices to work with diagonal non-negative $A$.

For (ii), here is a brief description of the idea of stability analysis: consider each iteration of GD as a mapping $\psi$ from $u_k \sim (X_k, Y_K)$ to $u_{k+1} \sim (X_{k+1}, Y_{k+1})$, where matrices $X$ and $Y$ are flattened

and concatenated into a vector so that $\psi$ is a closed map on vector space $\mathbb{R}^{2dn}$. GD iteration is thus a discrete time dynamical system on state space $\mathbb{R}^{2dn}$ given by

$$u_{k+1} = \psi(u_k) = u_k - h\nabla f(u_k),$$

where $f$ is the objective function $f(u_k) = \frac{1}{2}\left\|A - X_k Y_k^\top\right\|_{\mathsf{F}}^2$, and gradient returns a vector that collects all component-wise partial derivatives.

It's easy to see that any stationary point of $f$, denoted by $u^*$, is a fixed point of $\psi$, i.e., $u^* = \psi(u^*)$. What fixed point will the iterations of $\psi$ converge to? For this, the following notions are helpful:

**Proposition 5.2.** *Consider a fixed point $u^*$ of $\psi$. If all the eigenvalues of Jacobian matrix $\nabla\psi(u^*)$ are of complex modulus less than* 1*, it is a stable fixed point.*

**Proposition 5.3.** *Consider a fixed point $u^*$ of $\psi$. If at least one eigenvalue of Jacobian matrix $\nabla\psi(u^*)$ is of complex modulus greater than* 1*, it is an unstable fixed point.*

Roughly put, the stable set of an unstable fixed point is of negligible size when compared to that of a stable fixed point, and thus what GD converges to is a stable fixed point for almost all initial conditions (Alligood et al., 1996). Thus, we investigate the stability of each global minimum of $f$ (each saddle of $f$ is an unstable fixed point of $\psi$ and thus is irrelevant). By a detailed evaluation of $\nabla\psi$'s eigenvalues (Appendix I), we see that a global minimum $(X, Y)$ of $f$ corresponds to a stable fixed point of GD iteration if $\left|1 - ch\left(\|X\|_{\mathsf{F}}^2 + \|Y\|_{\mathsf{F}}^2\right)\right| < 1$, i.e., it is balanced as in Thm. 5.1.

## 6 Conclusion and Discussion

In this paper, we demonstrate an implicit regularization effect of large learning rate on the homogeneous matrix factorization problem solved by GD. More precisely, a phenomenon termed as "balancing" is theoretically illustrated, which says the difference between the two factors $X$ and $Y$ may decrease significantly at the limit of GD, and the extent of balancing can increase as learning rate increases. In addition, we provide theoretical analysis of the convergence of GD to the global minimum, and this is with large learning rate that can exceed the typical limit of $2/L$, where $L$ is the largest eigenvalue of Hessian at GD initialization.

For the matrix factorization problem analyzed here, large learning rate avoids bad regularities induced by the homogeneity between $X$ and $Y$. We feel it is possible that such balancing behavior can also be seen in problems with similar homogeneous properties, for example, in tensor decomposition (Kolda & Bader, 2009), matrix completion (Keshavan et al., 2010; Hardt, 2014), generalized phase retrieval (Candes et al., 2015; Sun et al., 2018), and neural networks with homogeneous activation functions (e.g., ReLU). Besides the balancing effect, the convergence analysis under large learning rate may be transplanted to other non-convex problems and help discover more implicit regularization effects.

In addition, factorization problems studied here are closely related to two-layer linear neural networks. For example, one-dimensional regression via a two-layer linear neural network can be formulated as the scalar factorization problem (3): Suppose we have a collection of data points $(x_i, y_i) \in \mathbb{R} \times \mathbb{R}$ for $i = 1, \ldots, n$. We aim to train a linear neural network $y = (u^\top v)x$ with $u, v \in \mathbb{R}^d$ for fitting the data. We optimize $u, v$ by minimizing the quadratic loss,

$$(u^*, v^*) \in \arg\min_{u,v} \frac{1}{n}\sum_{i=1}^n \left(y_i - (u^\top v)x_i\right)^2 = \arg\min_{u,v} \frac{1}{n}\left(\frac{\sum_{i=1}^n x_i y_i}{\sum_{i=1}^n x_i^2} - u^\top v\right)^2. \tag{6}$$

As can be seen, taking $\mu = \frac{\sum_{i=1}^n x_i y_i}{\sum_{i=1}^n x_i^2}$ recovers (3). In this regard, our theory indicates that training of $u, v$ by GD with large learning rate automatically balances $u, v$, and the obtained minimum is flat. This may provide some initial understanding of the improved performance brought by large learning rates in practice. Note that (6) generalizes to arbitrary data distribution of training a two-layer linear network with atomic data (i.e., $x = 1$ and $y = 0$) in Lewkowycz et al. (2020).

It is important to clarify, however, that there is a substantial gap between this demonstration and extensions to general neural networks, including deep linear and nonlinear networks. Although we suspect that large learning rate leads to similar balancing effect of weight matrices in the network, rigorous theoretical analysis is left as a future direction.

## ACKNOWLEDGMENTS

We thank anonymous reviewers and area chair for suggestions that improved the quality of this paper. The authors are grateful for partial supports from NSF DMS-1847802 (YW and MT) and ECCS-1936776 (MT).

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

# Supplementary Materials for "Large Learning Rate Tames Homogeneity: Convergence and Balancing Effect"

## A  ADDITIONAL EXPERIMENTS

### A.1  MORE RESULTS FOR MATRIX FACTORIZATION

In this section, we present more experiments with different choices of $n, d$, various initializations and scalings, and a broader range of learning rates. All these experiments verify our claim on the balancing effect of large learning rate, i.e., the shrinkage of the gap between the magnitudes of $X$ and $Y$ exists for general matrix factorization problems, i.e., for any choice of $n, d \in \mathbb{N}^+$.

We first provide examples of scalar factorization in Figure 5, 6, and 7 to numerically justify our theory in Section 3. In the three figures, the initial conditions randomly generated, respectively with $(\|x_0\|, \|y_0\|) = (9, 1)$, $(\|x_0\|, \|y_0\|) = (19, 1)$, and $(\|x_0\|, \|y_0\|) = (99, 1)$; the learning rates are chosen within the range of Theorem 3.1 from large to small as $h_0, \frac{6}{7}h_0, \frac{5}{7}h_0, \frac{4}{7}h_0, \frac{3}{7}h_0, \frac{2}{7}h_0$ for the 1st-6th columns respectively where $h_0 = 4/(\|x_0\|^2 + \|y_0\|^2 + 8)$. The learning rates of the left three columns are larger than $2/L$ ($L$ is the local Lipschitz constant of gradient near the initial condition), where we can see the decrease in the gap between $\|x_k\|$ and $\|y_k\|$ and larger learning rate leads to smaller gap; the right three correspond to $h < 2/L$ where there are almost no changes in $\|x_k\|$ and $\|y_k\|$ as $k$ increases. Moreover, the loss does not decrease monotonically at the beginning of the iterations for all the $h > 2/L$ cases while in the later iterations GD shows monotone convergence; for the right three columns ($h < 2/L$ cases), we can see monotone decrease of the loss. This validates our two-phase pattern of convergence (see e.g. Figure 1 and Section 3 for detailed explanation).

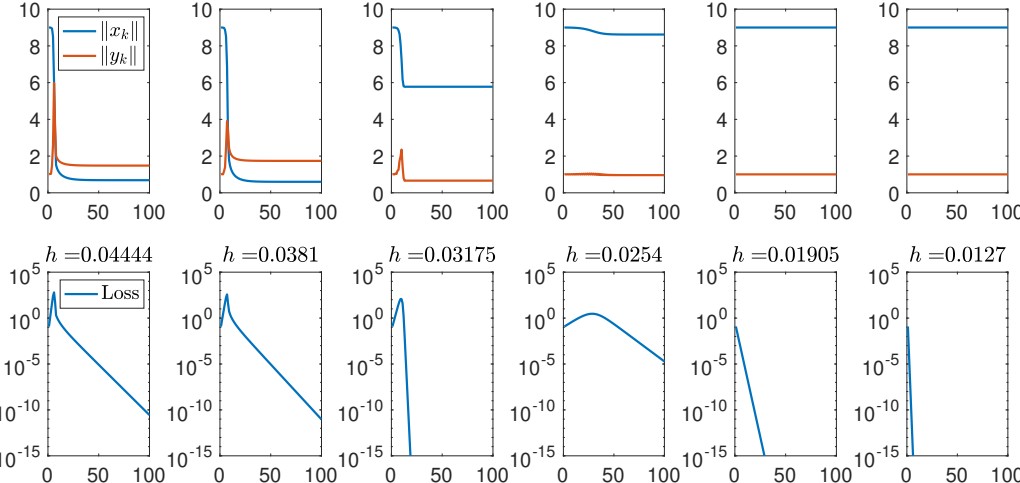

Figure 5: Scalar factorization with $\|x_0\| = 9, \|y_0\| = 1$. The $x$−axis represents the number of iterations $k$; the $y$−axis represents the value for the norm of $x_k$ and $y_k$ in the first row, and the value for loss in the second row; the learning rate $h$ for each column is the same.

Then we experiment with the general matrix factorization as a supplement of our theory in Section 5 where we only rigorously prove the balancing effect given the convergence of GD. In the following examples, we show that there indeed exist large learning rates that trigger the shrinkage of the gap between $\|X_k\|_\mathsf{F}$ and $\|Y_k\|_\mathsf{F}$ and at the same time guarantee the convergence of GD. Figure 8, 9, and 10 correspond to the over-parameterized version of matrix factorization problem (5) with $A \in \mathbb{R}^{6 \times 6}$ (asymmetric, generated by $i.i.d.$ Gaussian) and $X, Y \in \mathbb{R}^{6 \times 100}$. The initial conditions are randomly generated with $(\|X_0\|_\mathsf{F}, \|Y_0\|_\mathsf{F}) = (9, 1)$, $(\|X_0\|_\mathsf{F}, \|Y_0\|_\mathsf{F}) = (19, 1)$, and

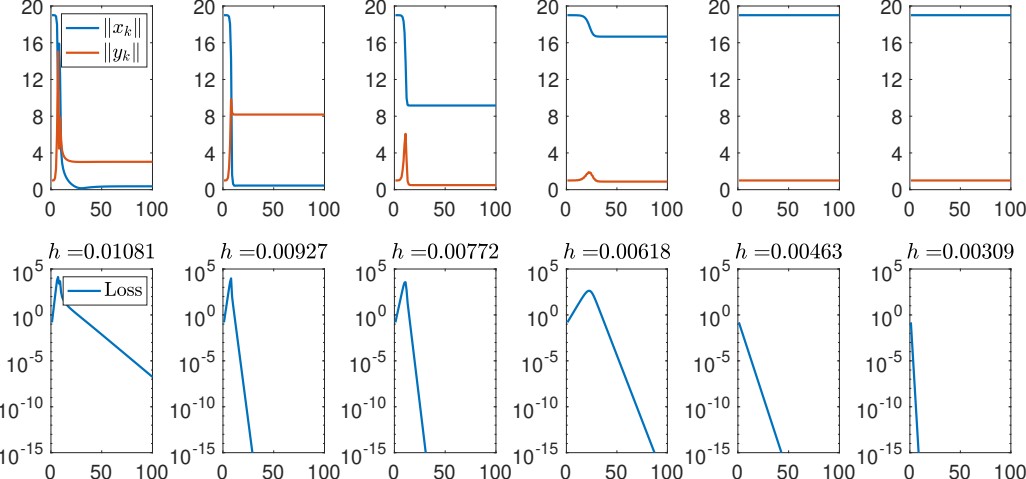

Figure 6: Scalar factorization with $\|x_0\| = 19, \|y_0\| = 1$. The $x-$axis represents the number of iterations $k$; the $y-$axis represents the value for the norm of $x_k$ and $y_k$ in the first row, and the value for loss in the second row; the learning rate $h$ for each column is the same.

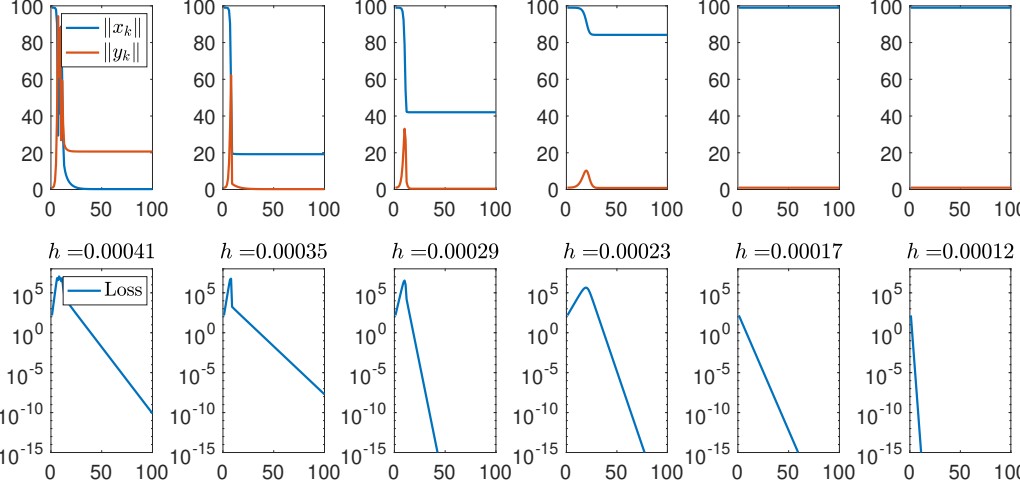

Figure 7: Scalar factorization with $\|x_0\| = 99, \|y_0\| = 1$. The $x-$axis represents the number of iterations $k$; the $y-$axis represents the value for the norm of $x_k$ and $y_k$ in the first row, and the value for loss in the second row; the learning rate $h$ for each column is the same.

$(\|X_0\|_\mathsf{F}, \|Y_0\|_\mathsf{F}) = (99, 1)$ respectively. Similarly, the learning rates are chosen from large to small as $h_0, \frac{6}{7}h_0, \frac{5}{7}h_0, \frac{4}{7}h_0, \frac{3}{7}h_0, \frac{2}{7}h_0$ for the 1st-6th columns respectively where $\frac{6}{7}h_0$ (the 2nd column) is picked near the stability limit. We also similarly provide two examples of the under-parameterized version in Figure 11 and 12. The two examples correspond to problem (5) with $A \in \mathbb{R}^{100 \times 100}$ (asymmetric, similarly generated as the previous one) and $X, Y \in \mathbb{R}^{100 \times 3}$. Here we use a shifted loss which is to subtract the global minimum error.

As is shown in Figure 8, 9, 10, 11, and 12, we can observe the similar phenomenon as the scalar case, in the sense that (1) larger learning rate gives a smaller the gap between $\|X_k\|_\mathsf{F}$ and $\|Y_k\|_\mathsf{F}$ in the limit, except for the overly large $h$ that causes GD to diverge; (2) when the learning rate becomes sufficiently big, a two-phase pattern of convergence appears, which does not manifest in traditional optimization analysis and thus indicates that the learning rate is already larger than that permitted by traditional theory, and yet one still has convergence.

More precisely, the 2nd-4th columns are the large learning rate cases, where one observes a shrinkage of the unbalancedness and non-monotonicity of the loss at the beginning, while the right two columns corresponds to the small learning rates for which $\|X_k\|_\mathsf{F}$ and $\|Y_k\|_\mathsf{F}$ barely change as $k$ increases, and the decrease of loss is monotone. These are all evidence of the consistency between the most general case of matrix factorization in Section 5 and the special cases in Section 3 and 4.

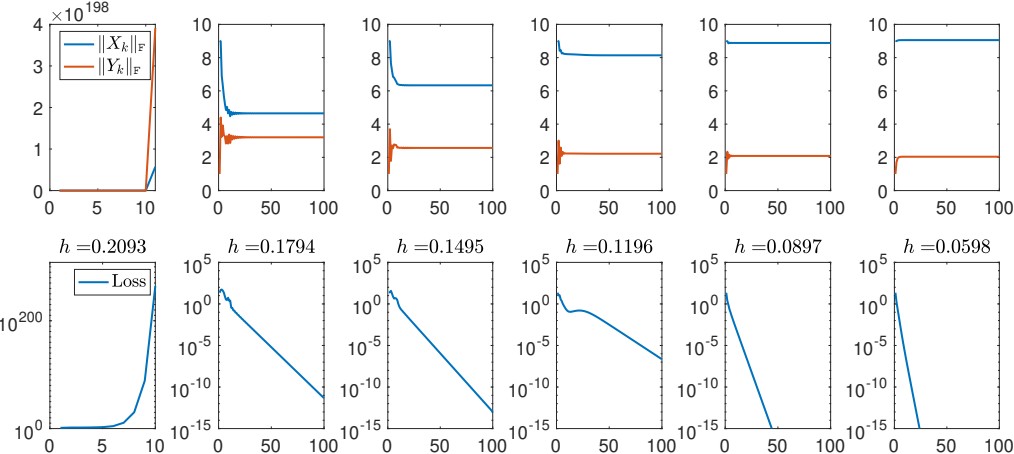

Figure 8: General over-parameterized matrix factorization with $\|X_0\|_\mathsf{F} = 9$ $\|Y_0\|_\mathsf{F} = 1$. The $x-$axis represents the number of iterations $k$; the $y-$axis represents the value for the norm of $X_k$ and $Y_k$ in the first row, and the value for loss in the second row; the learning rate $h$ for each column is the same.

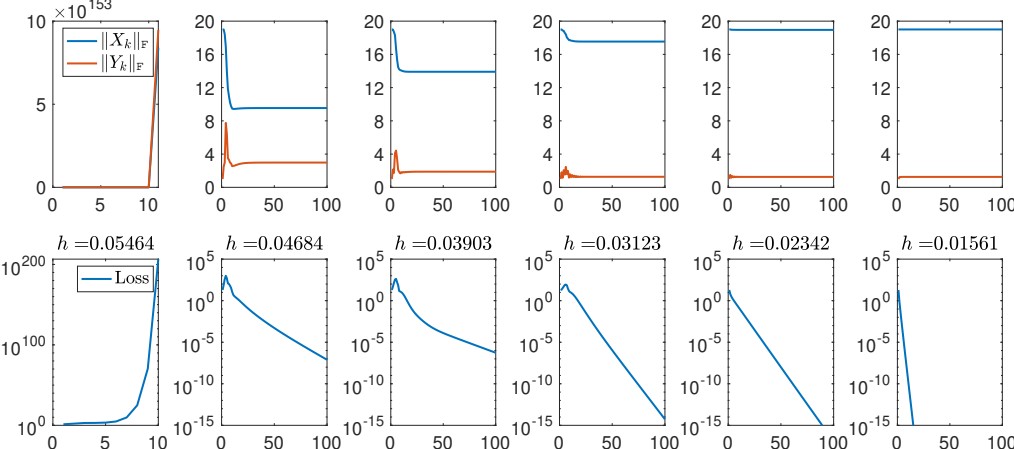

Figure 9: General over-parameterized matrix factorization with $\|X_0\|_\mathsf{F} = 19$ $\|Y_0\|_\mathsf{F} = 1$. The $x-$axis represents the number of iterations $k$; the $y-$axis represents the value for the norm of $X_k$ and $Y_k$ in the first row, and the value for loss in the second row; the learning rate $h$ for each column is the same.

### A.2 MATRIX SENSING AND MATRIX COMPLETION

As we demonstrated, the balancing effect is a nontrivial implicit bias created by large learning rate in GD, and this was rigorously established for the problem of matrix factorization. Matrix factorization already corresponds to a class of important problems as we consider arbitrary $n$ and $d$; however, we feel balancing is an effect even more general, and thus we now demonstrate it empirically on two related additional problems, namely matrix sensing and matrix completion.

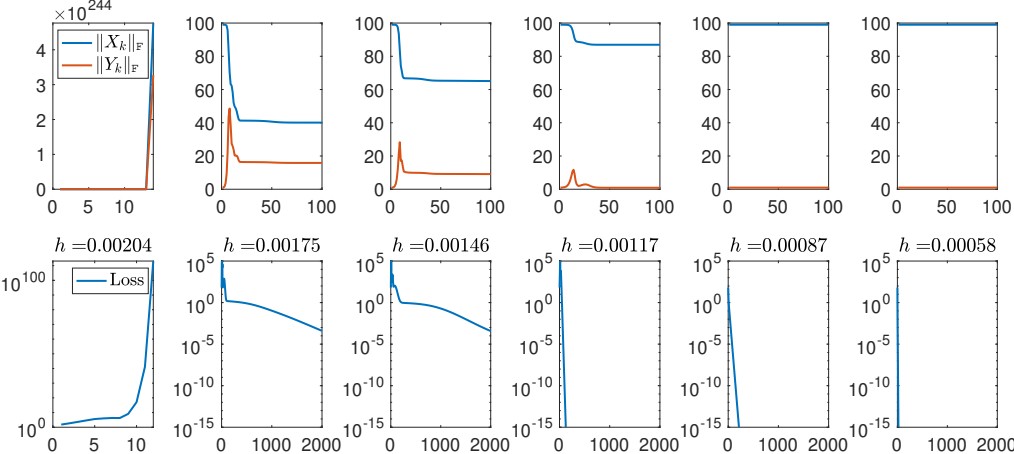

Figure 10: General over-parameterized matrix factorization with $\|X_0\|_\mathsf{F} = 99$ $\|Y_0\|_\mathsf{F} = 1$. The $x-$axis represents the number of iterations $k$; the $y-$axis represents the value for the norm of $X_k$ and $Y_k$ in the first row, and the value for loss in the second row; the learning rate $h$ for each column is the same. Note the $x-$axis range of the 1st row is shortened to better show the changes of $\|X_k\|_\mathsf{F}$ and $\|Y_k\|_\mathsf{F}$ at the beginning of the iterations.

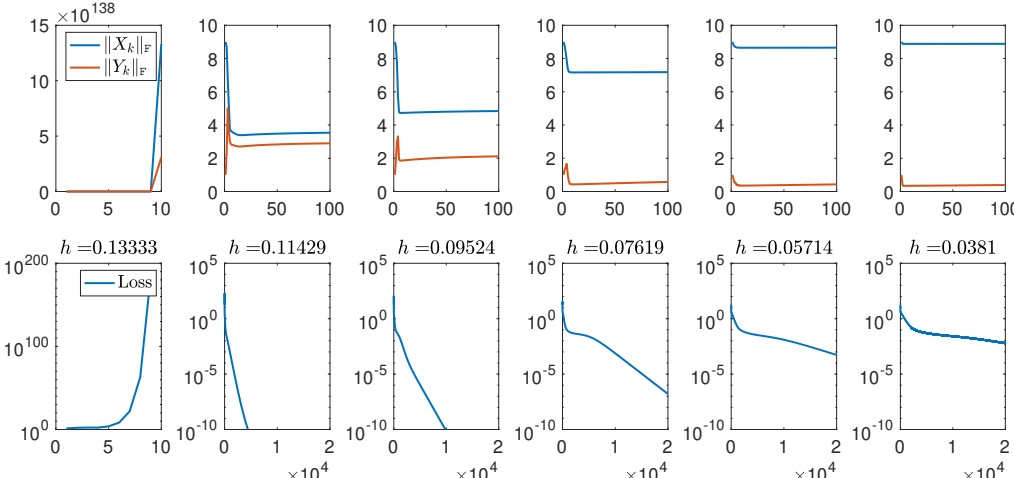

Figure 11: General under-parameterized matrix factorization with $\|X_0\|_\mathsf{F} = 9$ $\|Y_0\|_\mathsf{F} = 1$. The $x-$axis represents the number of iterations $k$; the $y-$axis represents the value for the norm of $X_k$ and $Y_k$ in the first row, and the value for loss in the second row; the learning rate $h$ for each column is the same. Note the $x-$axis range of the 1st row is shortened to better show the changes of $\|X_k\|_\mathsf{F}$ and $\|Y_k\|_\mathsf{F}$ at the beginning of the iterations.

In Figure 13, we consider matrix sensing corresponding to the problem

$$\min_{X,Y \in \mathbb{R}^{n \times d}} \frac{1}{2m} \sum_{i=1}^{m} (b_i - \langle A_i, XY^\top \rangle)^2.$$

Here $A_i \in \mathbb{R}^{100 \times 100}$ are generated from element-wise $i.i.d.$ Gaussian; $b_i \in \mathbb{R}$ are generated from uniform distribution $[0,1]$; $m = 10$; $X, Y \in \mathbb{R}^{100 \times 6}$; $\langle U, V \rangle = \operatorname{tr}(V^\top U)$. The problem is solved via GD. Experiments in the figure correspond to learning rates chosen from large to small as $h_0, \frac{4}{5}h_0, \frac{3}{5}h_0, \frac{2}{5}h_0$ where $h_0$ (the 1st column) is picked near the stability limit. We can see from the figure that matrix sensing exhibits a similar balancing effect with matrix factorization that larger learning rate leads to more balanced norms between $X$ and $Y$.

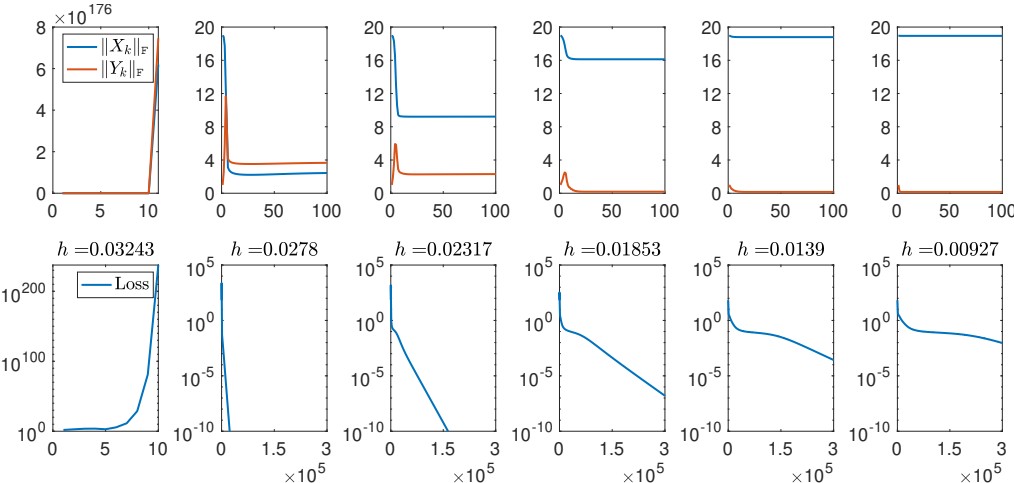

Figure 12: General under-parameterized matrix factorization with $\|X_0\|_\mathsf{F} = 19$ $\|Y_0\|_\mathsf{F} = 1$. The $x-$axis represents the number of iterations $k$; the $y-$axis represents the value for the norm of $X_k$ and $Y_k$ in the first row, and the value for loss in the second row; the learning rate $h$ for each column is the same. Note the $x-$axis range of the 1st row is shortened to better show the changes of $\|X_k\|_\mathsf{F}$ and $\|Y_k\|_\mathsf{F}$ at the beginning of the iterations.

In Figure 14, we consider matrix completion corresponding to the problem

$$\min_{X,Y\in\mathbb{R}^{n\times d}} \frac{1}{2}\left\|P_\Omega(A - XY^T)\right\|_\mathsf{F}^2,$$

where $A \in \mathbb{R}^{10\times10}$ is a low-rank matrix with rank 2; $X, Y \in \mathbb{R}^{10\times2}$; $P_\Omega(U) = (U_{ij})_{(i,j)\in\Omega} + (0)_{(i,j)\notin\Omega}$ with sparsity$-0.6$ where sparsity=(number of non-zero elements)/(number of all elements). The problem is solved via GD. In the above mentioned figure, the learning rates are similarly chosen as the above matrix sensing example. Likewise, the decrease of learning rate results in the increase in the gap between the norms of $X$ and $Y$.

Both the matrix sensing and matrix completion above hold the same homogeneity property between $X$ and $Y$. The "balancing effect" that we proved for matrix factorization is also observed when and only when the learning rate $h$ is large, in which case the norms of $X$ and $Y$ become close at the convergence of GD (and yes, GD still converges even though $h$ is large enough such that the convergence is not monotone).

We feel the techniques invented and employed in this paper can extend to these two cases, but that is beyond the scope of this paper.

## B  GD CONVERGING TO PERIODIC ORBIT

Consider the objective $(1 - xy)^2/2$. Take step size $h = 1.9$. Then GD can converge to periodic orbits with period 2, 3 and 4 respectively in Figure 15.

## C  MODIFIED EQUATION FAILS FOR LARGE LEARNING RATES

Consider the objective function $f(x, y) = (1 - xy)^2/2$. Then the GD update is the following

$$\begin{cases} x_{k+1} = x_k + h(1 - x_k y_k)y_k \\ y_{k+1} = y_k + h(1 - y_k x_k)x_k \end{cases} \Rightarrow \begin{pmatrix} x_{k+1} \\ y_{k+1} \end{pmatrix} = \begin{pmatrix} x_k \\ y_k \end{pmatrix} + h(-\nabla f(x_k, y_k)). \tag{7}$$

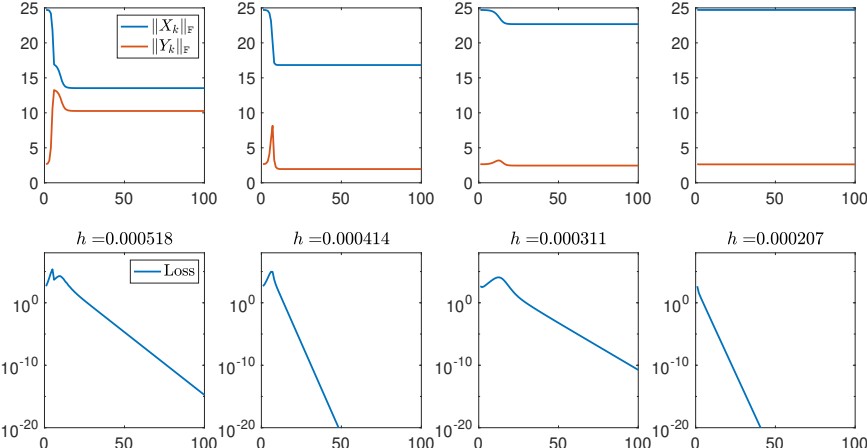

Figure 13: Matrix sensing. The $x-$axis represents the number of iterations $k$; the $y-$axis represents the value for the norm of $X_k$ and $Y_k$ in the first row, and the value for loss in the second row; the learning rate $h$ for each column is the same.

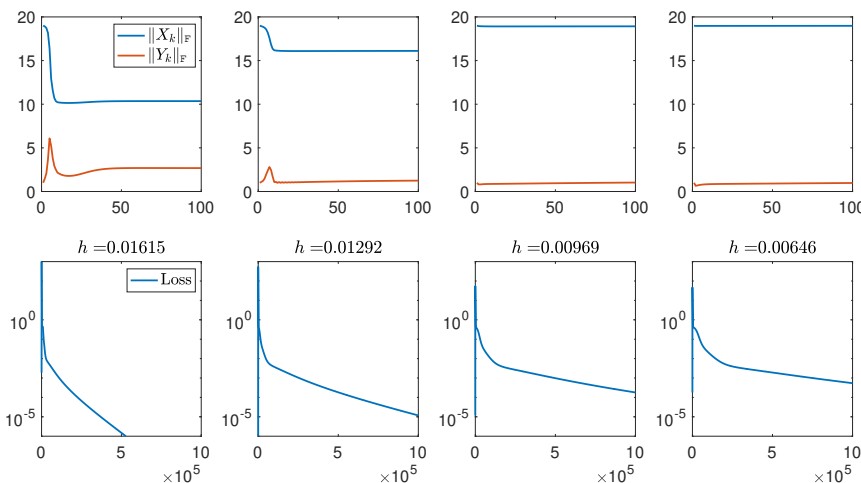

Figure 14: Matrix completion. The $x-$axis represents the number of iterations $k$; the $y-$axis represents the value for the norm of $X_k$ and $Y_k$ in the first row, and the value for loss in the second row; the learning rate $h$ for each column is the same. Note the $x-$axis range of the 1st row is shortened to better show the changes of $\|X_k\|_\mathsf{F}$ and $\|Y_k\|_\mathsf{F}$ at the beginning of the iterations.

By backward error analysis (Hairer et al., 2006, Chapter 9), the modified equation can better approximate GD than gradient flow and is defined as follows

$$\begin{pmatrix} \dot{x} \\ \dot{y} \end{pmatrix} = -\nabla f(x,y) - \frac{h}{2}\nabla^2 f(x,y)\nabla f(x,y) + \mathcal{O}(h^2). \tag{8}$$

Figure 16 shows the trajectories of the 1st order modified equation of (8) and GD (7) with initial condition $x = 4, y = 10$ and $h = 0.026$. The $x$-axis represents the time $t$ and for GD, the time point for $k$th step is $kh$. We compare the absolute values of $x$ and $y$ of both methods due to the symmetry of the global minima $xy = 1$. As is shown in the figure, even if GD almost converges to the most balanced minimizer, the solutions of modified equation are still far away from each other, $x \approx 0.1$ and $y \approx 9$. Actually, large learning rate $h$ fall outside the convergence domain of the modified equation which thus is not an appropriate tool for the analysis of norm balancing.

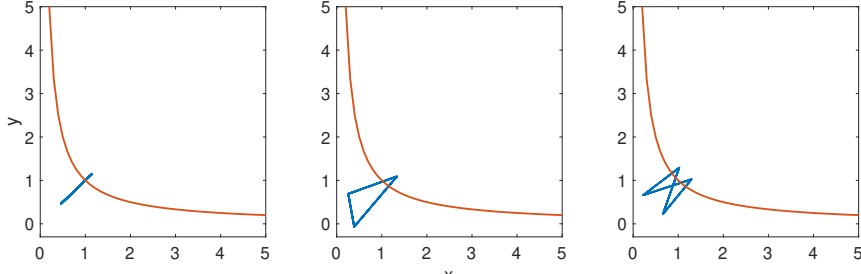

Figure 15: Three orbits of period 2, 3, and 4. The blue line are the orbits; the red line is a reference line of the global minima $xy = 1$.

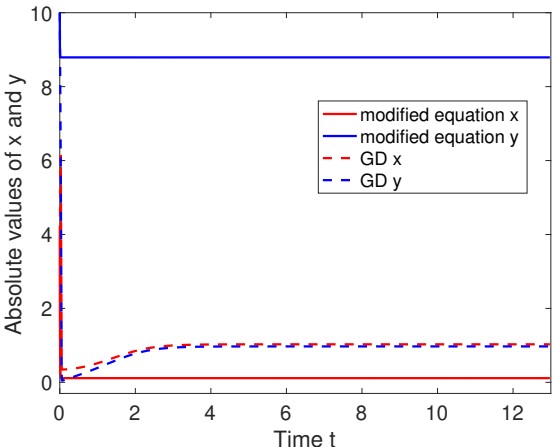

Figure 16: Trajectories: modified equation vs GD

## D    PROOF OF PROPOSITION 2.1

*Proof of Proposition 2.1.* This proposition is a direct consequence of Theorem F.2 and I.3.    □

## E    OVER-PARAMETRIZED SCALAR DECOMPOSITION: STABILITY LIMIT

Consider the objective $(1 - xy)^2/2$. Choose the initial condition to be $x_0 = 20$, $y_0 = 0.07$ and use GD update. The upper bound of $h$ in Theorem 3.1 is $\frac{4}{x_0^2+y_0^2+4\mu}$, where $\mu = 1$. In Figure 17, when $h = \frac{4}{x_0^2+y_0^2+4}$ (the left one), GD converges; however, when $h$ is slightly larger than this bound, it blows up. Hence our restriction for $h$ is very close to the stability limit.

## F    OVERPARAMETRIZED OBJECTIVE: LARGE LEARNING RATE

In this section, we use column vector instead of row vector for sake of better understanding, i.e., our objective function is $(\mu - x^\top y)^2/2$.

### F.1    EIGENVALUES OF HESSIAN

**Lemma F.1** (Matrix determinant lemma). *Suppose $A$ is an invertible $n \times n$ matrix and $u, v \in \mathbb{R}^n$ are column vectors. Then*

$$\det(A + uv^\top) = (1 + v^\top A^{-1} u) \det A.$$

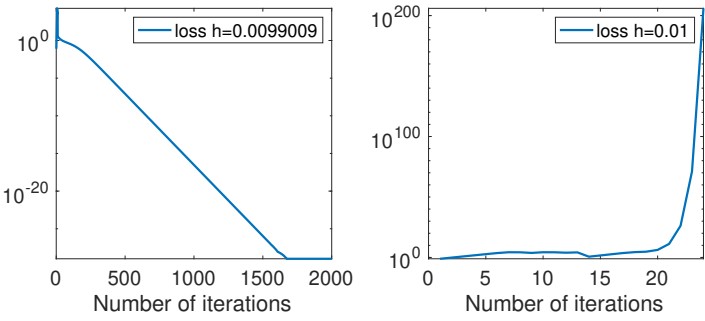

Figure 17: The loss with slightly different $h$ near its upper bound

**Theorem F.2.** *The eigenvalues of the Hessian of $(\mu - x^\top y)^2/2$ are $\pm(\mu - x^\top y)$ repeated $n-1$ times and $\frac{1}{2}\big(\|x\|^2 + \|y\|^2 \pm \sqrt{(\|x\|^2 + \|y\|^2)^2 + 4(\mu - x^\top y)^2 - 8(\mu - x^\top y)x^\top y}\big)$. Especially, at $x^\top y = \mu$, the eigenvalues are $\|x\|^2 + \|y\|^2$ and 0 repeated $2n-1$ times.*

*Proof.* Consider the objective $f(x,y) = (\mu - x^\top y)^2/2$. Its Hessian is the following

$$H = \begin{pmatrix} yy^\top & (x^\top y - \mu)I_n + yx^\top \\ (x^\top y - \mu)I_n + xy^\top & xx^\top \end{pmatrix}$$

$$= \begin{pmatrix} 0 & (x^\top y - \mu)I_n \\ (x^\top y - \mu)I_n & 0 \end{pmatrix} + \begin{pmatrix} y \\ x \end{pmatrix}\begin{pmatrix} y \\ x \end{pmatrix}^\top.$$

Then to calculate the eigenvalues, we also need

$$\lambda I_{2n} - H = \begin{pmatrix} \lambda I_n & (\mu - x^\top y)I_n \\ (\mu - x^\top y)I_n & \lambda I_n \end{pmatrix} - \begin{pmatrix} y \\ x \end{pmatrix}\begin{pmatrix} y \\ x \end{pmatrix}^\top \triangleq B - \begin{pmatrix} y \\ x \end{pmatrix}\begin{pmatrix} y \\ x \end{pmatrix}^\top.$$

By Lemma F.1, we have for invertible $B$

$$\det\left(B - \begin{pmatrix} y \\ x \end{pmatrix}\begin{pmatrix} y \\ x \end{pmatrix}^\top\right) = \left(1 - \begin{pmatrix} y \\ x \end{pmatrix}^\top B^{-1} \begin{pmatrix} y \\ x \end{pmatrix}\right)\det B.$$

Since $(\mu - x^\top y)I_n$ and $\lambda I_n$ commute, we have

$$\det B = \det((\lambda^2 - (\mu - x^\top y)^2)I_n) = (\lambda^2 - (\mu - x^\top y)^2)^n.$$

By the formula for inversion of block matrix, we have for $\lambda^2 - (\mu - x^\top y)^2 \neq 0$,

$$B^{-1} = \begin{pmatrix} \frac{\lambda}{\lambda^2 - (\mu - x^\top y)^2}I_n & -\frac{\mu - x^\top y}{\lambda^2 - (\mu - x^\top y)^2}I_n \\ -\frac{\mu - x^\top y}{\lambda^2 - (\mu - x^\top y)^2}I_n & \frac{\lambda}{\lambda^2 - (\mu - x^\top y)^2}I_n \end{pmatrix}.$$

Then combining all these and by the continuity of characteristic polynomial, we obtain the following expression for all $\lambda$

$$\det(\lambda I_{2n} - H) = (\lambda^2 - (\mu - x^\top y)^2)^{n-1}(\lambda^2 - \lambda(x^\top x + y^\top y) - (\mu - x^\top y)^2 + 2(\mu - x^\top y)x^\top y).$$

When $x^\top y = \mu$, it becomes

$$\det(\lambda I_{2n} - H) = \lambda^{2n-1}(\lambda - (x^\top x + y^\top y)).$$

$\square$

## F.2 LOCAL NON-CONVEXITY NEAR THE GLOBAL MINIMUM

Consider the region $D = \{\delta > x^\top y - \mu > 0\}$, a small neighbourhood of the global minima $\mu = x^\top y$ for some $\delta > 0$. Consider two points $[x,y], [\bar{x}, \bar{y}] \in D$ with $x = c\bar{x}$ and $y = \bar{y}/c$ for $c > 0$ a constant.

Then

$$f(x,y) + \langle \nabla f(x,y), [\bar{x} - x, \bar{y} - y] \rangle = (\mu - x^\top y)^2/2 + (\mu - x^\top y)\big(y^\top (x - \bar{x}) + x^\top (y - \bar{y})\big)$$
$$= (\mu - \bar{x}^\top \bar{y})^2/2 + (2 - c - 1/c)(\mu - \bar{x}^\top \bar{y})\bar{x}^\top \bar{y}$$
$$\geq (\mu - \bar{x}^\top \bar{y})^2/2 = f(\bar{x}, \bar{y}).$$

This contradicts the definition of convexity. Therefore the objective is not locally convex.

### F.3 A REVIEW OF TRADITIONAL CONVERGENCE ANALYSIS OF GD UNDER $L$-SMOOTHNESS

For general function $f$, GD is defined as follows

$$x_{k+1} = x_k - h\nabla f(x_k). \tag{9}$$

**Theorem F.3.** *If $f : \mathbb{R}^N \to \mathbb{R}$ is $L-$smooth, then with $h < \frac{2}{L}$, GD converges to a stationary point.*

*Proof.* Let $\min f = f^*$.

$$f(x_{k+1}) \leq f(x_k) + \langle \nabla f(x_k), x_{k+1} - x_k \rangle + \frac{L}{2}\|x_{k+1} - x_k\|^2$$
$$= f(x_k) - h(1 - \frac{L}{2}h)\|\nabla f(x_k)\|^2.$$

Then

$$\sum_{k=1}^{N} \|\nabla f(x_k)\|^2 \leq \frac{1}{h(1 - \frac{L}{2}h)}(f(x_0) - f(x_N))$$
$$\leq \frac{1}{h(1 - \frac{L}{2}h)}(f(x_0) - f^*).$$

Therefore $\lim_{k \to \infty} \|\nabla f(x_k)\|^2 = 0$, i.e., GD converges to a stationary point. $\square$

## G PROOF OF THEOREM 3.1, THEOREM 3.2, AND COROLLARY 3.3

In this section, we use column vectors for $x$ and $y$ instead of row vectors for better understanding. Then the objective function is $\frac{1}{2}(\mu - x^\top y)^2$.

The following Theorem G.1 and G.2 are the main theorems of the over-parametrized scalar case. Next, for sake of convenience, let $u_k^2 = \|x_k\|^2 + \|y_k\|^2$.

**Theorem G.1** (Main 1). *Let $h = \frac{4}{u_0^2 + 4c\mu}$. Assume $c \geq 1$ and $u_0^2 > 8\mu$. Then GD converges to a point in $\{(x,y) : \|x\|^2 + \|y\|^2 \leq \frac{2}{h}, x^\top y = \mu\}$ except for a Lebesgue measure-0 set of $(x_0, y_0)$.*

*Proof.* By Lemma G.7 and Lemma G.11, the theorem holds. $\square$

**Theorem G.2** (Main 2). *Let $h = \frac{4}{8\mu + 4c\mu} = \frac{1}{(2+c)\mu}$. Assume $c \geq 1$ and $u_0^2 \leq 8\mu$. Then GD converges to a point in $\{(x,y) : \|x\|^2 + \|y\|^2 \leq \frac{2}{h}, x^\top y = \mu\}$ except for a Lebesgue measure-0 set of $(x_0, y_0)$.*

*Proof.* For the measure-0 set, by Lemma G.5, the set of points converging to $\{\|x\|^2 + \|y\|^2 > \frac{2}{h}\}$ is measure-0; by the proof of Lemma G.11, the set of points converging to the origin is measure-0; also $\{u_0^2 = \|x\|^2 + \|y\|^2 = 0\}$ is a hyperplane and thus is measure-0. Hence the set of all the initial conditions not converging to $\{\|x\|^2 + \|y\|^2 > \frac{2}{h}, x^\top y = \mu\}$ is measure-0.

Since $u_0^2 \leq 8\mu$, if $u_0^2 > \frac{2}{h}$, by Lemma G.7, it will decrease to $u_k^2 \leq \frac{2}{h}$ for some $k$. Then by Lemma G.11, we have the convergence. $\square$

With the above two theorems, we can prove all the theorems and corollary in Section 3.

*Proof of Theorem 3.1 and Theorem 3.2.* From Theorem G.1, $h = \frac{4}{u_0^2 + 4c\mu}$ for $c \geq 1$ which implies $h \leq \frac{4}{u_0^2 + 4\mu}$ for all $u_0^2 > 8\mu$. Similarly, for Theorem G.2, $h \leq \frac{1}{3\mu}$. When $u_0^2 = 8\mu$, $\frac{4}{u_0^2 + 4\mu} = \frac{4}{8\mu + 4\mu} = \frac{1}{3\mu}$; when $u_0^2 > 8\mu$, $\frac{4}{u_0^2 + 4\mu} < \frac{1}{3\mu}$; when $u_0^2 < 8\mu$, $\frac{4}{u_0^2 + 4\mu} > \frac{1}{3\mu}$. Also, all the limit points are in $\{\|x\|^2 + \|y\|^2 \leq \frac{2}{h}\}$. Then we can get Theorem 3.1 and 3.2 where the second inequality of Theorem 3.2 is because at the global minimum $x^\top y = \mu$ and also $\|x - y\|^2 = \|x\|^2 + \|y\|^2 - 2x^\top y$. $\qquad\square$

*Proof of Corollary 3.3.* For $|x_0^\top y_0 - \mu| < \delta$, $\|x_0\|^2 + \|y_0\|^2 > 8\mu$, we have

$$x_0^\top y_0 < \mu + \delta \Rightarrow \|x_0 - y_0\|^2 = \|x_0\|^2 + \|y_0\|^2 - 2x_0^\top y_0 > \frac{4}{h} - 4\mu - 2(\mu + \delta) = \frac{4}{h} - 6\mu - 2\delta.$$

Also, from Theorem 3.2, $\|x - y\|^2 \leq \frac{2}{h} - 2\mu$ and $\frac{2}{h} - 2\mu < \frac{4}{h} - 6\mu - 2\delta < \frac{4}{h} - 8\mu$. We then obtain $\|x - y\|^2 \leq \frac{1}{2}\|x_0 - y_0\|^2 + 2\mu$. $\qquad\square$

We will divide the proof of the two main theorems into two phases: (1) when $\frac{2}{h} < x_k^2 + y_k^2 < \frac{4}{h}$, we would like to show that GD escapes to a smaller ball $x_k^2 + y_k^2 \leq \frac{2}{h}$ except for a measure-0 set; (2) once GD enters $x_k^2 + y_k^2 \leq \frac{2}{h}$, it will converge to the global minimum inside this region except for a measure-0 set.

## G.1 Phase 1: $\frac{2}{h} < u_k^2 < \frac{4}{h}$

We first deal with the situation where GD just converges in this region by showing that such points form a null set. This is stated in Theorem G.5.

Theorem G.3 and Corollary G.4 are the preliminary of proving Theorem G.5. Also Corollary G.4 is a direct result of Theorem G.3.

**Theorem G.3.** *Let $f : \mathbb{R}^N \to \mathbb{R}^M$ and $f \in \mathcal{C}^1$. If the set of critical points of $f$ is a null-set, i,e.,*

$$\mathcal{L}(\{x \in \mathbb{R}^N : \nabla f(x) \text{ is not invertible}\}) = 0,$$

*then $\mathcal{L}(f^{-1}(B)) = 0$ for any null-set $B$.*

*Proof.* Let $G = \{|\nabla f| \neq 0\}$ and $G$ is an open set, where $|\nabla f|$ denotes the determinant of $\nabla f$. By implicit function theorem, we have $G \cap f^{-1}(z)$ is a $(N-1)$-submanifold in $\mathcal{C}^1$.
Consider $G \cap \{f \in B\} = \bigcup_{z \in B} G \cap f^{-1}(z)$, where $\mathcal{L}(G \cap f^{-1}(z)) = 0$ from the above discussion. Hence, if $B$ is countable, $\mathcal{L}(G \cap \{f \in B\}) = 0$.

When $B$ is uncountable, using co-area formula from geometric measure theory, we have for any bounded open ball $B_r$,

$$\int_{G \cap B_r} g|\nabla f| d\mathcal{L} = \int_{\mathbb{R}} \int_{f^{-1}(z) \cap G \cap B_r} g(x) d\mathcal{L}_{N-1}(x) dz = \int_B \int_{f^{-1}(z) \cap G \cap B_r} g(x) d\mathcal{L}_{N-1}(x) dz.$$

Let $g$ be the indicator of $\bigcup_{z \in B} G \cap B_r \cap f^{-1}(z)$. Then since $B$ is a null-set, we have

$$\int_B \int_{f^{-1}(z) \cap G \cap B_r} g(x) d\mathcal{L}_{N-1}(x) dz = \int_B \int_{f^{-1}(z) \cap G \cap B_r} d\mathcal{L}_{N-1}(x) dz = 0$$

$$\Rightarrow \int_{G \cap B_r} g|\nabla f| d\mathcal{L} = 0.$$

Then $g|\nabla f| = 0$ a.e. in $G \cap B_r$. Also, $|\nabla f| \neq 0$ a.e.. Therefore, $g = 0$ a.e. in $G \cap B_r$, i.e.,

$$\mathcal{L}\left(\bigcup_{z \in B} G \cap B_r \cap f^{-1}(z)\right) = \int g d\mathcal{L} = 0.$$

Since we can find a sequence of bounded open ball $B_r$, s.t., $\mathbb{R}^N = \bigcup_{r=1}^{\infty} B_r$, and also $\mathcal{L}(G^c) = 0$, we have

$$\mathcal{L}(f^{-1}(B)) \le \mathcal{L}\left(\bigcup_{z \in B} G \cap f^{-1}(z)\right) + \mathcal{L}(G^c) = \mathcal{L}\left(\bigcup_{r=1}^{\infty} \bigcup_{z \in B} G \cap B_r \cap f^{-1}(z)\right) + \mathcal{L}(G^c) = 0.$$

$\square$

**Corollary G.4.** *Let $\psi : \mathbb{R}^{2d} \to \mathbb{R}^{2d}$ be the GD iteration map, i.e., $\psi(x, y) = [x + h(\mu - x^\top y)y, y + h(\mu - x^\top y)x]^\top$. Then if*

$$\mathcal{L}(\{\det(D\psi) = 0\}) = 0,$$

*then $\mathcal{L}(\psi^{-1}(B)) = 0$ for any null-set B.*

**Lemma G.5.** *Given $h = \frac{4}{u_0^2 + 4c\mu}$ and $u_k^2 > \frac{2}{h}$, GD will not converge to the points in $\{\|x\|^2 + \|y\|^2 > \frac{2}{h}, x^\top y = \mu\}$, except for a measure-0 set.*

*Proof.* For $d = 1$,

$$\det(D\psi) = 1 - hx^2 - hy^2 - 3h^2x^2y^2 + 4h^2xy\mu - h^2\mu^2 = p_1(x, y) \cdots p_m(x, y) = 0,$$

where $p_i(x, y)$ is irreducible polynomial and $p_i(x, y) = 0$ is a co-dimensional-1 manifold. Then $\{(x, y) : \det(D\psi) = 0\} = \bigcup_{i=1}^{m}\{p_i(x, y) = 0\}$ are measure zero, i.e., $\mathcal{L}(\{\det(D\psi) = 0\}) = 0$. Similarly for $d > 1$, we also have $\mathcal{L}(\{\det(D\psi) = 0\}) = 0$.

From the GD iteration, we have

$$x_{k+1}^\top y_{k+1} - \mu = (x_k^\top y_k - \mu) \cdot (1 - h(\|x_k\|^2 + \|y_k\|^2) - h^2 x_k^\top y_k(\mu - x_k^\top y_k)).$$

Then let $B = \{(x, y) : 1 - h(\|x\|^2 + \|y\|^2) - h^2 x^\top y(\mu - x^\top y) = 0\}$. For any $(x, y) \in B$, let $[x_+, y_+]^\top = \psi(x, y)$. Then $x_+ y_+ = \mu$. Similarly, $\mathcal{L}(B) = 0$. By Corollary G.4 $\mathcal{L}(\psi^{-1}(B)) = 0$ and then $\mathcal{L}(\psi^{-n}(B)) = \mathcal{L}(\psi^{-1} \circ \cdots \circ \psi^{-1}(B)) = 0$. Let $\psi^{-0}(B) = B$ and $G = \bigcup_{i=0}^{\infty} \psi^{-n}(B)$. Hence

$$\mathcal{L}(G) = \mathcal{L}\left(\bigcup_{i=0}^{\infty} \psi^{-n}(B)\right) \le \sum_{i=0}^{\infty} \mathcal{L}(\psi^{-n}(B)) = 0.$$

Moreover, for any $0 < \epsilon < \mu$, assume $|x_k^\top y_k - \mu| < \epsilon$. When $u_k^2 \ge \frac{2}{h} + \epsilon h(\mu + \epsilon)$,

$$\begin{aligned}
|x_{k+1}^\top y_{k+1} - \mu| &= |x_k^\top y_k - \mu| \cdot |1 - hu_k^2 - h^2 x_k^\top y_k(\mu - x_k^\top y_k)| \\
&= |x_k^\top y_k - \mu| \cdot (-1 + hu_k^2 + h^2 x_k^\top y_k(\mu - x_k^\top y_k)) \\
&\ge |x_k^\top y_k - \mu| \cdot \left(-1 + h\left(\frac{2}{h} + \epsilon h(\mu + \epsilon)\right) - h^2(\mu + \epsilon)\epsilon\right) \\
&> |x_k^\top y_k - \mu|.
\end{aligned}$$

Hence, $\{\|x\|^2 + \|y\|^2 > \frac{2}{h}, x^\top y = \mu\}$ is not the limit of this GD map $\psi$, except for the measure-0 set $G$. $\square$

Next we show that GD will be bounded inside $\{\|x\|^2 + \|y\|^2 < \frac{4}{h}\}$.

**Lemma G.6.** *Given $h = \frac{4}{u_0^2 + 4c\mu}$, then for $0 \le k < \min\{k : u_k^2 \le \frac{2}{h}\}$, we have $u_k^2 \le \frac{4}{h} - 3\mu < \frac{4}{h}$ for all k.*

*Proof.* First $u_0^2 = \frac{4}{h} - 4c\mu \le \frac{4}{h} - 4\mu < \frac{4}{h}$. Then if $x_0^\top y_0 > \mu$ or $x_0^\top y_0 < -\frac{h\mu u_0^2}{4 - hu_0^2}$, from Lemma G.8, $u_1^2 < u_0^2 < \frac{4}{h}$. If $-\frac{h\mu u_0^2}{4 - hu_0^2} \le x_0^\top y_0 < \mu$, from Lemma G.9 and its proof, $u_2^2 < u_0^2 < \frac{4}{h}$ and $u_1 \le u_0^2 + \frac{\mu}{c} \le u_0^2 + \mu \le \frac{4}{h} - 3\mu < \frac{4}{h}$. Therefore, iteratively we have $u_k^2 < \frac{4}{h}$. $\square$

Therefore, without loss of generality, we can just assume $u_k^2 \leq u_0^2$ for a fixed $k$th iteration that we need to analyze because for every two step there exists an $i$th iteration such that $u_i^2 \leq u_0^2$ and we can choose $k = i$ to do the analysis.

Lemma G.8 and G.9 describe one-step or two-step decay of $u_k^2$, which lead to the primary result, Lemma G.7, in phase one. Moreover, the proof of Lemma G.7 contains a finer characterization of $u_k^2$ making it possible to end phase one and enter phase two.

**Lemma G.7.** *Given $h = \frac{4}{u_0^2 + 4c\mu}$ for $c \geq 1$, GD will enter $\{\|x\|^2 + \|y\|^2 \leq \frac{2}{h}\}$ except for a measure-0 set of initial conditions.*

*Proof.* By Lemma G.6 and its discussion, assume without loss of generality $\frac{2}{h} < u_k^2 \leq u_0^2$. From Lemma G.8 and G.9, the region where the decrease of $u_k^2$ may be small is when $x_k, y_k$ are close to $x_k^\top y_k = \mu$ or $x_k^\top y_k = -\frac{h\mu u_k^2}{4 - hu_k^2}$.

Since

$$x_{k+1}^\top y_{k+1} - \mu = (x_k^\top y_k - \mu)(1 - hu_k^2 - h^2 x_k^\top y_k(\mu - x_k^\top y_k)),$$

consider $s_k = 1 - hu_k^2 - h^2 x_k^\top y_k(\mu - x_k^\top y_k)$. From the proof of Lemma G.9, we know when $x_k^\top y_k = -\frac{h\mu u_k^2}{4 - hu_k^2}$, $s_k < -1 + 2h^2\mu^2 < 0$. Therefore, there exists $\delta > 0$, s.t., for all $x_k, y_k \in \{|x_k^\top y_k + \frac{h\mu u_k^2}{4 - hu_k^2}| < \delta, \frac{2}{h} < u_k^2 \leq u_0^2\}$, $s_k < 0$. Then $x_{k+1}^\top y_{k+1} > \mu$. Hence, when $-\delta \leq x_k^\top y_k + \frac{h\mu u_k^2}{4 - hu_k^2} < 0$, we will skip this step and consider the decrease of the next step with $x_{k+1}^\top y_{k+1} > \mu$. Also, there exist $\beta_1 = \beta_1(\delta) > 0$, s.t., when $x_k^\top y_k + \frac{h\mu u_k^2}{4 - hu_k^2} < -\delta$,

$$u_{k+1}^2 - u_k^2 = h(\mu - x_k^\top y_k)((4 - hu_k^2)x_k^\top y_k + hu_k^2\mu) < -\beta_1.$$

From the proof of Lemma G.9, when $-\frac{h\mu u_k^2}{4 - hu_k^2} \leq x_k^\top y_k \leq 0$, for $\beta_2 = \max\{h\mu^2(4 - hu_0^2)(1 - 2h^2\mu^2)^2, h\mu^2(8 - h(2u_0^2 + \frac{\mu}{c}))\} > 0$,

$$u_{k+2}^2 - u_k^2 < \max\{-h(\mu - x_k^\top y_k)\mu(4 - hu_{k+1}^2)(1 - 2h^2\mu^2)^2, -h(\mu - x_k^\top y_k)^2(8 - h(u_k^2 + u_{k+1}^2))\}$$
$$\leq -\beta_2.$$

Fix an small $\epsilon$ in $0 < \epsilon < \mu$. When $x_k^\top y_k \geq \mu + \epsilon$, from the proof of Lemma G.8, we have for $\beta_3 = 4h\mu\epsilon > 0$,

$$u_{k+1}^2 - u_k^2 \leq 4h\mu(\mu - x_k^\top y_k) \leq -4h\mu\epsilon = -\beta_3.$$

When $0 < x_k^\top y_k \leq \mu - \epsilon$ with $\epsilon > 0$, from the proof of Lemma G.9, we have for $\beta_4 = h\epsilon^2(8 - h(2u_0^2 + \frac{\mu}{c})) > 0$,

$$u_{k+2}^2 - u_k^2 \leq -h(\mu - x_k^\top y_k)^2(8 - h(u_k^2 + u_{k+1}^2)) \leq -\beta_4.$$

When $|x_k^\top y_k - \mu| < \epsilon$, assume $|x_k^\top y_k - \mu| = \epsilon_k$. In this case, we have $s_k < 0$ meaning GD oscillates around $x^\top y = \mu$. Also, if $0 < x_k^\top y_k < \mu$ and $u_k^2 > \frac{2}{h}$, $s_k = 1 - hu_k^2 - h^2 x^\top y(\mu - x^\top y) < -1$, i.e., if GD is in $0 < x^\top y < \mu$, then we have $|x_{k+1}^\top y_{k+1} - \mu| > |x_k^\top y_k - \mu|$. Hence we only need to focus on the other side which is $x_k^\top y_k > \mu$. From Lemma G.5, if $u_k^2 \geq \frac{2}{h} + \epsilon h(\mu + \epsilon)$, $|x_{k+1}^\top y_{k+1} - \mu| > |x_k^\top y_k - \mu|$. Then within finite steps (note all these steps satisfy either one-step or two-step decrease of $u_k^2$; we ignore these steps only because the decrease maybe small), we will have either $|x_k^\top y_k - \mu|$ keeps increasing or $u_K^2 < \frac{2}{h} + \epsilon_K h(\mu + \epsilon_K)$ (here we also consider $x_K^\top y_K > \mu$). For the former one, the decrease of $u_k^2$ for each step will be lower bounded away from 0; for the latter one, from the discussion above, $u_{K+1}^2 - u_K^2 \leq -4h\mu\epsilon_K \Rightarrow u_{K+1}^2 \leq \frac{2}{h}$.

From Lemma G.5, we know GD will not terminate in finite steps except for measure-0 set. From all the discussion above, we have that $u_k^2$ decreases by a constant for either one-step or two-step except and hence GD will enter $\|x\|^2 + \|y\|^2 \leq \frac{2}{h}$ except for a measure-0 set. $\qquad\square$

**Lemma G.8.** *Given $h = \frac{4}{u_0^2 + 4c\mu}$ and $u_k^2 < \frac{4}{h}$, when $x_k^\top y_k > \mu$ or $x_k^\top y_k < -\frac{h\mu u_k^2}{4 - hu_k^2}$, we have $u_{k+1}^2 - u_k^2 < 0$.*

*Proof.*
$$u_{k+1}^2 - u_k^2 = h(\mu - x_k^\top y_k)((4 - hu_k^2)x_k^\top y_k + hu_k^2\mu) = h(\mu - x_k^\top y_k)(4x_k^\top y_k + hu_k^2(\mu - x_k^\top y_k))$$
If $x_k^\top y_k > \mu$, by $u_k^2 < \frac{4}{h}$,
$$u_{k+1}^2 - u_k^2 = h(\mu - x_k^\top y_k)(4x_k^\top y_k + hu_k^2(\mu - x_k^\top y_k)) \le 4h\mu(\mu - x_k^\top y_k) < 0.$$
If $x_k^\top y_k < -\frac{h\mu u_k^2}{4 - hu_k^2} \Rightarrow x_k^\top y_k < \mu$, then,
$$u_{k+1}^2 - u_k^2 = h(\mu - x_k^\top y_k)((4 - hu_k^2)x_k^\top y_k + hu_k^2\mu) < 0.$$
$\square$

**Lemma G.9.** *Given $h = \frac{4}{u_0^2 + 4c\mu}$ , $c > \max\{\frac{1}{2}, 2h\mu\}$, $c > 0$, and $u_k^2 \le u_0^2$, when $-\frac{h\mu u_k^2}{4 - hu_k^2} \le x_k^\top y_k < \mu$, then $u_{k+2}^2 - u_k^2 < 0$.*

*Proof.* For every $u_k^2$, there exist a constant $c_k > 0$, s.t. $h = \frac{4}{u_k^2 + 4c_k\mu}$. Since $\frac{2}{h} < u_k^2 \le u_0^2 < \frac{4}{h}$, we have $c \le c_k < \frac{1}{2h\mu}$. Since
$$x_{k+1}^\top y_{k+1} - \mu = (x_k^\top y_k - \mu)(1 - hu_k^2 - h^2 x_k^\top y_k(\mu - x_k^\top y_k)) = (x_k^\top y_k - \mu)s_k,$$
then $s_k = 1 - hu_k^2 - h^2 x_k^\top y_k(\mu - x_k^\top y_k)$ is bounded by the value at $x_k^\top y_k = -\frac{h\mu u_k^2}{4 - hu_k^2}$ or $x_k^\top y_k = \mu$, i.e.,
$$s_k = 1 - hu_k^2 - h^2 x_k^\top y_k(\mu - x_k^\top y_k) \le \max\left\{1 - hu_k^2 + \frac{4h^3 u_k^2 \mu^2}{(4 - hu_k^2)^2}, 1 - hu_k^2\right\}$$
Since $u_k^2 > \frac{2}{h}$, we have $1 - hu_k^2 < -1$. For the other one, since $u_k^2 > \frac{2}{h}$ and $c_k < \frac{1}{2h\mu}$,
$$1 - hu_k^2 + \frac{4h^3 u_k^2 \mu^2}{(4 - hu_k^2)^2} = \frac{(1 - 3c_k^2)u_k^2 + 4c_k^3\mu}{c_k^2(u_k^2 + 4c_k\mu)} < -1 + 2h^2\mu^2,$$
where $c > \frac{h\mu + \sqrt{2 - h^2\mu^2}}{2(1 - h^2\mu^2)}$. This is because either $u_0^2 > 8\mu \Rightarrow h\mu < \frac{1}{3\mu}$ or $h \le \frac{1}{3\mu}$ and then $\frac{h\mu + \sqrt{2 - h^2\mu^2}}{2(1 - h^2\mu^2)} < 1 \le c$. Also note here this is $c$ not $c_k$; this value $-1 + 2h^2\mu^2$ is achieved when $u_k^2 = 2/h$ and $c_k = \frac{h\mu + \sqrt{2 - h^2\mu^2}}{2(1 - h^2\mu^2)}$ or $\frac{1}{2h\mu}$.

Also, when $0 \le x_k^\top y_k < \mu$,
$$s_k = 1 - hu_k^2 - h^2 x_k^\top y_k(\mu - x_k^\top y_k) \le 1 - hu_k^2 < -1.$$

We then prove $4 - hu_{k+1}^2 > 0$. When $x_k^\top y_k < \mu$,
$$u_{k+1}^2 - u_k^2 = h(\mu - x_k^\top y_k)((4 - hu_k^2)x_k^\top y_k + hu_k^2\mu).$$
The maximum is achieved at $x_k^\top y_k = \frac{2 - hu_k^2}{4 - hu_k^2}\mu$, i.e., $u_{k+1}^2 - u_k^2 \le \frac{\mu}{c_k} \le \frac{\mu}{c}$. Hence $4 - hu_{k+1}^2 > 0$ when $c > 1/2$.

Since $x_k^\top y_k < \mu$ and $s_k \le 0$,
$$\begin{aligned}
u_{k+2}^2 - u_k^2 &= u_{k+2}^2 - u_{k+1}^2 + u_{k+1}^2 - u_k^2 \\
&= h(\mu - x_{k+1}^\top y_{k+1})((4 - hu_{k+1}^2)x_{k+1}^\top y_{k+1} + hu_{k+1}^2\mu) \\
&\quad + h(\mu - x_k^\top y_k)((4 - hu_k^2)x_k^\top y_k + hu_k^2\mu) \\
&= h(\mu - x_k^\top y_k)(((4 - hu_k^2)x_k^\top y_k + hu_k^2\mu) + s_k(hu_{k+1}^2\mu + (4 - hu_{k+1}^2)(s_k(x_k^\top y_k - \mu) + \mu))) \\
&= h(\mu - x_k^\top y_k)\big(x_k^\top y_k(4 - hu_k^2) + s_k^2(4 - hu_{k+1}^2)(x_k^\top y_k - \mu) + hu_k^2\mu + s_k(hu_{k+1}^2\mu + (4 - hu_{k+1}^2)\mu)\big).
\end{aligned}$$

When $0 \le x_k^\top y_k < \mu$, $s < -1$, then

$$u_{k+2}^2 - u_k^2 \le -h(\mu - x_k^\top y_k)^2(8 - h(u_k^2 + u_{k+1}^2)) < 0$$

When $-\frac{h\mu u_k^2}{4 - hu_k^2} \le x_k^\top y_k < 0$, $s_k \le 0$, then for $c \ge 2h\mu$,

$$u_{k+2}^2 - u_k^2 < h(\mu - x_k^\top y_k)\mu(-4 + hu_k^2 + 8h^2\mu^2 - (4 - hu_{k+1}^2)(1 - 2h^2\mu^2)^2)$$
$$< -h(\mu - x_k^\top y_k)\mu(4 - hu_{k+1}^2)(1 - 2h^2\mu^2)^2 < 0$$

where this bound is achieved by taking $s_k = -1 + 2h^2\mu^2$ and $x_k^\top y_k = 0$. $\qquad\square$

## G.2 PHASE 2: $u_k^2 \le \frac{2}{h}$

In this part, we will show the convergence of GD in Lemma G.11.

There are two convergence patterns: (i) transversal convergence, i.e., oscillating around the valley; (ii) unilateral convergence, i.e., converging from one side of the valley.

The key point of the proof of pattern (i) is to analyze the change of $s_k = 1 - hu_k^2 - h^2 x_k^\top y_k(\mu - x_k^\top y_k)$. We know $x_{k+1}^\top y_{k+1} - \mu = s_k(x_k^\top y_k - \mu)$, i.e., $s_k$ measures the change of the loss. If for some constant $K > 0$ we have $|s_k| < 1 \, \forall k \ge K$ such that $|x_k^\top y_k - \mu|$ is guaranteed to decrease to 0, then the convergence follows. Moreover, we will need to analyze $s_{2k}$ and $s_{2k+1}$ separately because in this case $s_k < 0$ and $x_k^\top y_k - \mu$ changes sign at each step.

For pattern (ii), we will show that the trajectory of GD is bounded in a subset of this region $|s_k| < 1$ that guarantees the decrease of the loss.

Before presenting our main result in phase 2, we first show a boundedness theorem when GD enters this phase.

**Lemma G.10.** *Once GD enters $\{\|x\|^2 + \|y\|^2 \le \frac{2}{h}\}$, it will stay bounded inside $\{\|x\|^2 + \|y\|^2 \le \frac{2}{h} + 2\mu\}$ and re-enter $\{\|x\|^2 + \|y\|^2 \le \frac{2}{h}\}$ within finite steps where the number of such steps do not depend on the number of iteration.*

*Proof.* If at step $K$, $u_K^2 \le \frac{2}{h}$, then from Lemma G.15, $u_{K+1}^2 \le \frac{2}{h} + \mu$ and it returns to phase 1. From the proof of Lemma G.9, $u_{K+2}^2 \le u_{K+1}^2 + \mu \le \frac{2}{h} + 2\mu$ and we know either $u_{K+2}^2 < u_{K+1}^2$ or $u_{K+3}^2 < u_{K+1}^2$ and so on until it re-enters $u_i^2 \le \frac{2}{h}$ for some $i \ge K$. Therefore, all the $u_k^2 \le \frac{2}{h} + 2\mu$ once GD enters $\{\|x\|^2 + \|y\|^2 \le \frac{2}{h}\}$. $\qquad\square$

Then by Lemma G.10, we can always pick a $k$th iteration such that $u_k^2 \le \frac{2}{h}$. Also, when $u_0^2 > 8\mu$, we have $h\mu < \frac{1}{3}$. Together with the choice of $h = \frac{1}{(2+c)\mu}$ when $0 < u_0^2 \le 8\mu$, we have $h\mu \le \frac{1}{3}$. Then we obtain the following convergence of GD inside phase 2.

**Lemma G.11.** *Given $h\mu \le \frac{1}{3}$, if GD enters $\{\|x\|^2 + \|y\|^2 \le \frac{2}{h}\}$, it converge to $x^\top y = \mu$ inside this region except for a measure-0 set of initial conditions.*

*Proof.* First, from Corollary G.4, we know the set of points converging to $(0,0)$ in finite steps is measure 0. For all the other points, we have the following discussion.

If $s_k = 0$, then $x_{k+1}^\top y_{k+1} = \mu$, i.e., it converges.

If $x_k^\top y_k < -\mu$ and $\frac{1}{h} + 2h\mu^2 < u_k^2 \le \frac{2}{h}$, then by Lemma G.14,

$$u_{k+1}^2 - u_k^2 = h(\mu - x_k^\top y_k)((4 - hu_k^2)x_k^\top y_k + hu_k^2\mu) \le 2h(\mu^2 - (x_k^\top y_k)^2) < 0.$$

Hence for $\delta_1 > 0$, when $-\mu - \delta_1 < x_k^\top y_k < -\mu$, we have $x_{k+1}^\top y_{k+1} > \mu$ and we will skip this step and directly consider the $(k+1)$th iteration; when $x_k^\top y_k \le -\mu - \delta_1$, $u_{k+1}^2 - u_k^2 \le -2h((\mu + \delta_1)^2 - \mu^2)$. Namely when $x_k^\top y_k < -\mu$, either $u_{k+1}^2$ is some constant away from $u_k^2$ or we can ignore the decrease in this step and directly look at the next one with $x_{k+1}^\top y_{k+1} > \mu$.

If $x_k^\top y_k < -\mu$ and $u_k^2 \le \frac{1}{h} + 2h\mu^2$, then

$$
\begin{aligned}
u_{k+1}^2 - u_k^2 &= h(\mu - x_k^\top y_k)(4x_k^\top y_k + hu_k^2(\mu - x_k^\top y_k)) \\
&\le h(\mu - x_k^\top y_k)(4x_k^\top y_k + (1 + 2h^2\mu^2)(\mu - x_k^\top y_k)) \\
&\le -4h\mu^2(1 - 2h^2\mu^2) < 0.
\end{aligned}
$$

If $x_k^\top y_k > 3\mu$, then $u_{k+1}^2 - u_k^2 \le 2h(\mu^2 - x_k^2 y_k^2) < -16h\mu^2$. Also, when $u_k^2 \le 6\mu$, $x_k^\top y_k \le 3\mu$.

Next, for the rest of the region, we first consider $\frac{1}{h} + 2h\mu^2 < u_k^2 \le \frac{2}{h}$ (by Lemma G.10, we are always able to find $u_k^2 \le \frac{2}{h}$). In this region, by Lemma G.18, we know $s_k < 0$, i.e., $(x_k^\top y_k - \mu)(x_{k+1}^\top y_{k+1} - \mu) < 0$. We divide the whole region into two parts: $x_k^\top y_k > \mu$ and $x_k^\top y_k < \mu$. Therefore all the $x_{k+2i}$ should be on the same side for $i$ such that $\frac{1}{h} + 2h\mu^2 < u_{k+2i}^2 \le \frac{2}{h}$.

If $-\mu \le x_k^\top y_k < \mu$ and $s_k \le -1$, by Lemma G.16, we have $u_{k+2}^2 - u_k^2 \le -4h(1 - h\mu)(\mu - x_k^\top y_k)^2 < 0$. Hence when GD does not converge, i.e., $\{(x_{k+2n}, y_{k+2n})\}_{n=0}^\infty$ does not converge , $u_k^2$ will keep decreasing with $u_{k+2i}^2 \le \frac{2}{h}$ for all $K+2i$ with $i \ge 0$ such that $s_{k+2i} \le -1$. Moreover, there exists $N > 0$ s.t. $s_{k+2N} > -1$ because $s_k = 1 - hu_k^2 - h^2 x_k^\top y_k(\mu - x_k^\top y_k)$ and if $x_{k+2i}, y_{k+2i}$ is the first iteration to leave this region, then it has to satisfy $x_{k+2i}^\top y_{k+2i} < -\mu$ which implies $s_{k+2i} > -1$, and then it follows from Lemma G.13 and G.12 just as the following two paragraphs of discussion; if it never leaves this region and GD is not converging, i.e., $|\mu - x_k^\top y_k|$ has a lower bound, then $u_{k+2i}^2$ will keep decreasing until $s_{k+2N} > -1$ (from the proof of Lemma G.12, if $\mu - x_k^\top y_k$ is very small, then $s_{k+1} < s_k$ meaning both sides are blowing up and therefore $|\mu - x_k^\top y_k|$ has a lower bound if $s_k \le -1$).

If $-\mu \le x_k^\top y_k < \mu$ and $s_k > -1$, from Lemma G.12 and Lemma G.13, similar to the previous discussion, there exists $N_1 > 0$, s.t., for all $k > N_1$, $s_k > \min\{-1 + c_0, -1 + c_1(\mu - x_k^\top y_k)^2\}$ for some $c_0, c_1 > 0$, namely $|\mu - x_k^\top y_k|$ strictly decreases. Then it will either converge in $\frac{1}{h} + 2h\mu^2 < \|x\|^2 + \|y\|^2 \le \frac{2}{h}$ or enter $\|x\|^2 + \|y\|^2 \le \frac{1}{h} + 2h\mu^2$.

If $\mu < x_k^\top y_k \le 3\mu$, from the definition of $s_k$, we have $s_k > -1$. Also by Lemma G.13, we know $s_m > \min\{-1 + c_0, -1 + c_1(\mu - x_k^\top y_k)^2\}$ for all $m > k$. Hence similarly, it will either converge or enter $u_k^2 \le \frac{1}{h} + 2h\mu^2$.

Then we consider $u_k^2 \le \frac{1}{h} + 2h\mu^2$. From Lemma G.17 and Lemma G.20, we know $|s_k| < 1$ and will be away from 1 by a constant in this area. We will show that GD stays bounded in the converging domain. If $s_k \le 0$, it follows from the previous discussion and the proof of Lemma G.17 and Lemma G.20. If $s_k > 0$, when the trajectory is in $\{|x^\top y| > \mu\}$, from Lemma G.14, we have $u_{k+1}^2 < u_k^2$; when it is in $\{|x^\top y| < \mu\}$, then $|x_{k+1} - y_{k+1}| < |x_k - y_k|$ from Lemma G.19. Hence the trajectory will stay inside the monotone decreasing region. Next, since $|\mu - x_k^\top y_k|$ monotonically decreases and is lower bounded by 0, we have that $|\mu - x_k^\top y_k|$ converges. If $|\mu - x_k^\top y_k|$ converges to $C > 0$, then $\min\{-1 + c_0, -1 + c_1(\mu - x_k^\top y_k)^2\} < s_k < 1 - c_2$ for some $c_0, c_1, c_2 > 0$ meaning $|\mu - x_k^\top y_k|$ does not converges to C. Contradiction. Therefore, $|\mu - x_k^\top y_k|$ converges to 0. □

**Lemma G.12.** *When $-\mu \le x_k^\top y_k < \mu$ and $\frac{1}{h} + 2h\mu^2 < u_k^2 \le \frac{2}{h} + 2\mu$, if $s_k > -1$, then $s_{k+2} > \min\{-\frac{3}{4} - \frac{h^2\mu^2}{4}, -\frac{5h^2\mu^2}{2}, -1 + h^2(\mu - x_k^\top y_k)^2\} > -1$, for $\frac{9}{2}h^2\mu^2 < 1$.*

*Proof.* From previous statement, $s_k < 0$ when $u_k^2 > \frac{1}{h} + 2h\mu^2$. For $s_k = 1 - hu_k^2 - h^2 x_k^\top y_k(\mu - x_k^\top y_k)$, let $\epsilon = \mu - x_k^\top y_k$. Then $x_{k+1}^\top y_{k+1} - \mu = -s_k \epsilon$.

$$
s_{k+1} = 1 - hu_{k+1}^2 - h^2 x_{k+1}^\top y_{k+1}(\mu - x_{k+1}^\top y_{k+1}) = s_k - h^2 \epsilon\mu(3 + s_k) + h^2\epsilon^2(3 + s_k^2 - hu_k^2).
$$

Hence if $\epsilon$ is very small, we have $s_{k+1} < s_k$. Also, $x_{k+2}y_{k+2} - \mu = s_{k+1}(x_{k+1}^\top y_{k+1} - \mu) = -s_{k+1}s_k\epsilon$. Moreover, $u_{k+1}^2 \le \frac{2}{h} + 2\mu$ by Lemma G.6.

$$\begin{aligned}
s_{k+2} &= 1 - hu_{k+2}^2 - h^2 x_{k+2}^\top y_{k+2}(\mu - x_{k+2}^\top y_{k+2}) \\
&= s_k - h^2\epsilon\mu(3 + 4s_k + s_k s_{k+1}) + h^2\epsilon^2(3 + s_k^2 - hu_k^2 + s_k^2(3 + s_{k+1}^2 - hu_{k+1}^2))) \\
&\geq s_k - h^2\epsilon\mu(3 + 4s_k + s_k s_{k+1}) + h^2\epsilon^2(1 + (1 + s_{k+1}^2)s_k^2) \\
&\overset{s_{k+1} = \mu/(2s_k\epsilon)}{\geq} s_k - \frac{h^2\mu^2}{4} - h^2\epsilon(3 + 4s_k) + h^2\epsilon^2(1 + s_k^2)
\end{aligned}$$

If $3 + 4s_k \geq 0$, then

$$\begin{aligned}
s_{k+2} &\geq s_k - \frac{h^2\mu^2}{4} - h^2\epsilon(3 + 4s_k) + h^2\epsilon^2(1 + s_k^2) \\
&\overset{\epsilon = (3+4s_k)\mu/(2(1+s_k^2))}{\geq} s_k - \frac{h^2\mu^2}{4} - \frac{h^2\mu^2(3 + 4s_k)^2}{4(1 + s_k^2)} \\
&\overset{s_k = 0 \text{ or } s_k = -3/4}{\geq} \min\{-\frac{3}{4} - \frac{h^2\mu^2}{4}, -\frac{5h^2\mu^2}{2}\} > -1.
\end{aligned}$$

If $3 + 4s_k < 0$, i.e., $-1 < s_k < -\frac{3}{4}$, then

$$\begin{aligned}
s_{k+2} &= s_k - h^2\epsilon\mu(3 + 4s_k + s_k s_{k+1}) + h^2\epsilon^2(3 + s_k^2 - hu_k^2 + s_k^2(3 + s_{k+1}^2 - hu_{k+1}^2)) \\
&= s_k - h^2\epsilon\mu(3 + 4s_k + s_k(s_k - h^2\epsilon\mu(3 + s_k) + h^2\epsilon^2(3 + s^2 - hu_k^2))) \\
&\quad + h^2\epsilon^2(3 + s_k^2 - hu_k^2 + s_k^2(3 + s_{k+1}^2 - hu_{k+1}^2)) \\
&= s_k - h^2\mu\epsilon(s_k(3 + s_k) + 3 + s_k) - h^4\mu\epsilon^3 s_k(3 + s_k^2 - hu_k^2) \\
&\quad + h^2\epsilon^2(3 + s_k^2 - hu_k^2 + s_k^2(3 + s_{k+1}^2 - hu_{k+1}^2) + h^2\mu^2 s_k(3 + s_k)).
\end{aligned}$$

When $-1 < s_k < \frac{1 - 3h^2\epsilon\mu}{h^2\epsilon\mu}$, $s_k - h^2\mu\epsilon(s_k(3 + s_k) + 3 + s_k) > -1$. Since $\epsilon \leq 2\mu$, we have $\frac{1 - 3h^2\epsilon\mu}{h^2\epsilon\mu} \geq \frac{1 - 6h^2\mu^2}{2h^2\mu^2} \geq -\frac{3}{4}$ for $\frac{9}{2}h^2\mu^2 < 1$. Also,

$$\begin{aligned}
&- h^4\mu\epsilon^3 s_k(3 + s_k^2 - hu_k^2) > \frac{3}{4}h^4\mu\epsilon^3 > 0, \\
&h^2\epsilon^2(3 + s_k^2 - hu_k^2 + s_k^2(3 + s_{k+1}^2 - hu_{k+1}^2) + h^2\mu^2 s_k(3 + s_k)) > h^2\epsilon^2 > 0.
\end{aligned}$$

Hence

$$\begin{aligned}
s_{k+2} &> -1 - h^4\mu\epsilon^3 s_k(3 + s_k^2 - hu_k^2) \\
&\quad + h^2\epsilon^2(3 + s_k^2 - hu_k^2 + s_k^2(3 + s_{k+1}^2 - hu_{k+1}^2) + h^2\mu^2 s_k(3 + s_k)) > -1 + h^2\epsilon^2
\end{aligned}$$

$\square$

**Lemma G.13.** When $x_k^\top y_k > \mu$ and $\frac{1}{h} + 2h\mu^2 < u_k^2 \leq \frac{2}{h} + 2\mu$, if $s_k > -1$, then $s_{k+1} > s_k$ and $s_{k+2} > \min\{-1 + h^2(\mu - x_k^\top y_k)^2(\frac{1}{2} - \frac{3}{2}h^2\mu^2), -\frac{1}{2} - \frac{3}{2}h^4(\mu - x_k^\top y_k)^2\mu^2\} > -1$, for $8h^2\mu^2 < 1$. Moreover, if $u_{k+1}^2 \leq \frac{2}{h}$, $s_k \leq -1$, and $s_{k+1} > -1$, then $s_{k+2} > s_k + h^2(\mu - x_k^\top y_k)^2(1 - 8h^2\mu^2)$.

*Proof.* From previous statement, $s_k < 0$ when $u_k^2 > \frac{1}{h} + 2h\mu^2$. Without loss of generality, we can just assume $u_k^2 < \frac{3}{h}$ (otherwise $u_{k+2}^2 < \frac{3}{h}$ and we can use this as our starting point). For $x_k^\top y_k > \mu$, $s_k = 1 - hu_k^2 - h^2 x_k^\top y_k(\mu - x_k^\top y_k)$. Let $x_k^\top y_k - \mu = \epsilon$. Then

$$s_{k+1} = 1 - hu_{k+1}^2 - h^2 x_{k+1}^\top y_{k+1}(\mu - x_{k+1}^\top y_{k+1}) = s_k + h^2\epsilon\mu(3 + s_k) + h^2\epsilon^2(3 + s_k^2 - hu_k^2) > s_k,$$

where $s_k > -3$ in this region. Also, $x_{k+2}^\top y_{k+2} - \mu = s_{k+1}s_k\epsilon$.

Then for $s_k \leq -1$, $u_{k+1}^2 \leq \frac{2}{h}$, and $s_{k+1} > -1$,

$$\begin{aligned}
s_{k+2} &= s_k + h^2\epsilon\mu(3 + 4s_k + s_k s_{k+1}) + h^2\epsilon^2(3 + s_k^2 - hu_k^2 + s_k^2(3 + s_{k+1}^2 - hu_{k+1}^2)) \\
&= s_k + h^2\epsilon\mu(3 + 4s_{k+1} + s_{k+1}^2) + h^2\epsilon^2\big[(3 + s_k^2 - hu_k^2)(1 - 4h^2\epsilon\mu - h^2\epsilon\mu s_{k+1}) \\
&\quad + s_k^2(2 - hu_{k+1}^2) + s_k^2(1 + s_{k+1}^2) - h^2\mu^2(4 + s_{k+1})(3 + s_k)\big] \\
&> s_k + h^2\epsilon^2(1 - 8h^2\mu^2)
\end{aligned}$$

where the inequality is achieved by $s_{k+1} > -1, u_k^2 \leq \frac{4}{h}, u_{k+1}^2 \leq \frac{2}{h}, 8h^2\mu^2 < 1, \epsilon \leq 2\mu$.

For $s_k > -1$,

$$
\begin{aligned}
s_{k+2} &= s_k + h^2\epsilon\mu(3 + 4s_k + s_k^2) + h^2\epsilon^2\big[(3 + s_k^2 - hu_k^2)(1 + s_kh^2\epsilon\mu) \\
&\quad + s_k^2(3 + s_{k+1}^2 - hu_{k+1}^2) + h^2\mu^2 s_k(3 + s_k)\big] \\
&> s_k + h^2\epsilon\mu(3 + 4s_k + s_k^2) + h^2\epsilon^2\big[s_k^2(1 + 2s_kh^2\mu^2) + s_k^2 + h^2\mu^2 s_k(3 + s_k)\big] \\
&= s_k + h^2\epsilon\mu(3 + 4s_k + s_k^2) + h^2\epsilon^2\big[2s_k^2 + h^2\mu^2 s_k(3 + 2s_k^2 + s_k)\big]
\end{aligned}
$$

When $s_k > -1$, we have $s_k + h^2\epsilon\mu(3 + 4s_k + s_k^2) > -1$ and $h^2\epsilon\mu(3 + 4s_k + s_k^2) > 0$. Since $8h^2\mu^2 < 1$, $2s_k^2 + h^2\mu^2 s_k(3 + 2s_k^2 + s_k) \geq \frac{1}{2} - \frac{3}{2}h^2\mu^2 > 0$ for $-1 < s_k \leq -\frac{1}{2}$. For $-\frac{1}{2} < s_k \leq 0$, $2s_k^2 + h^2\mu^2 s_k(3 + 2s_k^2 + s_k) \geq h^2\mu^2 s_k(3 + 2s_k^2 + s_k) \geq -\frac{3}{2}h^2\mu^2 > -\frac{1}{2}$. Hence $s_{k+2} > \min\{-1 + h^2\epsilon^2(\frac{1}{2} - \frac{3}{2}h^2\mu^2), -\frac{1}{2} - \frac{3}{2}h^4\epsilon^2\mu^2\} > -1$.

$\square$

**Lemma G.14.** *Given $u_k^2 \leq \frac{2}{h}$, when $|x_k^\top y_k| > \mu$, $u_{k+1}^2 - u_k^2 < 0$.*

*Proof.*

$$
\begin{aligned}
u_{k+1}^2 - u_k^2 &= h(\mu - x_k^\top y_k)(4x_k^\top y_k + hu_k^2(\mu - x_k^\top y_k)) \\
&= h(\mu - x_k^\top y_k)4x_k^\top y_k + h^2 u_k^2(\mu - x_k^\top y_k)^2 \\
&\leq h(\mu - x_k^\top y_k)4x_k^\top y_k + 2h(\mu - x_k^\top y_k)^2 \\
&= 2h(\mu^2 - (x_k^\top y_k)^2) < 0
\end{aligned}
$$

$\square$

**Lemma G.15.** *Given $u_k^2 \leq \frac{2}{h}$, $u_{k+1}^2 - u_k^2 \leq \mu$.*

*Proof.*

$$
u_{k+1}^2 - u_k^2 = h(\mu - x_k^\top y_k)(4x_k^\top y_k + hu_k^2(\mu - x_k^\top y_k)) \leq \frac{4h\mu^2}{4 - hu_k^2},
$$

where the maximum is achieved at $x_k^\top y_k = \frac{2 - hu_k^2}{4 - hu_k^2}\mu$. When $h = \frac{4}{u_0^2 + 4c\mu}$, from the proof of Lemma G.9, $u_{k+1} - u_k^2 \leq \mu$. When $h = \frac{1}{(2+c)\mu}$ and in this case $u_k^2 \leq \frac{2}{h} = (1 + \frac{c}{2}\mu)$, $u_{k+1} - u_k^2 \leq \frac{8}{7(2+c)}\mu < \mu$. $\square$

**Lemma G.16.** *When $-\mu \leq x_k^\top y_k < \mu$, if $s_k = 1 - hu_k^2 - h^2 x_k^\top y_k(\mu - x_k^\top y_k) \leq -1$, $u_{k+2}^2 - u_k^2 \leq -h(4 - 4h\mu)(\mu - x_k^\top y_k)^2 < 0$.*

*Proof.*

$$
\begin{aligned}
u_{k+2}^2 - u_k^2 &= h(\mu - x_k^\top y_k)(((4 - hu_k^2)x_k^\top y_k + hu_k^2\mu) + s(hu_{k+1}^2\mu + (4 - hu_{k+1}^2)(s(x_k^\top y_k - \mu) + \mu))) \\
&= h(\mu - x_k^\top y_k)\left(-(4 - hu_k^2 + s^2(4 - hu_{k+1}^2))(\mu - x_k^\top y_k) + 4(1 + s)\mu\right) \\
&\leq -h(8 - h(u_k^2 + u_{k+1}^2))(\mu - x_k^\top y_k)^2 \\
&\leq -h(4 - 4h\mu)(\mu - x_k^\top y_k)^2 < 0
\end{aligned}
$$

where the first inequality is because when $s_k = -1$ it has the maximum value; the second is from Lemma G.10. $\square$

**Lemma G.17.** *When $u_k^2 \leq \frac{2}{h}$, then $s_k < 1$ and will be away from 1 by a constant.*

*Proof.*

$$s_k = 1 - hu_k^2 - h^2 x_k^\top y_k(\mu - x_k^\top y_k) \le 1 - hu_k^2 + h^2\frac{u_k^2}{2}\left(\mu + \frac{u_k^2}{2}\right)$$

where the equality is achieved when $x_k^\top y_k = -\frac{u_k^2}{2}$.

Since $0 < u_k^2 \le \frac{2}{h}$,

$$s_k \le 1 - hu_k^2 + h^2\frac{u_k^2}{2}\left(\mu + \frac{u_k^2}{2}\right) \le \max\{1, h\mu\} < 1.$$

Actually this $s_k$ will be away from 1 by a constant. This is because when $u_k^2$ are close to 0, $s_k$ will be close to 1; however, $u_k^2$ will be increasing because of the decrease of $|\mu - x_k^\top y_k|$ and thus $s_k$ will decrease accordingly. $\square$

**Lemma G.18.** *When $u_k^2 > \frac{1}{h} + 2h\mu^2$ and $-\mu \le x_k^\top y_k \le 3\mu$, $(x_{k+1}^\top y_{k+1} - \mu)(x_k^\top y_k - \mu) < 0$, meaning it oscillates around $x^\top y = \mu$.*

*Proof.*

$$1 - hu_k^2 - h^2 x_k^\top y_k(\mu - x_k^\top y_k) < 1 - h(\frac{1}{h} + 2h\mu^2) + 2h^2\mu^2 = 0$$

$\square$

**Lemma G.19.** *When $-\mu \le x_k^\top y_k < \mu$ and $h\mu \le \frac{1}{3}$, we have $|x_{k+1} - y_{k+1}| < |x_k - y_k|$.*

*Proof.*

$$|x_{k+1} - y_{k+1}| = |x_k - y_k|\,|1 - h(\mu - x_k^\top y_k)|.$$

Since $h\mu < 1$, then

$$-1 < 1 - 2h\mu \le 1 - h(\mu - x_k^\top y_k) < 1.$$

$\square$

**Lemma G.20.** *When $u_k^2 \le \frac{1}{h} + 2h\mu^2$ and $h\mu \le \frac{1}{3}$, $s_k \ge -\frac{1}{4}$.*

*Proof.* By $h\mu \le \frac{1}{3}$,

$$1 - hu_k^2 - h^2 x_k^\top y_k(\mu - x_k^\top y_k) \ge -2h^2\mu^2 - \frac{h^2\mu^2}{4} \ge -\frac{9h^2\mu^2}{4} \ge -\frac{1}{4}.$$

$\square$

# H   PROOF OF THEOREM 4.1, THEOREM 4.2, AND THEOREM 4.3

The proof in this section mainly follows the strategy of the scalar case. More precisely, all the expressions, e.g., $u_{k+1}^2$ and $x_{k+1}^\top y_{k+1} - \mu$, in this rank-1 approximation case contains two parts: the one that can be handled using the technique in scalar case, and the other one measuring the extent of alignment between $x_k$ and $y_k$. Again, let $u_k^2 = \|x_k\|^2 + \|y_k\|^2$.

**Theorem H.1** (Main 1). *Let $h = \frac{4}{u_0^2 + 4c\mu}$. Assume $c \ge \sqrt{7}$ and $u_0^2 > 4\sqrt{7}\mu$. Then GD converges to a point in $\{(x, y) : \|x\|^2 + \|y\|^2 \le \frac{2}{h}, \ x^\top y = \mu\}$ except for a Lebesgue measure-0 set of $(x_0, y_0)$.*

*Proof.* The proof follows from Lemma H.3 and Lemma H.8. $\square$

**Theorem H.2** (Main 2). *Let $h = \frac{4}{4\sqrt{7}\mu + 4c\mu} = \frac{1}{(\sqrt{7}+c)\mu}$. Assume $c \geq \sqrt{7}$ and $u_0^2 \leq 4\sqrt{7}\mu$. Then GD converges to a point in $\{(x,y) : \|x\|^2 + \|y\|^2 \leq \frac{2}{h}, \ x^\top y = \mu\}$ except for a Lebesgue measure-0 set of $(x_0, y_0)$.*

*Proof.* Since $c \geq \sqrt{7}$, in this case we have $u_0^2 \leq 4\sqrt{7}\mu \leq 2(\sqrt{7}+c)\mu = \frac{2}{h}$. Therefore, the convergence follows from Lemma H.8. □

Hence we can show the proof of the theorems in Section 4.

*Proof of Theorem 4.1.* By Lemma H.3 and H.7, the proof follows from Lemma H.9. □

*Proof of Theorem 4.2 and 4.3.* Similar to scalar case, $h = \frac{4}{u_0^2 + 4c\mu}$ for $c \geq \sqrt{7}$ implies $h \leq \frac{4}{u_0^2 + 4\sqrt{7}\mu}$ and $h = \frac{1}{(\sqrt{7}+c)\mu}$ implies $h \leq \frac{1}{2\sqrt{7}}$. Therefore, the convergence and balancing of GD follow from Theorem H.1 and H.2. The second inequality of Theorem 4.3 follows from $x^\top y = \mu$ at the global minimum because of diagonal and non-negative $A$ and the best rank-1 approximation in SVD. □

## H.1 Phase 1

The main theorem in phase 1 is the following.

**Lemma H.3.** *Given $h = \frac{4}{u_0^2 + 4c\mu}$ for $c \geq \sqrt{7}$, GD will enter $\{\|x\|^2 + \|y\|^2 \leq \frac{2}{h}\}$ except for a measure-0 set of initial conditions.*

*Proof.* Since the expressions, $u_{k+2}^2 - u_k^2$ and $u_{k+2}^2 - u_k^2$, of the rank-1 version have an additional term upper bounded by $c_0((x_k^\top y_k)^2 - x_k^\top x_k y_k^\top y_k)$ for some constant $c_0 > 0$ which is non-positive, we have $u_k^2$ decreases by a constant for either one- or two-step if $(x_k^\top y_k)^2 - x_k^\top x_k y_k^\top y_k < -c_1$ for some constant $c_1 > 0$ when $\frac{2}{h} < u_k^2 < \frac{4}{h}$. Therefore, we can pick a $K > 0$ s.t. $(x_k^\top y_k)^2 - x_k^\top x_k y_k^\top y_k < -c_1$ for $k \leq K$. If GD enters $\{\|x\|^2 + \|y\|^2 \leq \frac{2}{h}\}$ within $K$ steps, then we are done. Otherwise, we can just assume that for all the $k > K$, $-c_1 \leq (x_k^\top y_k)^2 - x_k^\top x_k y_k^\top y_k \leq 0$. Then again, like the scalar case, we only need to consider when $x_k, y_k$ are close to $x_k^\top y_k = \mu$ and $x_k^\top y_k = -\frac{h\mu u_k^2}{4 - h u_k^2}$. For the region near $x_k^\top y_k = -\frac{h\mu u_k^2}{4 - h u_k^2}$, we can always choose a $c_1$ s.t. there exists a $\delta > 0$, when $-\delta \leq x_k^\top y_k + \frac{h\mu u_k^2}{4 - h u_k^2} < 0$, we have both $x_{k+1}^\top y_{k+1} > \mu$ and $s_k < 0$. Then all the discussion just follows the scalar case. For the region near $x_k^\top y_k = \mu$, i.e., $|x_k^\top y_k - \mu| < \epsilon$ for some $0 < \epsilon < \mu$, consider

$$x_{k+1}^\top y_{k+1} - \mu = (x_k^\top y_k - \mu)s_k + h^2(2\mu - x_k^\top y_k)((x_k^\top y_k)^2 - x_k^\top x_k y_k^\top y_k).$$

Similar to scalar decomposition, we only need to consider the case when $s_k \leq -1$ because if $s_k > -1$ which can only happen when $x_k^\top y_k > \mu$, we have $u_k^2 \leq \frac{2}{h} + \epsilon_k h(\mu + \epsilon)$ and then $u_{k+1}^2 \leq \frac{2}{h}$. If $s_k \geq -1$ for all the following iterations, together with the bounded second term in $x_{k+1}^\top y_{k+1} - \mu$, i.e., $-2h^2\mu c_1 \leq h^2(2\mu - x_k^\top y_k)((x_k^\top y_k)^2 - x_k^\top x_k y_k^\top y_k) \leq 0$, we have that there exists $K > 0$ s.t. $|x_K^\top y_K - \mu| \geq \epsilon_1$ for some $\epsilon_1 > 0$ unless it leaves this region $|x_k^\top y_k - \mu| < \epsilon$. Therefore, by the same discussion in scalar case, $u_k^2$ will decrease either in one step or two steps by at least a constant.

Moreover, by Lemma H.4, GD will not terminate inside $\{\frac{2}{h} < \|x\|^2 + \|y\|^2 < \frac{4}{h}\}$ in finite step. Thus, GD is guaranteed to enter $\{\|x\|^2 + \|y\|^2 \leq \frac{2}{h}\}$ in finite steps. □

**Lemma H.4.** *GD will not converge to any fixed points in $\{\|x\|^2 + \|y\|^2 > \frac{2}{h}\}$ except for a measure-0 set.*

*Proof.* Similar to the discussion in scalar case, the set of points converging to $x^\top y = \mu$ in finite step is measure-0.

Also, since $\mu x^\top y = x^\top x y^\top y = \mu^2$, the fixed points of GD map requires $x^\top x y^\top y = (x^\top y)^2$. This can be seen in the following expression

$$x_{k+1}^\top y_{k+1} - \mu = (x_k^\top y_k - \mu)(1 - hu_k^2 - h^2 x_k^\top y_k(\mu - x_k^\top y_k)) + h^2(2\mu - x_k^\top y_k)((x_k^\top y_k)^2 - x_k^\top x_k y_k^\top y_k),$$

where the convergence of $x_k^\top y_k - \mu$ requires the convergence of $x_k^\top x_k y_k^\top y_k - (x_k^\top y_k)^2$.

Hence if $x_k^\top x_k y_k^\top y_k - (x_k^\top y_k)^2$ does not converge, then GD will not converge. If $x_k^\top x_k y_k^\top y_k - (x_k^\top y_k)^2$ converges, then $c \geq x_k^\top x_k y_k^\top y_k - (x_k^\top y_k)^2 \geq 0$ for all the rest of the iterations with some $c > 0$, and we further assume $|x_k^\top y_k - \mu| < \epsilon$ for any $0 < \epsilon < \mu$. As is shown in scalar case, when $u_k^2 \geq \frac{2}{h} + \epsilon h(\mu + \epsilon)$, we have $s_k = 1 - hu_k^2 - h^2 x_k^\top y_k(\mu - x_k^\top y_k) \geq -1$. Then

$$x_{k+1}^\top y_{k+1} - \mu = (x_k^\top y_k - \mu)s_k + h^2(2\mu - x_k^\top y_k)((x_k^\top y_k)^2 - x_k^\top x_k y_k^\top y_k)$$

will never converge (because $s_k \leq -1$ and the second term is bounded). $\qquad\square$

**Lemma H.5.** *When* $x_k^\top y_k > \mu$ *or* $x_k^\top y_k < -\frac{h\mu u_k^2}{4 - hu_k^2}$, *we have* $u_{k+1}^2 - u_k^2 < 0$.

*Proof.*

$$u_{k+1}^2 - u_k^2 = h(\mu - x_k^\top y_k)((4 - hu_k^2)x_k^\top y_k + hu_k^2\mu) + h(4 - hu_k^2)((x_k^\top y_k)^2 - x_k^\top x_k y_k^\top y_k)$$
$$\leq h(\mu - x_k^\top y_k)((4 - hu_k^2)x_k^\top y_k + hu_k^2\mu).$$

Hence the proof follows 1D case. $\qquad\square$

**Lemma H.6.** *When* $-\frac{h\mu u_k^2}{4 - hu_k^2} \leq x_k^\top y_k < \mu$, *we have* $u_{k+2}^2 - u_k^2 < 0$.

*Proof.* Let $s_k = 1 - hu_k^2 - h^2 x_k^\top y_k(\mu - x_k^\top y_k)$, the same as 1D.

$$x_{k+1}^\top y_{k+1} - \mu = (x_k^\top y_k - \mu)(1 - hu_k^2 - h^2 x_k^\top y_k(\mu - x_k^\top y_k)) + h^2(2\mu - x_k^\top y_k)((x_k^\top y_k)^2 - x_k^\top x_k y_k^\top y_k)$$
$$\overset{\Delta}{=} (x_k^\top y_k - \mu)s_k + M.$$

$$u_{k+2}^2 - u_k^2 = u_{k+1}^2 - u_k^2 + u_{k+2}^2 - u_{k+1}^2$$
$$= h(\mu - x_k^\top y_k)((4 - hu_k^2)x_k^\top y_k + hu_k^2\mu) + h(4 - hu_k^2)((x_k^\top y_k)^2 - x_k^\top x_k y_k^\top y_k)$$
$$\quad + h(\mu - x_{k+1}^\top y_{k+1})((4 - hu_{k+1}^2)x_{k+1}^\top y_{k+1} + hu_{k+1}^2\mu)$$
$$\quad + h(4 - hu_{k+1}^2)((x_{k+1}^\top y_{k+1})^2 - x_{k+1}^\top x_{k+1} y_{k+1}^\top y_{k+1})$$
$$= \left[ h(\mu - x_k^\top y_k)(((4 - hu_k^2)x_k^\top y_k + hu_k^2\mu) + s_k(hu_{k+1}^2\mu + (4 - hu_{k+1}^2)(s_k(x_k^\top y_k - \mu) + \mu))) \right]$$
$$\quad + \left[ h(4 - hu_k^2)((x_k^\top y_k)^2 - x_k^\top x_k y_k^\top y_k) + hM^2(-4 + hu_{k+1}^2) \right.$$
$$\quad \left. + hM(-hu_{k+1}^2\mu + (-4 + hu_{k+1}^2)\mu + 2(4 - hu_{k+1}^2)(\mu - x_k^\top y_k)s_k) \right]$$
$$\overset{\Delta}{=} \text{I} + \text{II}.$$

I is the same as 1D $u_{k+2}^2 - u_k^2$ and hence $(1) < 0$. For II,

$$\text{II} = hM^2(-4 + hu_{k+1}^2) + h((x_k^\top y_k)^2 - x_k^\top x_k y_k^\top y_k)$$
$$\times \underbrace{(4 - hu^2 + h^2(2\mu - x_k^\top y_k)(-hu_{k+1}^2\mu + (-4 + hu_{k+1}^2)\mu + 2(4 - hu_{k+1}^2)(\mu - x_k^\top y_k)s_k))}_{\text{III}}.$$

From 1D results, $s_k < 0$. Then, III achieves its lower bound at $x_k^\top y_k = \mu$ or $x_k^\top y_k = -\frac{h\mu u_k^2}{4 - hu_k^2}$. For $x_k^\top y_k = \mu$,

$$\text{III} = 4 - hu_k^2 - 4h^2\mu^2 \geq 4h\mu(c - h\mu) > 0,$$

where the inequality is from $u_k^2 \leq u_0^2 = \frac{4}{h} - 4c\mu$ and $c > h\mu$.

For $x_k^\top y_k = -\frac{h\mu u_k^2}{4 - hu_k^2}$,

$$
\begin{aligned}
\text{III} &= 4 - hu_k^2 - \frac{4h^2(8 - hu_k^2)(4 - hu_k^2 - 2s_k(4 - hu_{k+1}^2))\mu^2}{(4 - hu_k^2)^2} \\
&\geq 4 - hu_k^2 - \frac{4h^2(8 - hu_k^2)(4 - hu_k^2 - 2s_k(4 - hu_k^2))\mu^2}{(4 - hu_k^2)^2} \\
&= 4 - hu_k^2 - \frac{4h^2(8 - hu_k^2)(1 - 2s_k)\mu^2}{4 - hu_k^2} \\
&\overset{u_k^2 \leq u_0^2}{\geq} 4 - hu_k^2 - \frac{4h^2(4 + 4ch\mu)(1 - 2s_k)\mu^2}{4ch\mu} \\
&\overset{u_k^2 \leq u_0^2}{\geq} \frac{4h\mu}{c}(-1 + c^2 + 2s_k + ch(-1 + 2s_k)\mu) \\
&\overset{*}{\geq} c^2(1 + 8h^2\mu^2) + c(h\mu - \frac{h^3\mu^3}{2}) - 7 - \frac{h^2\mu^2}{2} \\
&\overset{**}{\geq} 0,
\end{aligned}
$$

where $*$ follows from

$$
\begin{aligned}
s_k &= 1 - hu_k^2 - h^2 x_k^\top y_k(\mu - x_k^\top y_k) \\
&\overset{x_k^\top y_k = \mu/2}{\geq} 1 - hu_k^2 - \frac{h^2\mu^2}{4} \\
&\overset{u_k^2 \leq u_0^2}{\geq} -3 + 4ch\mu - \frac{h^2\mu^2}{4},
\end{aligned}
$$

and $**$ follows from $c \geq \sqrt{7}$.

From 1D discussion, $-4 + hu_{k+1}^2 < 0$. Also, $(x_k^\top y_k)^2 - x_k^\top x_k y_k^\top y_k < 0$. Hence we have $(3) \geq 0$.

Overall, $(2) \leq 0$ and therefore the whole discussion of $(1)$ is the same as 1D. $\qquad\square$

## H.2  PHASE 2

The proof in phase 2 is partly different from the scalar case due to the alignment. Before presenting the main convergence lemma, we first state the boundedness of GD in the following lemma.

**Lemma H.7.** *Once GD enters $\{\|x\|^2 + \|y\|^2 \leq \frac{2}{h}\}$, it will stay bounded inside $\{\|x\|^2 + \|y\|^2 \leq \frac{2}{h} + 2h\mu^2(1 + \frac{1}{1 - h^2\mu^2})\}$ and re-enter $\{\|x\|^2 + \|y\|^2 \leq \frac{2}{h}\}$ within finite steps where the number of such steps do not depend on the number of iteration.*

*Proof.* Assume $u_k^2 \leq \frac{2}{h}$. Then

$$
\begin{aligned}
u_{k+1}^2 - u_k^2 &= h(\mu - x_k^\top y_k)((4 - hu_k^2)x_k^\top y_k + hu_k^2\mu) + h(4 - hu_k^2)((x_k^\top y_k)^2 - x_k^\top x_k y_k^\top y_k) \\
&\leq 2h(\mu^2 - x_k^\top x_k y_k^\top y_k) \leq 2h\mu^2,
\end{aligned}
$$

where the first inequality follows from $u_k^2 \leq \frac{2}{h}$. If further $u_{k+1}^2 \leq \frac{2}{h} + 2h\mu^2$, then by $x_{k+1}^\top y_{k+1} \geq -\sqrt{x_{k+1}^\top x_{k+1} y_{k+1}^\top y_{k+1}}$ and $-1 + h^2\mu^2 < 0$, we have

$$
\begin{aligned}
u_{k+2}^2 - u_{k+1}^2 &= h(\mu - x_{k+1}^\top y_{k+1})((4 - hu_{k+1}^2)x_{k+1}^\top y_{k+1} + hu_{k+1}^2\mu) \\
&\quad + h(4 - hu_{k+1}^2)((x_{k+1}^\top y_{k+1})^2 - x_{k+1}^\top x_{k+1} y_{k+1}^\top y_{k+1}) \\
&\leq 2h(x_{k+1}^\top x_{k+1} y_{k+1}^\top y_{k+1}(-1 + h^2\mu^2) + \mu^2(1 - 2x_{k+1}^\top y_{k+1}h^2\mu + h^2\mu^2)) \\
&\leq 2h(x_{k+1}^\top x_{k+1} y_{k+1}^\top y_{k+1}(-1 + h^2\mu^2) + \mu^2(1 + 2\sqrt{x_{k+1}^\top x_{k+1} y_{k+1}^\top y_{k+1}}h^2\mu + h^2\mu^2)) \\
&\leq \frac{2h\mu^2}{1 - h^2\mu^2}.
\end{aligned}
$$

The first inequality is from $u_{k+1}^2 \leq \frac{2}{h} + 2h\mu^2$; the second inequality follows from the two conditions mentioned above; the third inequality follows from choosing $\sqrt{x_{k+1}^\top x_{k+1} y_{k+1}^\top y_{k+1}} = \frac{h^2\mu^3}{1-h^2\mu^2}$.

From previous proof, we know when $u_k^2 > \frac{2}{h}$, by Lemma H.3, we have GD will enter $\{\|x\|^2 + \|y\|^2 \leq \frac{2}{h}\}$ in finite steps with the decrease of $u_k^2$ in either one step or two steps. Then, $u_i^2 \leq \frac{2}{h} + 2h\mu^2(1 + \frac{1}{1-h^2\mu^2})$ for all $i \geq k$. □

Let $U_k = x_k^\top y_k$, $V_k = x_k^\top x_k$, $W_k = y_k^\top y_k$. Then we can rewrite the iteration for the three variables

$$
\begin{cases}
U_{k+1} = h(1 - hV_k)\mu W_k + h\mu V_k(1 - hW_k) + U_k(1 + h^2 V_k W_k - h(V_k + W_k) + h^2\mu^2) \\
V_{k+1} = 2h\mu U_k + V_k - 2hV_k W_k + h^2(\mu^2 W_k - 2\mu U_k W_k + V_k W_k^2) \\
W_{k+1} = 2h\mu U_k + W_k - 2hV_k W_k + h^2(\mu^2 V_k - 2\mu U_k V_k + V_k^2 W_k)
\end{cases}
\tag{10}
$$

When $h = \frac{4}{u_0^2 + 4c\mu}$ or $h = \frac{1}{2\sqrt{7}\mu}$, we have $h\mu \leq \frac{1}{2\sqrt{7}}$. The following is the main lemma in phase 2.

**Lemma H.8.** *Given $h\mu \leq \frac{1}{2\sqrt{7}}$, if GD enters $\{\|x\|^2 + \|y\|^2 \leq \frac{2}{h}\}$, it converges to $x^\top y = \|x\| \|y\| = \mu$, i.e., the global minimum, inside this region except for a measure-0 set of initial conditions.*

*Proof.* This proof follows the same method as the scalar decomposition. We need to prove that all the conclusions in scalar case also holds in this rank one decomposition.

Consider $u_{k+1}^2 - u_k^2$ when $x_k^\top y_k < -\mu$ or $x_k^\top y_k > \mu$,

$$
\begin{aligned}
u_{k+1}^2 - u_k^2 &= h(\mu - x_k^\top y_k)((4 - hu_k^2)x_k^\top y_k + hu_k^2\mu) + h(4 - hu_k^2)((x_k^\top y_k)^2 - x_k^\top x_k y_k^\top y_k) \\
&= h(\mu - x_k^\top y_k)((4 - hu_k^2)x_k^\top y_k + hu_k^2\mu) + h(4 - hu_k^2)(U_k^2 - V_k W_k) \\
&\leq h(\mu - x_k^\top y_k)((4 - hu_k^2)x_k^\top y_k + hu_k^2\mu),
\end{aligned}
$$

where the last expression is the same as scalar case and therefore under the conditions such that $u_{k+1}^2 - u_k^2 < 0$ in scalar case, we also have $u_{k+1}^2 - u_k^2 < 0$ in rank-one approximation.

As in scalar case, we also denote $s_k = 1 - hu_k^2 - h^2 x_k^\top y_k(\mu - x_k^\top y_k)$. First consider $\frac{1}{h} + 2h\mu^2 < u_k^2 \leq \frac{2}{h}$. If $-\mu \leq x_k^\top y_k < \mu$, $u_{k+1}^2 \geq u_k^2$, and $s_k \leq -1$, consider $u_{k+2}^2 - u_k^2$. From Lemma H.6, the extra terms of $u_{k+2}^2 - u_k^2$ compared to scalar case is

$$
\begin{aligned}
&hM^2(-4 + hu_{k+1}^2) + h((x_k^\top y_k)^2 - x_k^\top x_k y_k^\top y_k) \\
&\quad \times \underbrace{(4 - hu^2 + h^2(2\mu - x_k^\top y_k)(-hu_{k+1}^2\mu + (-4 + hu_{k+1}^2)\mu + 2(4 - hu_{k+1}^2)(\mu - x_k^\top y_k)s_k))}_{\text{III}}
\end{aligned}
$$

where $M = h^2(2\mu - x_k^\top y_k)((x_k^\top y_k)^2 - x_k^\top x_k y_k^\top y_k)$. Then we will prove that III is positive and then all the extra terms will be less than 0. By rewriting III in terms of $x_k^\top y_k$, we get

$$
\begin{aligned}
\text{III} &= 2(x_k^\top y_k)^2 h^2 s_k(4 - hu_{k+1}^2) + x_k^\top y_k(h^3 u_{k+1}^2\mu - 6h^2 s_k(4 - hu_{k+1}^2)\mu - h^2(-4 + hu_{k+1}^2)\mu) \\
&\quad - 2h^3 u_{k+1}^2\mu^2 + 4h^2 s_k(4 - hu_{k+1}^2)\mu^2 + 2h^2(-4 + hu_{k+1}^2)\mu^2 + 4 - hu_k^2 \\
&\geq 4 - hu_k^2 + 12h^2(-1 + 4s_k)\mu^2 - 12h^3 s_k u_{k+1}^2\mu^2 \\
&\geq 4 - hu_k^2 + 12h^2(-1 + 4s_k)\mu^2 - 12h^3 s_k u_k^2\mu^2 \\
&= 4 - hu_k^2 + 12h^2(-1 + s_k(4 - hu_k^2))\mu^2 > 0,
\end{aligned}
$$

where the first inequality is because it obtains the minimum at $x_k^\top y_k = -\mu$; the second inequality follows from $u_{k+1}^2 \geq u_k^2$; the last inequality is from the range of $u_k^2$ and $s_k$. Therefore the extra terms of $u_{k+2}^2 - u_k^2$ is always negative and then we have the same conclusion as scalar case, i.e., $u_{k+2}^2 - u_k^2 < -C(\mu - x_k^\top y_k)^2$ for some constant $C > 0$.

Next, consider $s_{k+1}$ when $-\mu \le x_k^\top y_k \le 3\mu$ and $\frac{1}{h} + 2h\mu^2 < u_k^2 \le \frac{2}{h}$ which implies $-1 - \frac{h^2\mu^2}{4} \le s_k < 0$. The extra terms of $s_{k+1}$ compared to scalar case is

$$h^2(U_k^2 - V_k W_k)\underbrace{\left(-(4 - hu_k^2) + (-x_k^\top y_k + 2\mu)(2h^2 s_k(x_k^\top y_k - \mu) + h^2\mu) + h^4 x(-x_k^\top y_k + 2\mu)^2\right)}_{\text{IV}},$$

where $U_k^2 - V_k W_k \le 0$. Then consider the upper bound of IV

$$\text{IV} = -4 - 2(x_k^\top y_k)^2 h^2 s_k + x_k^\top y_k(-h^2\mu + 6h^2 s\mu) + hu_k^2 + 2h^2\mu^2 - 4h^2 s_k\mu^2$$
$$\le -4 + hu_k^2 + (-h^2 - 4h^2 s_k)\mu^2 < 0,$$

where the first inequality follows from $x_k^\top y_k \le 3\mu$. Therefore the extra terms is non-negative, namely all the lower bound for $s_{k+1}$ in the scalar case also holds for rank-one approximation.

Also we need to consider $s_{k+2}$ when $s_k > -1$, $-\mu \le x_k^\top y_k \le 3\mu$, and $\frac{1}{h} + 2h\mu^2 < u_k^2 \le \frac{2}{h}$. For scalar case, from Lemma G.13 with $\epsilon = x_{k+1}^\top y_{k+1} - \mu$, we have

$$s_{k+2} = s_{k+1} + (h^2(3 + s_{k+1}^2) - h^3 u_k^2)\epsilon^2 + h^2\epsilon(-3\mu - s_{k+1}\mu)$$
$$= 3h^2\epsilon^2 + h^2 s_{k+1}^2 \epsilon^2 - h^3 u_k^2 \epsilon^2 - 3h^2 \epsilon\mu + s_{k+1}(1 - h^2\epsilon\mu).$$

Since the minimum point w.r.t. $s_{k+1}$ is $\frac{-(1 - h^2\epsilon\mu)}{2h^2\epsilon^2} < -1 - \frac{h^2\mu^2}{4}$, we have if $s_{k+1}$ is larger, then $s_{k+2}$ is also larger. Given the same value for $s_k$ for both scalar and rank-one cases, $s_{k+1}$ is larger from the above discussion and thus $s_{k+2}$ is also larger, namely, the lower bound for $s_{k+2}$ also holds in rank-one cases.

Next consider $u_k^2 \le \frac{1}{h} + 2h\mu^2$ where $|s_k| < 1$ and we need the iteration to have certain restriction such that it will be bounded in a region. If $s_k < 0$, then it follows the same from the above discussion. If $s_k > 0$ and $x_k^\top y_k > \mu$ or $x_k^\top y_k < -\mu$, then by $u_{k+1}^2 - u_k^2 < 0$, it will stay in $\{x_k^\top y_k > \mu, u_k^2 \le \frac{1}{h} + 2h\mu^2\}$ until convergence. If $s_k > 0$ and $-\mu \le x_k^\top y_k < \mu$, similar to scalar case, we consider $|x_k^\top x_k - y_k^\top y_k| = |V_k - W_k|$ and

$$|V_{k+1} - W_{k+1}| = |V_k - W_k| \cdot |1 - h^2 V_k W_k + 2h^2 U_k\mu - h^2\mu^2|,$$

where $U_k = x_k^\top y_k$ and $U_k^2 \le V_k W_k$. Then

$$1 - h^2 V_k W_k + 2h^2 U_k\mu - h^2\mu^2 \le 1 - h^2 U_k^2 + 2h^2 U_k\mu - h^2\mu^2 < 1,$$

$$1 - h^2 V_k W_k + 2h^2 U_k\mu - h^2\mu^2 \ge 1 - h^2\frac{u_k^4}{4} - 2h^2\mu - h^2\mu^2 \ge 1 - \frac{1}{4}(1 + 2h^2\mu^2)^2 - 3h^2\mu^2 > 0.$$

Therefore if $-\mu \le x_k^\top y_k < \mu$ and $u_k^2 \le \frac{1}{h} + 2h\mu^2$, we have $|V_{k+1} - W_{k+1}| < |V_k - W_k|$, meaning together with $0 \le \mu - x_{k+1}^\top y_{k+1} < \mu - x_k^\top y_k$, $x_{k+1}$ and $y_{k+1}$ is bounded in the monotone convergence region.

So far we retrieve all the conditions for convergence. Therefore, from the proof of the convergence in scalar case, we have that there exists a constant $K > 0$, s.t., $|s_k| < 1$ and $0 < 2\mu - x_k^\top y_k < 2\mu$ for all $k > K$. Then let $S_k = h^2(2\mu - x_k^\top y_k)$ and we have

$$x_{k+1}^\top y_{k+1} - \mu = (x_k^\top y_k - \mu)(1 - hu_k^2 - h^2 x_k^\top y_k(\mu - x_k^\top y_k))$$
$$+ h^2(2\mu - x_k^\top y_k)((x_k^\top y_k)^2 - x_k^\top x_k y_k^\top y_k)$$
$$= (x_k^\top y_k - \mu)s_k + S_k(U_k^2 - V_k W_k)$$

In fact $x_k^\top y_k - \mu \to 0$ as $k \to 0$. First, since $S_k(U_k^2 - V_k W_k) \to 0$ as $k \to 0$ by Lemma H.9 and $|s_k| < 1$, we have $x_k^\top y_k - \mu$ is bounded. Also, from previous discussion, $|s_k| \le \max\{C_2, 1 - C_3(x_k^\top y_k - \mu)^2\}$ for some constant $C_2, C_3 > 0$. For any $\epsilon_0 > 0$, there exists $K_1 > 0$ and $\epsilon_1 \ll \epsilon_0$ such that $|S_k(U_k^2 - V_k W_k)| < \epsilon_1$ for all $k > K_1$ and $C_2 \le 1 - C_3\epsilon_1^2$. Therefore $x_k^\top y_k - \mu$ will

decreases to $\mathcal{O}(\epsilon_1)$. This is because

$$
\begin{aligned}
|x_{k+1}^\top y_{k+1} - \mu| &\leq (1 - C_3\epsilon_1^2)|x_k^\top y_k - \mu| + 2\mu|U_{K_1}^2 - V_{K_1}W_{K_1}|C_0^{2(k-K_1)} \\
&\leq |(x_{K_1}^\top y_{K_1} - \mu)|(1 - C_3\epsilon_1^2)^{k-K_1} \\
&\quad + 2h^2\mu(V_{K_1}W_{K_1} - U_{K_1}^2)\sum_{i=K_1}^{k} C_0^{2(i-K_1)}(1 - C_3\epsilon_1^2)^{k-i} \\
&\leq |(x_{K_1}^\top y_{K_1} - \mu)|(1 - C_3\epsilon_1^2)^{k-K_1} + 2h^2\mu(V_{K_1}W_{K_1} - U_{K_1}^2)(k - K_1)C_4^{k-K_1}
\end{aligned}
$$

where $C_4 = \max\{C_0, 1 - C_3\epsilon_1^2\} < 1$ and we have $kC_4^k \to 0$ as $k \to \infty$. Therefore there exists $K_2 \geq K_1$ s.t. $x_{K_2}^\top y_{K_2} - \mu = \mathcal{O}(\epsilon_1)$. Then we will show for all $k > K_2$, $|x_k^\top y_k - \mu| < \epsilon_0$. As is shown before there exists $K_3 > K_2$ and $\epsilon_2 \ll \epsilon_1$ s.t. $|S_k(U_k^2 - V_kW_k)| < \epsilon_2$. Since $|s_k| < 1$, the increase of $|x_k^\top y_k - \mu|$ is also $\mathcal{O}(\epsilon_1)$ and thus $|x_k^\top y_k - \mu| = \mathcal{O}(\epsilon_1) < \epsilon_0$ for $k > K_2$. Then we have the convergence of $|x_k^\top y_k - \mu| \to 0$.

In general, we have $x_k^\top y_k \to \mu$ and $|\cos(\angle(x_k, y_k))| \to 0$, which implies GD converges to the global minimum $x^\top y = \|x\|\,\|y\| = \mu$. $\qquad\square$

For the proof of alignment, we only need to consider when $\|x_k\|^2 + \|y_k\|^2 \leq \frac{2}{h} + 2h\mu^2(1 + \frac{1}{1-h^2\mu^2})$. Also, we have $V_kW_k - U_k^2 \geq 0$ and

$$
V_{k+1}W_{k+1} - U_{k+1}^2 = (V_kW_k - U_k^2)(1 - h(V_k + W_k) + h^2(V_kW_k - \mu^2))^2.
$$

Thus we denote $l_k = 1 - h(V_k + W_k) + h^2(V_kW_k - \mu^2)$ which characterizes the change of $V_kW_k - U_k^2$.

**Lemma H.9.** *If $\|x_k\|^2 + \|y_k\|^2 \leq \frac{2}{h} + 2h\mu^2(1 + \frac{1}{1-h^2\mu^2})$, then $V_kW_k - U_k^2$ converges to 0 and also $|\cos(\angle(x_k, y_k))|$ converges to 1.*

*Proof.* By $0 \leq V_kW_k \leq (\frac{V_k+W_k}{2})^2$ and the boundedness theorem, we have

$$
V_k + W_k \leq \frac{2}{h} + 2h\mu^2(1 + \frac{1}{1 - h^2\mu^2}),
$$

and

$$
l_k = 1 - h(V_k + W_k) + h^2(V_kW_k - \mu^2) \geq -1 - h^2\mu^2(3 + \frac{2}{1 - h^2\mu^2}), \tag{11}
$$

$$
\begin{aligned}
l_k = 1 - h(V_k + W_k) + h^2(V_kW_k - \mu^2) &\leq \frac{h^2}{4}(V_k + W_k)^2 - h(V_k + W_k) + 1 - h^2\mu^2 \\
&\leq 1 - h^2\mu^2 < 1.
\end{aligned}
$$

We will show that there exists $K > 0$ and a constant $C > 0$ only depending on $h\mu$, s.t. when $k > K$, $|l_k| < 1 - C$.

If $V_k + W_k < \frac{2}{h} - h\mu^2$, then

$$
l_k = 1 - h(V_k + W_k) + h^2(V_kW_k - \mu^2) > -1 + h^2 V_kW_k > -1.
$$

(If $V_kW_k = 0$, then $V_kW_k - U_k^2 = 0$ meaning it just converge.)

Therefore when $V_k + W_k < \frac{2}{h} - h\mu^2$, we have $|l_k| < 1$. We only need to consider $l_{k+1}$ when $\frac{2}{h} - h\mu^2 \leq V_k + W_k \leq \frac{2}{h} + 2h\mu^2(1 + \frac{1}{1-h^2\mu^2})$. Next we replace $l_k, V_k, W_k, U_k$ by $l, V, W, U$ for

simplicity.

$$
\begin{aligned}
l_{k+1} &= 1 - h(V_{k+1} + W_{k+1}) + h^2(V_{k+1}W_{k+1} - \mu^2) \\
&= 1 - h(V_{k+1} + W_{k+1}) + h^2(V_{k+1}W_{k+1} - U_{k+1}^2) + h^2(U_{k+1}^2 - \mu^2) \\
&= 1 - h(V_{k+1} + W_{k+1}) + h^2 l^2 (VW - U^2) + h^2(U_{k+1}^2 - \mu^2) \\
&\stackrel{(10)}{=} l + h^2(3 + l^2 - h(1 + 4h^2\mu^2)(V + W))(VW - U^2) \\
&\quad + h^2 U^2 (3 + (l + 2h^2\mu^2)^2 - h(1 + 4h^2\mu^2)(V + W)) \\
&\quad + h^2\mu^2(h^2(W + V)^2 - h(W + V) + 4h^4 V^2 W^2) \\
&\quad + 2h^2 U\mu((1 - (l + 2h^2\mu^2))h(W + V) - 2 + 2(l + 2h^2\mu^2)(1 - l - h^2\mu^2)) \quad (12)
\end{aligned}
$$

If $l \geq -4h^2\mu^2$, we have

$$
\begin{aligned}
l_{k+1} &= l + h^2(3 + l^2 - h(1 + 4h^2\mu^2)(V + W))(VW - U^2) \\
&\quad + h^2 U^2 (3 + (l + 2h^2\mu^2)^2 - h(1 + 4h^2\mu^2)(V + W)) \\
&\quad + h^2\mu^2(h^2(W + V)^2 - h(W + V) + 4h^4 V^2 W^2) \\
&\quad + 2h^2 U\mu((1 - (l + 2h^2\mu^2))h(W + V) - 2 + 2(l + 2h^2\mu^2)(1 - l - h^2\mu^2)) \\
&\geq l + 2h^2 U\mu((1 - (l + 2h^2\mu^2))h(W + V) - 2 + 2(l + 2h^2\mu^2)(1 - l - h^2\mu^2)) \\
&\geq l - h^2\mu(V + W)(5h^2\mu^2 + (9h^4\mu^4)/8) \\
&\geq -4h^2\mu^2 - h\mu(2 + 2h^2\mu^2(1 + \frac{1}{1 - h^2\mu^2}))(5h^2\mu^2 + (9h^4\mu^4)/8) \\
&> -1,
\end{aligned}
$$

where the first inequality is because all the removed terms are positive when $\frac{2}{h} - h\mu^2 \leq V + W \leq \frac{2}{h} + 2h\mu^2(1 + \frac{1}{1-h^2\mu^2})$; the second inequality follows from recollecting $l$ and take $l$ from quadratic formula with the $h(V + W)$ taken at the bound, and $U > -\frac{V+W}{2}$; the third inequality follows from the bounds of $h(V + W)$ and $l$; the last inequality is from $h\mu \leq \frac{1}{2\sqrt{7}}$.

If $l < -4h^2\mu^2 \Rightarrow l + h^2\mu^2 < 0$. By rewriting $l_{k+1}$ in the following way

$$
\begin{aligned}
l_{k+1} &= l + h^2(4h^4 V^2 W^2 - h(V + W) + h^2(V + W)^2)\mu^2 + 4h^4 U^2\mu^2(l + h^2\mu^2) \\
&\quad + 2h^2 U\mu(-2 - h(V + W)(-1 + l + 2h^2\mu^2) - 2(-1 + l + h^2\mu^2)(l + 2h^2\mu^2)) \\
&\quad + h^2 VW(3 + l^2 - h(V + W)(1 + 4h^2\mu^2)),
\end{aligned}
$$

we have that the minimum of $l_{k+1}$ is achieved when $U = \pm\sqrt{VW}$. Thus we will consider two cases: $U \geq 0$ and $U < 0$. Note we always have $U^2 < VW$ (otherwise $VW - U^2 = 0$ just converges) and hence $l_{k+1}$ is larger than the corresponding value at $U = \pm\sqrt{VW}$. Let $q = h|U|, v = h\mu, q = cv$ for $c \geq 0$. Then we have

$$
q^2 = c^2 v^2 \leq h^2 VW = h(V + W) - 1 + l - v^2 < 1, \quad i.e., \quad c < \frac{1}{v}. \quad (13)
$$

Next, let

$$
l'_{k+1} = l_{k+1} - h^2(3 + l^2 - h(1 + 4h^2\mu^2)(V + W))(VW - U^2) < l_{k+1} - h^2(\frac{1}{2} + l^2)(VW - U^2).
$$

**First** let $U \geq 0$ and $q = hU < h\sqrt{VW}$. Choose $q = hU = h\sqrt{VW}$. Then by

$$
h(V + W) = 1 - l + h^2(VW - \mu^2) = 1 - l + c^2 v^2 - v^2,
$$

we have the minimum value of $l_{k+1}$ when $U \geq 0$

$$
\begin{aligned}
l_{k+1} \geq l'_{k+1} &= l + (-1 + c)(-((-1 + l)l) + c(2 + l + l^2))v^2 \quad (14) \\
&\quad - (-1 + c)^2(1 + c^2 - 2l + 2cl)v^4 + (-1 + c)^4 v^6.
\end{aligned}
$$

1) When $-1 < l < -4h^2\mu^2$ and $c = 1$, we have $l_{k+1} = l'_{k+1} = l = l_k$.

2) When $-1 < l < -4h^2\mu^2$ and $c > 1$, the derivative of (14) with respect to $c$ is

$$\frac{d}{dc}l'_{k+1} = (1-l)lv^2 + (-2-l-l^2)v^2 + 2v^4 - 6lv^4 - 4v^6 + 3c^2(2v^4 - 2lv^4 - 4v^6)$$
$$+ 4c^3(-v^4 + v^6) + 2c((2+l+l^2)v^2 - 2v^4 + 6lv^4 + 6v^6),$$

which is a third order polynominal with negative coefficient of the highest degree term and has a root in $[0, 1]$. Also, when $c = 1$, $\frac{d}{dc}l'_{k+1} = 2v^2(1+l) > 0$. Therefore, when $1 < c < \frac{1}{v}$, $l'_{k+1}$ either increases or first increases then decreases, namely, the minimum of $l'_{k+1}$ is achieved at $c = 1$ or $c = \frac{1}{v}$. If $c = 1$, by the previous discussion, $l'_{k+1} = l$. If $c = \frac{1}{v}$,

$$l'_{k+1} = 1 + l^2(-1+v)^2 - v^2 - 2v^3 + 5v^4 - 4v^5 + v^6 + l(2 - 2v + 5v^2 - 6v^3 + 2v^4)$$
$$\geq \frac{v^2(-24 + 44v - 25v^2 + 4v^3)}{4(-1+v)^2} > -1,$$

where the first inequality follows from the property of quadratic function by taking $l = -\frac{(2-2v+5v^2-6v^3+2v^4)}{2(-1+v)^2}$. Therefore when $-1 < l < -4h^2\mu^2$ and $c > 1$, $l_{k+1} > -1$.

3) When $-1 < l < -4h^2\mu^2$ and $c < 1$, rewrite $l'_{k+1}$ with respect to $l$ and we have

$$l'_{k+1} = 2(-1+c)cv^2 - (-1+c)^2v^4 - (-1+c)^2c^2v^4 + (-1+c)^4v^6 + l^2((1-c)v^2$$
$$+ (-1+c)cv^2) + l(1 + (-1+c)v^2 + (-1+c)cv^2 + 2(-1+c)^2v^4 - 2(-1+c)^2cv^4),$$

where the minimum point is

$$l = -\frac{1}{2(-1+c)^2v^2} - \frac{1}{2} - (1-c)v^2 < -\frac{1}{2v^2} - \frac{1}{2} < -1.$$

Therefore the minimum of $l'_{k+1}$ is obtained at $l = -1$, i.e.,

$$l'_{k+1} \geq -1 + (-1+c)^2v^2(2 - (3 - 2c + c^2)v^2 + (-1+c)^2v^4)$$
$$\geq -1 + (-1+c)^2v^2(2 - 3v^2 + (-1+c)^2v^4) > -1,$$

where the inequality is from $0 \leq c < 1$. Therefore when $-1 < l < -4h^2\mu^2$ and $c < 1$, $l_{k+1} > -1$.

4) When $l \leq -1$ and $c \geq \frac{5}{4}$, consider $l'_{k+1} - l$ (15) and take the derivative $\frac{d}{dc}(l'_{k+1} - l) = \frac{d}{dc}l'_{k+1}$. From previous discussion, consider $\frac{d}{dc}l'_{k+1}$ at $c = \frac{5}{4}$, i.e.,

$$\frac{d}{dc}l'_{k+1} = 1/16v^2(48 + 8l^2 - 23v^2 + v^4 + l(40 - 6v^2)) > v^2 - (161v^4)/16 + (325v^6)/16 > 0,$$

where the first inequality follows from $l > -1 - 6v^2$ by (11). Therefore the minimum of $l'_{k+1} - l$ is at $c = \frac{5}{4}$ or $c = \frac{1}{v}$. If $c = \frac{5}{4}$,

$$l'_{k+1} - l = \frac{v^2}{256}(160 + 16l^2 - 41v^2 + v^4 - 8l(-18 + v^2)) > \frac{v^2}{256}(32 - 705v^2 + 625v^4) > 0$$

where the first inequality is from $l > -1 - 6v^2$ by (11). If $c = \frac{1}{v}$,

$$l'_{k+1} - l = l^2(-1+v)^2 + l(-1+v)(-1+v-4v^2+2v^3) + (-1+v)(-1-v+2v^3-3v^4+v^5)$$
$$\geq (1-v)^3(1+3v+v^2-v^3) > 0$$

where the first inequality is from $l \leq -1$. Therefore when $l \leq -1$ and $c \geq \frac{5}{4}$, $l_{k+1} - l_k > \min\{\frac{v^2}{256}(32 - 705v^2 + 625v^4), (1-v)^3(1+3v+v^2-v^3)\} > 0$.

In the next two cases $c \leq 1$ and $1 < c < \frac{5}{4}$, we no longer assume $q = h\sqrt{VW}$ but want the same lower bound of $l'_{k+1}$. Since $q = hU < h\sqrt{VW}$, we have $h(V+W) > 1 - l + c^2v^2 - v^2 > 0$. By rewriting $l'_{k+1}$ to be

$$l'_{k+1} = l - 4cv^2 + 3c^2v^2 + 4c^4v^6 + 4cv^2(1-l-v^2)(l+2v^2) + v^2h^2(V+W)^2$$
$$+ c^2v^2(l+2v^2)^2 + (-v^2 + 2cv^2(1-l-2v^2) - c^2v^2(1+4v^2))h(V+W),$$

it can be seen from the minimum point w.r.t. $h(V+W)$ that when $h(V+W) > 0$ and $0 \le c < \frac{5}{4}$, $l'_{k+1}$ decreases as $h(V+W)$ decreases. Therefore by $h(V+W) > 1 - l + c^2v^2 - v^2$, we have the same expression for the lower bound of $l'_{k+1}$ but with $U^2 < VW$, i.e.,

$$l_{k+1} - l'_{k+1} = h^2(3 + l^2 - h(1 + 4h^2\mu^2)(V+W))(VW - U^2) > h^2(\frac{1}{2} + l^2)(VW - U^2),$$

$$l'_{k+1} > l + (-1+c)(-((-1+l)l) + c(2 + l + l^2))v^2$$
$$- (-1+c)^2(1 + c^2 - 2l + 2cl)v^4 + (-1+c)^4v^6.$$

5) When $l \le -1$ and $c \le 1$, consider $l'_{k+1} - l$

$$l'_{k+1} - l > (-1+c)(-((-1+l)l) + c(2 + l + l^2))v^2 \tag{15}$$
$$- (-1+c)^2(1 + c^2 - 2l + 2cl)v^4 + (-1+c)^4v^6$$
$$= (-1+c)^2l^2v^2 + (-1+c)lv^2(1 + c - 2(1-c)v^2 + 2(1-c)cv^2)$$
$$+ (-1+c)v^2(2c + (1-c)v^2 + (1-c)c^2v^2 + (-1+c)^3v^4)$$

where the minimum point is $l = -\frac{1}{2} + \frac{1 - (1-c)v^2 + (1-c)cv^2}{1-c} > -1$. By $l \le -1$ we have

$$l'_{k+1} - l > v^2(2(-1+c)^2 - (-1+c)^2(3 - 2c + c^2)v^2 + (-1+c)^4v^4)$$
$$= v^2((-1+c)^2(2 - (3 - 2c + c^2)v^2) + (-1+c)^4v^4)$$
$$\ge v^2((-1+c)^2(2 - 3v^2) + (-1+c)^4v^4) \ge 0.$$

Therefore when $l \le -1$ and $c \le 1$, $l_{k+1} - l_k > h^2(\frac{1}{2} + l^2)(VW - U^2) \ge \frac{3}{2}h^2(VW - U^2)$.

6) When $l \le -1$ and $1 < c < \frac{5}{4}$, consider $U_{k+1}$. By $h(V+W) - h^2VW = 1 - l - v^2$, we have

$$U_{k+1} = h(1 - hV)\mu W + h\mu V(1 - hW) + U(1 + h^2VW - h(V+W) + h^2\mu^2)$$
$$= \mu((c-1)(l + v^2) + 1 + cv^2 - h^2VW).$$

Thus denote $hU_{k+1} = c_{k+1}h\mu = c_{k+1}v$ and we have

$$c_{k+1} = (c-1)(l + v^2) + 1 + cv^2 - h^2VW$$
$$< (c-1)(l + v^2) + 1 + cv^2 - c^2v^2 < 1.$$

Then if $l_{k+1} - l \ge 0$, then $l_{k+2} - l = l_{k+2} - l_{k+1} + l_{k+1} - l > l_{k+2} - l_{k+1} > \frac{3}{2}h^2(VW - U^2)$. Otherwise, we have $l_{k+1} < l$ and consider $l_{k+2}$. Also, the distance between 1 and $c_{k+1}$ is larger than that of 1 and $c$, i.e.,

$$1 - c_{k+1} - (c-1) > (1 - (c-1)(l + v^2) + 1 + cv^2 - c^2v^2) - (c-1) \tag{16}$$
$$= 1 + l + v^2 + c^2v^2 + c(-1 - l - 2v^2) > 0.$$

From previous discussion, $l'_{k+1} - l$ increases as $l$ decreases. Also $\frac{d}{dc}l'_{k+1} < 0$ when $0 \le c < 1$, so $l'_{k+1} - l$ increases as $c$ decreases. By (16), we have $c_{k+1} < 2 - c$. Then

$$l'_{k+2} - l = l'_{k+2} - l_{k+1} + l_{k+1} - l > l'_{k+2} - l_{k+1} + l'_{k+1} - l$$
$$> (-1+c)(-((-1+l)l) + c(2 + l + l^2))v^2$$
$$- (-1+c)^2(1 + c^2 - 2l + 2cl)v^4 + (-1+c)^4v^6$$
$$+ (1-c)v^2(-((-1+l)l) - (-2+c)(2 + l + l^2)$$
$$+ (-1+c)(5 + c^2 + 2l - 2c(2+l))v^2 - (-1+c)^3v^4)$$
$$= 2v^2(-1+c)^2((2 + l + l^2) + (-2 - (-1+c)^2)v^2 + (-1+c)^2v^4)$$
$$> 2v^2(-1+c)^2((2 - 3v^2) + (-1+c)^2v^4) > 0,$$

where the second inequality follows from $c_{k+1} < 2 - c$ and $l_{k+1} < l$; the last inequality is by $l \le -1$ and $1 < c < \frac{5}{4}$. Therefore when $l \le -1$ and $1 < c < \frac{5}{4}$, $l_{k+2} - l_k > \frac{3}{2}h^2(VW - U^2)$ or $2v^2(-1+c)^2((2 - 3v^2) + (-1+c)^2v^4)$.

**Second**, let $U < 0$ and $q = -hU < h\sqrt{VW}$. Choose $q = -hU = h\sqrt{VW}$. Then similarly we have the minimum value of $l_{k+1}$ when $U < 0$

$$
\begin{aligned}
l'_{k+1} &= l + (1+c)((-1+l)l + c(2+l+l^2))v^2 - (1+c)^2(1+c^2 \\
&\quad - 2l - 2cl)v^4 + (1+c)^4v^6 \\
&= (1+c)^2 l^2 v^2 + l(1 + (-1+c^2)v^2 + 2(1+c)^3 v^4) \\
&\quad + (1+c)v^2(v^2(-1+v^2) + c^3 v^2(-1+v^2) + c^2 v^2(-1+3v^2) + c(2 - v^2 + 3v^4)) \\
&\geq -1 + (-4 + 4c + 3c^2)v^2 + (15 + 16c - 5c^2 - 8c^3 - 2c^4)v^4 \\
&\quad + (-5 - 4c + 2c^2 + c^3)^2 v^6,
\end{aligned}
$$

(17)

(18)

where the inequality is because when $l > -1 - 6v^2 + c^2 v^2$ (from (11)), $l'_{k+1}$ increases as $l$ increases.

1) When $c \geq 1$, since $(-5 - 4c + 2c^2 + c^3)^2 v^6 \geq 0$, we consider $(-4 + 4c + 3c^2)v^2 + (15 + 16c - 5c^2 - 8c^3 - 2c^4)v^4$ denoted to be $l^{part}_{k+1}$

$$
\frac{d}{dc} l^{part}_{k+1} = (4 + 6c)v^2 + (16 - 10c - 24c^2 - 8c^3)v^4.
$$

It has a root between -2 and 0. Also when $c = 1$, $\frac{d}{dc} l^{part}_{k+1} > 0$; when $c = \frac{1}{v}$, $\frac{d}{dc} l^{part}_{k+1} < 0$. Thus, the minimum of $l^{part}_{k+1}$ is achieved at $c = 1$ or $c = \frac{1}{v}$, i.e., $l^{part}_{k+1} \geq v^2(3 + 16v^2)$. Then

$$
l'_{k+1} \geq -1 + v^2(3 + 16v^2) + (-5 - 4c + 2c^2 + c^3)^2 v^6 > 0.
$$

Therefore when $c \geq 1$, $l_{k+1} > -1$.

2) When $0 \leq c < 1$ and $l > -1$, from previous discussion of $l'_{k+1}$ we have

$$
\begin{aligned}
l'_{k+1} &\geq -1 + 2(1+c)^2 v^2 - (1+c)^2(3 + 2c + c^2)v^4 + (1+c)^4 v^6 \\
&\geq -1 + (1+c)^2 v^2(2 - 6v^2) + (1+c)^4 v^6 \\
&\geq -1 + v^2(2 - 6v^2) + v^6 > -1.
\end{aligned}
$$

Therefore when $0 \leq c < 1$ and $l > -1$, $l_{k+1} > -1$.

3) When $0 \leq c < 1$ and $l \leq -1$, consider $l'_{k+1} - l$. By (17),

$$
\begin{aligned}
l'_{k+1} - l &= (1+c)^2 l^2 v^2 + l((-1+c^2)v^2 + 2(1+c)^3 v^4) \\
&\quad + (1+c)v^2(v^2(-1+v^2) + c^3 v^2(-1+v^2) + c^2 v^2(-1+3v^2) + c(2 - v^2 + 3v^4)) \\
&\geq 2(1+c)^2 v^2 - (1+c)^2(3 + 2c + c^2)v^4 + (1+c)^4 v^6 \\
&\geq v^2(2 - 6v^2) + v^6,
\end{aligned}
$$

where the first inequality is because the minimum point of $l$ is greater than -1. Therefore when $0 \leq c < 1$ and $l \leq -1$, $l_{k+1} - l_k > v^2(2 - 6v^2) + v^6$.

Actually all the proofs of $l_k > -1 \Rightarrow l_{k+1} > -1 + C$ for some $C > 0$ are valid for all $0 < V_k + W_k \leq \frac{2}{h} + 2h\mu^2(1 + \frac{1}{1 - h^2\mu^2})$.

In general, if $l_k > -1$, then there exists a constant $C > 0$ only depending on $h\mu$ s.t. $l_i > -1 + C$ for all $i > k$. If $l_k \leq -1$, then either (1) $l_{k+1}$ increases at least by a fixed value only depending on $h\mu$, or (2) $l_{k+2}$ or $l_{k+1}$ increases by $\frac{3}{2}h^2(V_k W_k - U_k^2)$. If it keeps staying in case (2), then since $l_k \leq -1$, $V_k W_k - U_k^2$ will be larger and larger until $l_i > -1$ for some $i$. Therefore, there exists a $K > 0$, s.t. when $k > K$, $l_k > -1 + C$. Then because $l_k \leq 1 - h^2\mu^2$, let $0 < C_0 = \max\{1 - C, 1 - h^2\mu^2\} < 1$ and we have $|l_k| \leq C_0 < 1$ for all $k > K$. Further,

$$
V_k W_k - U_k^2 \leq C_0^{2(k-K)}(V_K W_K - U_K^2) \to 0 \quad \text{as } k \to \infty,
$$

i.e., $1 - \frac{U_k^2}{V_k W_k} \to 0 \Rightarrow |\cos(\angle(x_k, y_k))| \to 1$.

Otherwise, there exists a $K > 0$, s.t., $V_K W_K - U_K^2 = 0$, then $V_k W_k - U_k^2 = 0$ for all $k \geq K$. There are two cases. (i) $x_K$ and $y_K$ are already aligned. (ii) one of $V_K$ and $W_K$ is 0, WOLG assume

$V_K = 0$. Then from the GD update,

$$\begin{cases} x_{k+1} = x_k + h(\mu I - x_k y_k^\top) y_k \\ y_{k+1} = y_k + h(\mu I - y_k x_k^\top) x_k \end{cases},$$

$x_{K+1}$ and $y_{K+1}$ will be aligned for both cases. Then for all $k \geq K+1$, $x_k$ and $y_k$ is aligned, i.e., $|\cos(\angle(x_k, y_k))| = 1$. □

# I  PROOF OF THEOREM 5.1

## I.1  GENERAL RANK 1 APPROXIMATION FOR NON-NEGATIVE DIAGONAL MATRIX

We first discuss an easy case where $A$ is a non-negative diagonal matrix (later on we will see this is actually the canonical case) and consider

$$\min_{x,y \in \mathbb{R}^n} \|A - xy^\top\|_F^2 / 2. \tag{19}$$

Assume $A = \begin{pmatrix} \mu_1 I_{n_1} & & \\ & \ddots & \\ & & \mu_m I_{n_m} \end{pmatrix}$, where $I_{n_i}$ is an identity matrix of dimension $n_i \times n_i$, $\sum_{i=1}^m n_i = n$, $\mu_i \geq 0$, $\mu_i \neq \mu_j$ for $i \neq j$. Let $S_i = \{1, 2, \cdots, n_i\} + \sum_{j=1}^{i-1} n_j$, i.e., an index set describing the positional indices of $I_{n_i}$. Let $\mu_{max} = \max\{\mu_1, \cdots, \mu_m\}$ and $S_{max}$ be the corresponding index set. Let $[x_\infty, y_\infty]$ be a fixed point of the GD map. Let $x_{s,\infty}$ and $y_{s,\infty}$ be the $s$th element of $x_\infty$ and $y_\infty$.

The following theorem characterizes the fixed points of the GD map.

**Theorem I.1.** *The fixed points of GD map satisfy* $\sum_{s \in S_i} x_{s,\infty}^2 \cdot \sum_{s \in S_i} y_{s,\infty}^2 = \mu_i^2$, $x_{s,\infty} = y_{s,\infty} = 0$ *for* $s \notin S_i$, $\forall i = 1, \cdots, n$. *Thus* $\|x_\infty\| \|y_\infty\| = \mu_i$, $\forall i = 1, \cdots, n$.

*Proof.* To solve the fixed points, we have $\forall i = 1, \cdots, n$ and $x \in S_i$

$$\begin{cases} (A - x_\infty y_\infty^\top) y_\infty = A y_\infty - (y_\infty^\top y_\infty) x_\infty = 0 \\ (A - x_\infty y_\infty^\top)^\top x_\infty = A^\top x_\infty - (x_\infty^\top x_\infty) y_\infty = 0 \end{cases}$$

i.e.,

$$\begin{cases} \mu_i y_{s,\infty} = (y_\infty^\top y_\infty) x_{s,\infty} \\ \mu_i x_{s,\infty} = (x_\infty^\top x_\infty) y_{s,\infty} \end{cases} \Rightarrow \mu_i^2 x_{s,\infty} y_{s,\infty} = (x_\infty^\top x_\infty)(y_\infty^\top y_\infty) x_{s,\infty} y_{s,\infty}$$

Obviously, $x_\infty = y_\infty = 0$ is a fixed point. If $x_{s,\infty} \neq 0$, then $y_{s,\infty} \neq 0$ and we have $\mu_i^2 = (x_\infty^\top x_\infty)(y_\infty^\top y_\infty)$. Since $\mu_i \neq \mu_j$ for $i \neq j$, there exists at most one $i$, s.t. $\mu_i^2 = (x_\infty^\top x_\infty)(y_\infty^\top y_\infty)$ and all the $x_{s,\infty}, y_{s,\infty}$ s.t. $s \in S_j$, $j \neq i$ is zero. □

This theorem identifies that each of these fixed points is concentrated in one of the positions of the eigenvalue blocks. Note the fixed points contain both stable and unstable ones, i.e., all the global minima and saddles are of the above form. Moreover, the global minima are the points obeying $\sum_{s \in S_{max}} x_{s,\infty}^2 \cdot \sum_{s \in S_{max}} y_{s,\infty}^2 = \mu_{max}^2$, $x_{s,\infty} = y_{s,\infty} = 0$ for $s \notin S_{max}$ and also $\|x_\infty\| \|y_\infty\| = \mu_{max}$. The saddles are the ones with $\mu_i \neq \mu_{max}$, $\forall i = 1, \cdots, n$.

**Remark I.2** (The alignment of $x_\infty$ and $y_\infty$ at the global minimum). At the global minimum of the objective (19) with diagonal and non-negative $A$ defined above, $x_\infty y_\infty^\top$ is the rank-1 approximation of $A$ with the largest eigenvalue, i.e., we have $\mu_{max} = \text{tr}(x_\infty y_\infty^\top) = x_\infty^\top y_\infty$. Also, from the above theorem, $\|x_\infty\| \|y_\infty\| = \mu_{max}$. Therefore for each global minimum, there exists a scalar $l > 0$, s.t., $x_\infty = l y_\infty$, i.e., $x_\infty$ and $y_\infty$ are aligned at the global minimum.

Based on the analytical form of eigenvalues of the Jacobian of GD map (19) in Theorem I.1, we can establish the following relation between $x_\infty$ and $y_\infty$ at the global minimum via stability analysis.

**Theorem I.3.** *For almost all initial conditions, if GD converges to a minimum, then $x_\infty^\top y_\infty = \mu_{max}$ and*

$$\|x_\infty\|^2 + \|y_\infty\|^2 < \frac{2}{h}; \tag{20}$$

*the extent of balancing is quantified by*

$$\|x_\infty - y_\infty\|^2 < \frac{2}{h} - 2\mu_{max}. \tag{21}$$

*Proof.* Since each unstable fixed point has its stable set with negligible size when compared to the stable set of stable fixed point, then for almost all initial conditions, if GD converges to a global minimum, this minimum is a stable fixed point of the GD map.

From the above theorem, we know the fixed points are $\mu_i^2 = (x_\infty^\top x_\infty)(y_\infty^\top y_\infty)$. Assume $\mu^2 = (x_\infty^\top x_\infty)(y_\infty^\top y_\infty)$.

For the Jacobian $J = I_{2n} + \begin{pmatrix} -h(y_\infty^\top y_\infty)I_n & hA - 2hx_\infty y_\infty^\top \\ hA^\top - 2hy_\infty x_\infty^\top & -h(x_\infty^\top x_\infty)I_n \end{pmatrix} = I_{2n} + \tilde{A}$, since the lower two blocks commute, we have

$$
\begin{aligned}
\det(\lambda I - \tilde{A}) &= \det(\lambda^2 I_n + \lambda h(x_\infty^\top x_\infty + y_\infty^\top y_\infty)I_n + h^2\mu^2 I_n - h^2 A^2 \\
&\quad + h^2(2Ay_\infty x_\infty^\top + 2x_\infty y_\infty^\top A - 4x_\infty y_\infty^\top y_\infty x_\infty^\top)) \\
&= \det(\lambda^2 I_n + \lambda h(x_\infty^\top x_\infty + y_\infty^\top y_\infty)I_n + h^2\mu^2 I_n - h^2 A^2 \\
&\quad + h^2(2(A - x_\infty y_\infty^\top)y_\infty x_\infty^\top + 2x_\infty y_\infty^\top(A - y_\infty x_\infty^\top))) \\
&= \det(\lambda^2 I_n + \lambda h(x_\infty^\top x_\infty + y_\infty^\top y_\infty)I_n + h^2\mu^2 I_n - h^2 A^2) \\
&= \prod_{i=1}^n (\lambda^2 + h(x_\infty^\top x_\infty + y_\infty^\top y_\infty)\lambda + h^2(\mu^2 - \mu_i^2)).
\end{aligned}
$$

Hence the two values, 1 and $1 - h(x_\infty^\top x_\infty + y_\infty^\top y_\infty)$, are eigenvalues of $J$ at each nonzero fixed point.

If $\mu = \max\{\mu_i\}$, then $|1 - h(x_\infty^\top x_\infty + y_\infty^\top y_\infty)| < 1 \Rightarrow$ all the eigenvalues are bounded by 1. Hence $x_\infty^\top x_\infty + y_\infty^\top y_\infty < \frac{2}{h}$.

If $\mu \neq \max\{\mu_i\}$, then $1 + \frac{1}{2}(-h(x_\infty^\top x_\infty + y_\infty^\top y_\infty) + \sqrt{h^2(x_\infty^\top x_\infty + y_\infty^\top y_\infty)^2 + 4h^2((\mu_i^2 - \mu^2))}) > 1$ for $\mu_i \geq \mu$. It has at least one unstable direction. It's stable set is measure 0.

The second inequality of the theorem follows from $x_\infty^\top y_\infty = \mu$ at the global minimum because of diagonal and non-negative $A$ and SVD. $\qquad\square$

## I.2 GENERAL MATRIX FACTORIZATION FOR NON-NEGATIVE DIAGONAL MATRIX

In this section, we consider problem (5) via GD iteration but with $A \in \mathbb{R}^{n \times n}$ to be an arbitrary non-negative diagonal matrix.

Let $\mu_1 \geq \mu_2 \geq \cdots \geq \mu_n \geq 0$ be the diagonal elements of $A$. Let $r = \mathrm{rank}(A)$ which of course satifies $r \leq n$. Also assume that $\|A\|_\mathsf{F}$ does not scale with $n$ or $d$ (without loss of generality we can assume $\|A\|_\mathsf{F} = \mathcal{O}(1)$). Then we will have our balancing result in the following.

**Theorem I.4.** *For almost all initial conditions, if GD for (5) converges to a global minimum, then there exists $c = c(n, d) > c_0$ with constant $c_0 > 0$ independent of $n, d$, and learning rate $h$, such that, $(X, Y)$ satisfies*

$$c(\|X_\infty\|_\mathsf{F}^2 + \|Y_\infty\|_\mathsf{F}^2) < \frac{2}{h},$$

*and the extent of balancing is quantified by*

$$\|X_\infty - Y_\infty\|_\mathsf{F}^2 < \frac{2}{ch} - 2\sum_{i=1}^{\min\{d,r\}} \mu_i.$$

*Proof.* Let $r =$ rank$(A) \leq n$. Since each unstable fixed point has its stable set with size negligible when compared to the stable set of stable fixed point, then for almost all initial conditions, if GD converges to a global minimum, this minimum is a stable fixed point of the GD map.

For matrix $X = (x_1\ x_2\ \cdots\ x_d) \in \mathbb{R}^{n \times d}$, where $x_i \in \mathbb{R}^n$ is the $i$th column vector, we vectorize the variable and get $vec(X) = \begin{pmatrix} x_1 \\ x_2 \\ \vdots \\ x_d \end{pmatrix}$. Then we can rewrite the GD iteration,

$$\begin{cases} X_{k+1} = X_k + h(A - X_k Y_k^\top)Y_k \\ Y_{k+1} = Y_k + h(A^\top - Y_k X_k^\top)X_k \end{cases}$$

$$\Rightarrow \begin{cases} vec(X_{k+1}) = vec(X_k) + h(I \otimes A - T(vec(Y_k)vec(X_k)^\top))vec(Y_k) \\ vec(Y_{k+1}) = vec(Y_k) + h(I \otimes A - T(vec(X_k)vec(Y_k)^\top))vec(X_k) \end{cases},$$

where $\otimes$ is the Kronecker product, and $T : \mathbb{R}^{nd \times nd} \to \mathbb{R}^{nd \times nd}$ is a linear operator, s.t.

$$T\left( \begin{pmatrix} M_{11} & \cdots & M_{1d} \\ \vdots & & \vdots \\ M_{d1} & \cdots & M_{dd} \end{pmatrix} \right) = \begin{pmatrix} M_{11}^\top & \cdots & M_{1d}^\top \\ \vdots & & \vdots \\ M_{d1}^\top & \cdots & M_{dd}^\top \end{pmatrix},$$

$$\text{then} \qquad T(vec(Y)vec(X)^\top) = \begin{pmatrix} x_1 y_1^\top & x_2 y_1^\top & \cdots & x_d y_1^\top \\ x_1 y_2^\top & x_2 y_2^\top & \cdots & x_d y_2^\top \\ \vdots & \vdots & & \vdots \\ x_1 y_d^\top & x_2 y_d^\top & \cdots & x_d y_d^\top \end{pmatrix}.$$

The Jacobian of the vectorized GD map is (replace $X_k$ and $Y_k$ by $X$ and $Y$) $I_{2nd} - hM$, where

$$M = \begin{pmatrix} Y^\top Y \otimes I_n & T(vec(Y)vec(X)^\top) + I \otimes (XY^\top - A) \\ T(vec(X)vec(Y)^\top) + I \otimes (YX^\top - A) & X^\top X \otimes I_n \end{pmatrix}.$$

We would like to obtain an estimation of the eigenvalues of $I - hM$ for stability analysis. Since $M$ is the Hessian of the objective at the global minimum defined, all the eigenvalues are non-negative. Also, since $\|A\|_\mathsf{F}$ is independent of $h$, $n$ and $d$, we can just assume without loss of generality $\|A\|_\mathsf{F} = \mathcal{O}(1)$ and then take $\|X\|_\mathsf{F}$ and $\|Y\|_\mathsf{F}$ to be $\mathcal{O}(1)$ at each global minimum. Then

$$\mathrm{tr}(M) = n(\|X\|_\mathsf{F}^2 + \|Y\|_\mathsf{F}^2) = \mathcal{O}(n).$$

Next consider $\mathrm{tr}(M^2)$. We only need the diagonal blocks of $M^2$. The two diagonal blocks are the following

$$M_1 = (Y^\top Y \otimes I_n)^2$$
$$+ [T(vec(Y)vec(X)^\top) + I \otimes (XY^\top - A)][T(vec(X)vec(Y)^\top) + I \otimes (YX^\top - A)],$$
$$M_2 = (X^\top X \otimes I_n)^2$$
$$+ [T(vec(X)vec(Y)^\top) + I \otimes (YX^\top - A)][T(vec(Y)vec(X)^\top) + I \otimes (XY^\top - A)].$$

Since we evaluate this matrix at the minimum, we have the gradient equals 0, i.e.,

$$(A - XY^\top)Y = 0, (A - YX^\top)X = 0.$$

By the mixed-product property of Kronecker product, we have

$$M_1 = (Y^\top Y)^2 \otimes I_n + T(vec(Y)vec(X)^\top)T(vec(X)vec(Y)^\top)$$
$$+ I_d \otimes [(A - XY^\top)(A - YX^\top)],$$
$$M_2 = (X^\top X)^2 \otimes I_n + T(vec(X)vec(Y)^\top)T(vec(Y)vec(X)^\top)$$
$$+ I_d \otimes [(A - YX^\top)(A - XY^\top)].$$

Then

$$\text{tr}(M^2) = 2\|X\|_{\mathsf{F}}^2\|Y\|_{\mathsf{F}}^2 + n(\|X^\top X\|_{\mathsf{F}}^2 + \|Y^\top Y\|_{\mathsf{F}}^2) + d\|A - XY^\top\|_{\mathsf{F}}^2.$$

Also, this is the global minimum, meaning

$$\|A - XY^\top\|_{\mathsf{F}}^2 = \text{tr}((A - XY^\top)(A - YX^\top)) = \begin{cases} \sum_{i=d+1}^r \mu_i^2, & d < r \\ 0, & d \geq r \end{cases}$$

Therefore, when $d \geq r$, we have

$$\text{tr}(M^2) = 2\|X\|_{\mathsf{F}}^2\|Y\|_{\mathsf{F}}^2 + n(\|X^\top X\|_{\mathsf{F}}^2 + \|Y^\top Y\|_{\mathsf{F}}^2) = \mathcal{O}(n).$$

When $d < r$, we have

$$\begin{aligned}\text{tr}(M^2) &= 2\|X\|_{\mathsf{F}}^2\|Y\|_{\mathsf{F}}^2 + n(\|X^\top X\|_{\mathsf{F}}^2 + \|Y^\top Y\|_{\mathsf{F}}^2) + d\|A - XY^\top\|_{\mathsf{F}}^2 \\ &< 2\|X\|_{\mathsf{F}}^2\|Y\|_{\mathsf{F}}^2 + n(\|X^\top X\|_{\mathsf{F}}^2 + \|Y^\top Y\|_{\mathsf{F}}^2) + n\|A - XY^\top\|_{\mathsf{F}}^2 = \mathcal{O}(n).\end{aligned}$$

Therefore, the number of non-zero eigenvalues is $N = \mathcal{O}(n)$. Let $\lambda_{max}$ be the largest eigenvalue of $M$. Then there exist a constant $c_0 > 0$

$$\lambda_{max} \geq \text{tr}(M)/N \geq c_0(\|X\|_{\mathsf{F}}^2 + \|Y\|_{\mathsf{F}}^2).$$

From stability analysis, we need the absolute values of the eigenvalues of $I - hM$ to be bounded by 1, i.e.,

$$1 - hc_0(\|X\|_{\mathsf{F}}^2 + \|Y\|_{\mathsf{F}}^2) \geq 1 - h\lambda_{max} \geq -1 \Rightarrow c_0(\|X\|_{\mathsf{F}}^2 + \|Y\|_{\mathsf{F}}^2) \leq \frac{2}{h}.$$

Obviously, if we pick a constant $c = c(n, d)$ for each $n$ and $d$ instead of $c_0$, the above inequality also holds with $c(n, d) > c_0$.

For the derivation of the second inequality, since at the global minimum $\text{tr}(X_\infty Y_\infty^\top) = \sum_{i=1}^{\min\{d,r\}} \mu_i = \sum_{i=1}^{\min\{d,n\}} \mu_i$ because for non-negative diagonal $A$, its SVD of the non-zero part satisfies $U_{non-zero} = V_{non-zero}$, we have

$$\|X_\infty - Y_\infty\|_{\mathsf{F}}^2 = \|X_\infty\|_{\mathsf{F}}^2 + \|Y_\infty\|_{\mathsf{F}}^2 - 2\,\text{tr}(X_\infty Y_\infty^\top) \leq \frac{2}{ch} - 2\sum_{i=1}^{\min\{d,n\}} \mu_i.$$

$\square$

### I.3 FROM DIAGONAL MATRICES TO GENERAL MATRICES

*Proof of Theorem 5.1.* First all the minima of this problem are global minima and all the saddles are unstable fixed points. Since each unstable fixed point has its stable set with size negligible when compared to the stable set of stable fixed point, then for almost all initial conditions, if GD converges to a point, for almost all situations, this point is a stable fixed point of the GD map.

By singular value decomposition (SVD), $A = UDV^\top$, where $U, D, V \in \mathbb{R}^{n \times n}$, $U$ and $V$ are orthogonal matrices, and $D$ is a non-negative diagonal matrix. Let $X_k = UR_k$, $Y_k = VS_k$. Then

$$\begin{cases} X_{k+1} = X_k + h(A - X_k Y_k^\top)Y_k \\ Y_{k+1} = Y_k + h(A - X_k Y_k^\top)^\top X_k \end{cases}$$

$$\Leftrightarrow \begin{cases} UR_{k+1} = UR_k + h(UDV^\top - UR_k S_k^\top V^\top)VS_k \\ VS_{k+1} = VS_k + h(UDV^\top - UR_k S_k^\top V^\top)^\top UR_k \end{cases}$$

$$\Leftrightarrow \begin{cases} R_{k+1} = R_k + h(D - R_k S_k^\top)S_k \\ S_{k+1} = S_k + h(D - R_k S_k^\top)^\top R_k \end{cases}.$$

Therefore, problem (5) is equivalent to the following problem solved by GD

$$\min_{R,S \in \mathbb{R}^{n \times d}} \|D - RS^\top\|_{\mathsf{F}}^2 / 2. \tag{22}$$

From Theorem I.3 and I.4, we obtain: if GD for (22) converges to a global minimum $(R, S)$, then there exists a constant $c > 0$ independent of $n$, $r$, $d$, and $h$, s.t., the limiting minimum satisfies

$$c(\|R\|_{\mathsf{F}}^2 + \|S\|_{\mathsf{F}}^2) \leq \frac{2}{h},$$

and the extent of balancing is quantified by

$$\|R - S\|_{\mathsf{F}}^2 \leq \frac{2}{ch} - 2 \sum_{i=1}^{\min\{d,r\}} \mu_i.$$

Especially, when $d = 1, n \in \mathbb{N}^+, c = 1$.

Since $X = UR, Y = VS \Rightarrow R = U^\top X, S = V^\top Y$, we have $\|R\|_{\mathsf{F}} = \|X\|_{\mathsf{F}}, \|S\|_{\mathsf{F}} = \|Y\|_{\mathsf{F}}$, and $\|R - S\|_{\mathsf{F}} = \|U^\top X - V^\top Y\|_{\mathsf{F}} = \|X - (UV^\top)Y\|_{\mathsf{F}}$. The we obtain the first and second inequalities in Theorem 5.1. Also, by SVD, $\|XY^\top\|_{\mathsf{F}}^2 = \sum_{i=1}^{\min\{d,n\}} \mu_i^2 \leq \|X\|_{\mathsf{F}}^2 \|Y\|_{\mathsf{F}}^2$, we obtain the third inequality. $\qquad\square$

