# OpenReview forum: "Large Learning Rate Tames Homogeneity: Convergence and Balancing Effect"
_ICLR.cc/2022/Conference — ICLR 2022 Poster_

### Official Review · Reviewer_MuC1 · 2021-10-30

**Correctness:** 4
**Technical Novelty And Significance:** 4
**Empirical Novelty And Significance:** Not applicable
**Recommendation:** 8
**Confidence:** 3

**Main Review:**

I marginally recommend rejection.

STRENGTHS: This paper tackles an interesting question of whether large learning rate implicitly regularizes gradient descent, which has not  previously received much theoretical attention. For the 1xd and dx1 special cases of matrix factorization, experimental and (very involved) theoretical evidence is given for this regularization. The paper is for the most part very clearly written.

WEAKNESSES: The main weakness, which the paper completely elides, is that the result does not actually extend to the general matrix factorization setting (the paper states "We theoretically show the convergence of GD with large learning rate", but I could not find a theorem statement corroborating this in the general case). In particular, the main theorem for the general case assumes that the learning rate is such that GD converges to a global minimum. This seems to ignore one of the main difficulties, which is proving that even with a large learning rate GD will converge (since classical optimization arguments break down for large learning rate, and it seems quite plausible that GD could have chaotic or periodic behavior). Indeed, under the assumption that GD converges, the proof of the theorem is simply a local argument about the curvature of the various critical points (the "stability analysis"), whereas I would expect that some global argument is necessary to get convergence.

In my opinion, the paper would be much stronger if convergence were properly addressed in the general case.

OTHER COMMENTS:
- The special case proofs are quite technically involved. Is there hope for any insights from these cases that might extend to the general matrix factorization problem?


**Summary Of The Paper:**

This paper studies the properties of gradient descent with large learning rate in the matrix factorization problem. The goal is to understand when gradient descent converges to a global minimum where the two factors are roughly balanced in norm, which is a shallow minimum that may be hoped to generalize better. For small learning rates (i.e. where classical optimization results guarantee convergence to a critical point), this does not happen. Hence, this paper considers learning rates beyond this regime.

For two special cases (when the matrices are 1xd or dx1), it is shown that under a large learning rate, gradient descent still converges to a global minimum, and this minimum is roughly "balanced". For the general case, it is shown that for almost any initial point, if the learning rate is such that gradient descent converges to a global minimum, then this minimum is roughly balanced.

**Summary Of The Review:**

Although this paper raises an interesting problem of implicit regularization in matrix factorization with large learning rate, and provides evidence in special cases, it does not live up to its claims for the general case: failure to converge is a well-known issue with large learning rates, so the results for the general case which assume convergence are somewhat incomplete. It's also not clear whether the proofs in the special cases can provide any insight for the general case.

---

> ### Author Response · Authors · 2021-11-22
> **Response to Reviewer MuC1**
>
> Thank you very much for the comments and questions. Indeed our proofs of convergence don't directly generalize to the most general case, and we revised the wording to make sure nothing misleading is implied and no overclaim is made. However, the technical difficulty associated with this proof should not be underestimated, and yet its absence does not significantly impair the main claim of this paper, which is that larger $h$ gives a tighter bound on the extent of eventual unbalancedness. Therefore, is there any possibility you could kindly reconsider, based on the significance of the existing results, and the high nontrivality of the problem? More specifics follow:
>
> Regarding the significance of this paper, may we emphasize that its main focus is the balancing effect, created by large learning rates? Unbalancedness between $X$ and $Y$ can create a lot of numerical and analytical difficulties, and that's why the community devoted years of efforts to this problem (see e.g., Cabral et al. (2013), Tu et al. (2016), Ge et al. (2017), Du et al. (2018), Li et al. (2019a), Ye & Du (2021), Ma et al. (2021), Liu et al. (2021), as discussed in our Sec.2). Some of the recent results (e.g., Du et al. (2018), Ye & Du (2021), Ma et al. (2021)) showed that the initial extent of norm balancing can be maintained by GD if the learning rate is small, and this is already well regarded as nontrivial important progress. This work goes one more step and studies the other side of the story, where learning rate is large, and discovered, rigorously, that even unbalanced initial condition is not a problem either and will eventually get balanced. Since balanced minimizer provably corresponds to flatter loss landscape nearby, this work also quantitatively shows that larger learning rate gives flatter minimizer, at least for the homogeneous problem of matrix factorization.
>
> Regarding the technical contribution, may we start by recalling that (1) traditional theoretical tools only allow general convergence analysis when $h<2/L$, where $L$ is the global Lipschitz constant of the gradient? Specialized rigorous results that go beyond this threshold exist, such as Kong & Tao (2020), but we're not aware of any result that could extend to the problem studied here. (2) Our objective function does not have global Lipschitz gradient. So, we invented a set of of new mathematical machinaries to prove convergence. Even though the most general case of matrix factorization still defies rigorous convergence guarantee, we did manage to prove convergence in several nontrivial cases including $1\times d$ and $n\times 1$, which cover both overparameterized and underparameterized situations.
>
> In fact, as the expert reviewer recognizes, our proofs for the special cases are already very involved. There is some intuition that we gained out of these proofs, namely there is a first phase in which GD searches for flatter (i.e., in our case, more balanced) minimum, and then a second phase in which convergence becomes monotone. We think this insight definitely extends to the general matrix factorization setup. However, again, the difficulty of translating the intuition into rigorous mathematics should not be underestimated. For example, a generalization of our alignment result to matrices might be needed, but that is highly nontrivial, and we know similiar problems remain to be a major challenge in many deep learning problems. While we have not succeeded in proving convergence with large learning rate for the most general case, we hope the reviewer could agree with us that balancing is a very challenging, yet meaningful problem. As it has been studied by many leading researchers for a decade, we hope it is okay that a single paper of ours doesn't complete everything and we are allowed to take step by step.
>
> Of course, one could be concerned that large learning rate may not lead to convergence in the unproved case, namely the most general setup of matrix factorization (Sec.5). Nevertheless, that's never observed in our empirical experiments, and we tend to think that having no proof of convergence is due to a limitation of analysis techniques, rather than the iterations being non-convergent. Thanks to the reviews, we added empirical results on this to the revision (in Appendix A). We conjecture that there is still a regime of learning rates, larger than $2/L$, that still lead to convergence and yet increasing balanceness.

---

> > ### Comment · Reviewer_MuC1 · 2021-11-28
> > **Thanks for the response**
> >
> > Thank you for the response and revision clarifying the theoretical results. I find the experiments for the general matrix factorization case to be a convincing demonstration that the two-phase intuition of balancing/convergence from the special cases should carry over to the general-dimension case (even if only in some Gaussian setting).
> >
> > As a result, I am inclined to increase my score. However, I am confused by one thing in the new experiments -- in Figure 11, this is an underparametrized factorization, and my impression is that the matrix A is generated randomly, so there should not be an exact factorization. So why does the loss seem to converge to 0 for several choices of step size? I wouldn't have expected that a Gaussian 100x100 matrix can be approximated by a product of 100x3 and 3x100 matrices to within error 10^{-10}; am I mistaken?

---

> > > ### Author Response · Authors · 2021-11-28
> > > **Round 2 response to Reviewer MuC1**
> > >
> > > We sincerely thank the reviewer for reading our response and revision, as well as the consideration of raising the score.
> > >
> > > > in Figure 11, this is an underparametrized factorization, and my impression is that the matrix A is generated randomly, so there should not be an exact factorization. So why does the loss seem to converge to 0 for several choices of step size?
> > >
> > > Sorry for the confusion but fortunately there is a simple explanation. It is indeed true that in the under-parameterized factorization case, the error of matrix approximation will not converge to zero. However, the loss we used in Fig.11 (and other figures too) is a shifted loss, which is the difference between the current matrix approximation error and the global minimum of error. The latter is analytically computable for our problem via SVD. This way, the shifted loss is expected to converge to zero no matter whether the problem is under- or over-parameterized. We chose to plot this shifted loss so that we could easily investigate the speed of convergence.
> > >
> > > We will include this explanation in the next version. Thanks very much for catching this important source of confusion.

---

> > > > ### Comment · Reviewer_MuC1 · 2021-11-29
> > > > **Thanks**
> > > >
> > > > Thank you for the clarification. I have increased my score.

---

### Official Review · Reviewer_Xmww · 2021-10-31

**Correctness:** 3
**Technical Novelty And Significance:** 3
**Empirical Novelty And Significance:** 3
**Recommendation:** 6
**Confidence:** 3

**Main Review:**

The problem is interesting. The proofs seem to be nontrivial. The technical contribution looks significant. However, I am not able to give a recommendation of acceptance due to the following issue:

* In line 160, the upper bound does not seem to be $\frac{4}{L}$. It seems to be closer to $\frac{2}{3L}$ because when $x_0, y_0$ are close to the solution, $x_0 \approx y_0$ and so $L \approx \|x_0\|^2  + \|y_0\|^2 = 2 \langle x_0, y_0 \rangle = 2 \mu.$ Then the bound on $h$ in line 152 becomes $\frac{1}{3\mu} = \frac{2}{3L}$.

Given this problem, I request the authors to argue how the proposed learning rate is large. Specifically, large learning rate compared to what?


**Summary Of The Paper:**

The authors study the problem of minimizing matrix factorization squared error with gradient descent. They show convergence with larger learning rates than previously considered. They further show that gradient descent has an implicit bias towards finding matrix factors which are magnitude-balanced in a certain sense.



**Summary Of The Review:**

With my current understanding, I recommend a reject. However, I am expecting that the authors will answer my question and I will be able to give a higher score.

---

> ### Author Response · Authors · 2021-11-22
> **Response to Reviewer Xmww**
>
> Thanks very much for your comments, an interesting question, and an opportunity of clarification.
>
> Your calculation is absolutely correct, but fortunately it is not contradicting our claims for the following reason: Our result  can be roughly stated as, whether or not the initial condition is balanced, large enough learning rate can ensure that the limit of gradient descent is balanced to some quantitative extent. (1) If the initial condition is already well balanced, we don't have a reason or a result to warrant that the final result is more balanced. (2) If the **initial condition** is however **unbalanced**, then we have two facts: (2a) learning rate can be pushed to almost as large as 4/L, and GD iterations will still converge; (2b) within this range (of stability), larger $h$ gives a smaller bound on the final extent of unbalancedness.
>
> To be more precise and also explicitly connect to the concrete example that you constructed (which is very helpful for us to explain, thank you), let us start by assuming that, "when $x_0,y_0$ are close to the solution" meant "when $x_0,y_0$ are close to one of the infinite many global minimizers, which is balanced, i.e. $x_0 \approx y_0$". In this case, our theorem is still correct, however, like mentioned, not interesting because it simply says, when $h\lessapprox 2/(3L)$, convergence is guaranteed and the limit is balanced. *Nevertheless*, we'd like to emphasize that besides balanced global minimizer ($\textbf{x} \approx \textbf{y}$ and $\langle\textbf{x},\textbf{y}\rangle=\mu$), there are also infinitely many unbalanced ones, because $c\textbf{x},\textbf{y}/c$ will also be a global minimizer for any nonzero scalar constant $c$. Suppose we start arbitrarily close to one of them, corresponding to $|c|\gg 1$, then the initial condition is very unbalanced, and $\|\textbf{x}_0\|^2+ \|\textbf{y}_0\|^2 \gg \mu$ despite that $\langle\textbf{x}_0,\textbf{y}_0\rangle=\mu$. If a small learning rate, defined as $h<2/L$ where $L$ is the local Lipschitz constant ($L=\|\textbf{x}_0\|^2+ \|\textbf{y}_0\|^2$), is used, then GD can simply just converge to this unbalanced near-by global min. However, if we use
> $$h = \min\bigg(\{\frac{4}{\|x_0\|^2+\|y_0\|^2+4\mu},\  \frac{1}{3\mu}\} \bigg\)= \frac{4}{\|x_0\|^2+\|y_0\|^2+4\mu} \approx \frac{4}{\|x_0\|^2+\|y_0\|^2}=\frac{4}{L},$$ which is  the bound given by Thm.3.1, then, surprisingly, GD will travel very far, despite it started already very close to a global min, to a different global min which is much more balanced. More details can be found in Corollary 3.3 and that's why we call $h>2/L$ a large learning rate. Finally, allow us to emphasize that the above discussion is just an example, and the initial condition doesn't have to be close to any minimizer; as long as $\|x_0\|^2+\|y_0\|^2 \gg \mu$, learning rate as large as $\approx 4/L$ still gives convergent iterates, as well as a shrinking bound on the eventual unbalanceness.

---

> > ### Comment · Reviewer_Xmww · 2021-11-28
> > **Thanks for the clarification, but I have another concern.**
> >
> > Thanks to the authors for clarifying. This clarification should go into the paper where they claim that the learning rate can be almost $\frac 4L.$
> >
> > There is one more sanity check, that I am not able to verify.
> >
> > Suppose $x_0 y_0 = \mu.$ I claim that Theorem 3.2 implies $h \leq \frac 2L$. If my claim is true, the argument that the learning rate is large is not valid. Following is the justification for my claim.
> >
> > First, because the gradient is 0, $x_k =x_0, y_k=y_0$ for all $k\geq 1.$   So, there should not be any shrinkage in |x-y| via GD.
> >
> > **Assume $ \frac{4}{x_0^2 + y_0^2 +4\mu} \leq \frac{1}{3\mu}$ that is, $x_0^2 + y_0^2 \geq 8 \mu.$**
> >
> > Then the largest possible learning rate via Theorem 3.1 is $h = \frac{4}{x_0^2 + y_0^2 + 4\mu}.$ Substituting this $h$, Theorem 3.2 yields
> > $$
> > x^2 + y^2 \leq \frac 2h  \Leftrightarrow x_0^2 + y_0^2 \leq \frac 12 (x_0^2 + y_0^2) + 2\mu \Leftrightarrow x_0^2 + y_0^2 \leq 4\mu
> > $$
> > This is in contradiction to our assumption that $x_0^2 + y_0^2 \geq 8 \mu.$
> >
> > **Therefore $x_0^2 + y_0^2 < 8 \mu.$**
> > Then $h = \frac{1}{3\mu}$ and Theorem 3.2 yields
> > $$
> > x^2 + y^2 \leq \frac 2h \Leftrightarrow x_0^2 + y_0^2 \leq 6\mu
> > $$
> > Then
> > $$
> > h = \frac{1}{3\mu} \leq \frac{2}{x_0^2 + y_0^2} = \frac 2L.
> > $$

---

> > > ### Author Response · Authors · 2021-11-28
> > > **Round 2 response to Reviewer Xmww**
> > >
> > > We sincerely thank the reviewer for reading our response and revision. In particular, we really appreciate the reviewer's willingness of digging deep in our theory.
> > >
> > > > I claim that Theorem 3.2 implies $h \leq 2/L$. If my claim is true, the argument that the learning rate is large is not valid.
> > >
> > > This concern is not a problem, and does not contradict with the large learning rate.
> > >
> > > More precisely, in Theorem 3.2, we refer to initial condition in Theorem 3.1, which excludes a measure-zero set of initial conditions. This measure-zero set includes all the global minima $x_0y_0=\mu$, and therefore our theory is not affected by behaviors starting from these points. Thanks to the reviewer, we now realize this technical point was not expressed clearly enough, and we will revise the theorems' statements in the next version.
> > >
> > > > Thanks to the authors for clarifying. This clarification should go into the paper where they claim that the learning rate can be almost $4/L$.
> > >
> > > Certainly. Thanks.

---

> > > > ### Comment · Reviewer_Xmww · 2021-11-28
> > > > **Thanks for the second clarification**
> > > >
> > > > Thank you for the clarifications. I raised my score.

---

### Official Review · Reviewer_VWDy · 2021-10-31

**Correctness:** 4
**Technical Novelty And Significance:** 2
**Empirical Novelty And Significance:** 2
**Recommendation:** 5
**Confidence:** 4

**Main Review:**

Unfortunately, I feel that both the theory and the simulations in this paper do not provide sufficient evidence of the aforementioned claims to merit acceptance. Also I feel that the presentation of the paper could be improved.

My two major concerns are the following:

1. I  feel that the theorems do not really support the claim that only large step size leads to balancing (in contrast to small step size). For example, Theorem 3.2 makes some statement provided the step size h is not too large. Then the theorem gives an upper bound on the balancing between x and y. This bound becomes stronger, if the step size is larger but how does one know that a similar bound is not also available for small step size? Even more importantly, Theorem 3.1 and 3.2 require an assumption on the step size h which depends on the initialization x_0 and y_0 (equation before l. 153). Note that the more unbalanced x_0 and y_0 are, the smaller the learning rate needs to be (and the weaker the statement in Theorem 3.2 becomes.) Hence, this result aper does not apply for unbalanced initialization and large learning rate. Analogous concerns hold for the theorems in Section 4 and Section 5. (One minor(?!) concern: Theorem 5.1 assumes already that there is convergence, i.e. the statement only holds provided that there is already convergence.)

2. The authors could provide much more experimental support. To me it seems that the only experiments which document that larger step size leads to more balancing are provided in Figure 4. However, I feel that much more extensive experiments are needed. (A larger choice of different step sizes and also a larger amount of initializations, for example scale of initialization and balancing at initialization.) Moreover, I feel that these experiments could be made more convincing if not only matrix factorization would have been considered but also the more challenging problems of matrix completion or matrix sensing.

Further comments:

-There is no labeling of the x-axis in Figures 2 and 4. I assume that this should be the number of iterations, right?!

-In both the first and third figure, gradient descent does not converge to zero loss, but the loss does become constant after some amount of iterations. Is this due to the fact that GD has reached machine precision? How do you exclude that scenario? Also it might be made more clear in the caption that the left two figures correspond to one learning rate and the two figures on the right correspond to one learning rate.

-l. 203 " We expect the convergence in (4) is 204 more complicated than ...". What does "more complicated convergence" mean?

-I feel that the readability could be increased if Theorem 3.1 and Theorem 3.2 would have been combined, i.e. one theorem instead of two. The same is true for Theorem 4.1, 4.2, and 4.3.

-I feel that the formulation of Theorem 1.1 is somewhat unclear. First of all, note that the variable "A" has been overloaded. It is the matrix in (1) as well as a scalar defined in Thm 1.1. But more importantly, the statement is unclear. For example, what happens if I set the scalar A=2/h. Then I obtained perfect balancing between X and Y, but this should not be possible?!



**Summary Of The Paper:**

This paper is concerned with the non-convex matrix factorization problem. This paper considers the square loss, which is optimized via gradient descent. Due to symmetries of the problem, there are infinitely many different global minima. This paper aims to understand to which minima gradient descent with large learning rate (=step size) converges to.

The main claim of this paper is that gradient descent with large learning rate has an implicit regularization effect, meaning that it converges to factors of the underlying matrix, which are implicitly balanced in norm. To substantiate the claim, the paper provides several theoretical results as well as some numerical experiments.



**Summary Of The Review:**

As both of theory and experiments do not substantiate the claims of the paper enough, I unfortunately cannot recommend acceptance.

---

> ### Author Response · Authors · 2021-11-22
> **Response 1 to Reviewer VWDy**
>
> Thanks very much for your comments and an opportunity of clarification. Below is an itemized list of our responses.
>
> > theorems do not really support the claim that only large step size leads to balancing (in contrast to small step size). Theorem 3.2 makes some statement provided the step size h is not too large. Then the theorem gives an upper bound on the balancing between x and y. This bound becomes stronger, if the step size is larger but how does one know that a similar bound is not also available for small step size?
>
> It is established in the literature that small step size maintains the initial extent of balancing; see e.g., Du et al. (2018), Ye & Du (2021), and Ma et al. (2021), which we cited and discussed in the Introduction. Therefore, an increased balance as quantified by our theorem won't happen with small step sizes. Experiments that we added to the revision in Appendix A also illustrate this, just in case something concrete is also helpful.
>
> > (continued discussion on large vs small learn rate) Theorem 3.1 and 3.2 require an assumption on the step size h which depends on the initialization x_0 and y_0 (equation before l. 153). Note that the more unbalanced x_0 and y_0 are, the smaller the learning rate needs to be (and the weaker the statement in Theorem 3.2 becomes.)
>
> Perhaps the source of this concerns lies in the definition of "large learning rate". What we call large learning rate is a relative notion, as it is not in terms of the numerical value but in comparison with the scale of the problem, quantified by what classical optimization theory can handle. More precisely, classical optimization theory gives that if $h<1/L$ where $L$ is the **global** Lipschitz constant of the objective function's gradient, then there is guaranteed monotonic convergence to a local minimum, and if $h<2/L$ then there is again convergence but monotonicity may be lost. When $h>2/L$, there is no general theory any more, and oftentimes gradient descent iterations can blow up; therefore, we call **such** $h$ a large learning rate. Things need to be compared with some scaling gauge of the objective function, because a learning rate of 0.1 is small for loss function $x^2/2$ but large for loss function $100x^2/2$, which is nevertheless essentially the same (only a change of unit) as the former.
>
> Now comes one of the keys to appreciating the theoretical contribution of this work: our objective function is **not** globally Lipschitz (and this is well acknowledged in the literature to be a significant challenge). Nevertheless, we can still make the analysis of the highly nonlinear dynamics to work, by only requiring the **local** Lipschitz constant of the gradient, at the initialization. This is not a drawback but a significant improvement, because there is no reason one can get around the dependence on scaling as mentioned before. Even if we had global Lipschitzness, dependence on the Lipschitz constant would still be needed for defining what learning rate is large, and the global Lipschitz constant would only be larger or equal to the local one, which makes our learning rate to be an **even larger** one when compared to the scaling of the problem, and thus further nontrivial.
>
> In addition, even if we had global Lipschitz gradient, the fact that we provably have convergence with learning rate as large as $h\lesssim 4/L$ is already highly nontrivial (even without results on balancing). This is because traditional optimization theory cannot give convergence for $h>2/L$, let alone balancing. Our theory is specific to the problem studied here though.

---

> > ### Author Response · Authors · 2021-11-22
> > **Response 2 to Reviewer VWDy**
> >
> > >Hence, this result does not apply for unbalanced initialization and large learning rate.
> >
> > The above response should (hopefully) dispel this concern, but just in case more explanations could be helpful, here is a concrete example based on Corollary 3.3, which is an example of Theorem 3.2 (similar arguments work for Section 4 too). In this corollary, let us choose an initial condition with arbitrarily large $\|x_0-y_0\|$ (and hence arbitrarily large unbalanceness) that is sufficiently far from the origin ($\|x_0\|^2+\|y_0\|^2>8\mu$), and a step size $h=4/(\|x_0\|^2+\|y_0\|^2+4\mu)$. This $h$ is $\lessapprox 4/\hat{L}$ and thus $\lesssim 4/L$, where $\hat{L}=\|x_0\|^2+\|y_0\|^2$ is the local Lipschitz constant at initialization, and $L\geq \hat{L}$ is the global Lipschitz constant (if any; if gradient is not global Lipschitz, use the convention $L=+\infty$); therefore, it is a large learning rate -- note absolute numerical value is not meaningful and "large" is comparative; kindly see explanations in the previous bullet.
> >
> > Then, no matter how close the initial condition already is to a global minimizer (i.e., $(\mu-x_0 y_0^T)^2/2$ can be arbitrarily close to 0), the initial unbalancedness can be significantly reduced because we proved
> > $$\|x_\infty-y_\infty\|^2<\frac{1}{2}\|x_0-y_0\|^2+2\mu, $$ and since it is easy to have $\|x_0-y_0\|^2/2 \gg 2\mu$, the reduction is at least nearly a half. This also implies that GD will converge to a point that is very far from the initial location. These results should be contrasted with the small learning rate situation, where $\|x_\infty-y_\infty\| \approx \|x_0-y_0\|$ and $(x_\infty,y_\infty)$ will be very close to $(x_0,y_0)$.
> >
> > >The authors could provide much more experimental support... A larger choice of different step sizes and also a larger amount of initializations, for example scale of initialization and balancing at initialization....the more challenging problems of matrix completion *or* matrix sensing.
> >
> > Thanks for the suggestion. We have added a large amount of additional experiments, including all the parameter variations suggested by the reviewer, for various setups studied in our paper, as well as empirical results for matrix completion *and* matrix sensing, in Appendix A. Our theoretical claims remained true in all these results. For example, these experiments clearly show increasingly strong balancing effects once $h$ goes beyond values for which traditional optimization theory works (i.e., increasing in what we called "large learning rate" regime).
> >
> > >There is no labeling of the x-axis in Figures 2 and 4. I assume that this should be the number of iterations, right?!
> > >Also it might be made more clear in the caption that the left two figures correspond to one learning rate and the two figures on the right correspond to one learning rate.
> >
> > The observation and suggestion are much appreciated. We have made the revision accordingly.
> >
> > >In both the first and third figure, gradient descent does not converge to zero loss, but the loss does become constant after some amount of iterations. Is this due to the fact that GD has reached machine precision? How do you exclude that scenario?
> >
> > Thanks very much for making this note. Yes, the loss is actually converging to zero but its numerical value can get staggered around machine precision. This finite precision is in some sense unavoidable for common data types (we used double). Our code was in MATLAB which uses LAPACK for linear algebra, and scalar, vector and matrix operations can have different precisions. LAPACK is widely used, not only by MATLAB, and error saturation at a negligible value due to finite machine precision is very typical in optimization and computing.
> >
> > To "exclude" this scenario, one could increase the precision by, for example, using advanced data type and symbolic calculation. We don't see a need for this in this work though.
> >
> > >l. 203 " We expect the convergence in (4) is 204 more complicated than ...". What does "more complicated convergence" mean?
> >
> > Thanks for a great question. The additional complication refers to the alignment convergence (Theorem 4.1) specific to Sec.4, for which we proved $x_k$ and $y_k$ will be eventually parallel to each other. In Sec.3 (scalar factorization) there is no such result. Given the already involved proof of convergence in Sec.3 (see our sketch of proof in Figure 3), we concluded that this alignment adds extra complication in the sense that first, the alignment itself is very difficult to prove, and second, we had to combine this result with the framework of the proof in Sec.3 to get the convergence result in Sec.4.

---

> > > ### Author Response · Authors · 2021-11-22
> > > **Response 3 to Reviewer VWDy**
> > >
> > > >the formulation of Theorem 1.1 is somewhat unclear. First of all, note that the variable "A" has been overloaded...But more importantly, the statement is unclear. For example, what happens if I set the scalar A=2/h. Then I obtained perfect balancing between X and Y, but this should not be possible?!
> > >
> > > Thanks for the interesting comments.
> > >
> > > Regarding "A", it is an $n\times n$ matrix where $n$ can be any integer, including $n=1$ for Section 3 and Theorem 1.1. We have made further explanation in Theorem 1.1.
> > >
> > > Regarding the question on $A=2/h$, note one is **first** given an optimization problem, which fixes $A$, and **then** picks an optimizer and hyperparameters (e.g., $h$), according to the problem. Therefore, the optimization problem per se, which prescribes $A$ in our case, should not depend on $h$. Otherwise one could adversarially break a lot of the most widely used methods.
> > >
> > > What if one is given $A$ first and then choose an $h$ that coincidentally satisfies $h=2/A$, and thus $A=2/h$? Well, this $h$ is too large (even larger than the large learning rates for which we have theorems; see e.g., the condition on $h$ in Theorem 3.1, which Theorem 1.1 is an informal version of), and gradient descent can become unstable. So, there is no contradiction.

---

### Official Review · Reviewer_C6VY · 2021-11-03

**Correctness:** 3
**Technical Novelty And Significance:** 3
**Empirical Novelty And Significance:** Not applicable
**Recommendation:** 8
**Confidence:** 4

**Main Review:**

Strengths:
--The paper is well-written and very clear. Intuition is given so as to help the reader dissect the results.
--The theorems are interesting and tackle a nontrivial setting, since few works (to my knowledge) have derived strong results for convergence and implicit bias in this regime. They treat both convergence of the optimization but (interestingly) the balancing effect that occurs when using large learning rates.

Weaknesses:
--I could find no significant shortcomings in the paper.
--However, I do think the citations need to be improved. In particular, a number of the findings & discussions in Lewkowycz, et al. (2020) have some overlap with the results proven here but are not cited. For example, Lewkowycz, et al. (2020) found (1) the upper stability limit h~4/L for gradient descent, and a catapult phase that occurs between h> 2/L and < 4/L that involves search (phase 1) & subsequent convergence to a flatter minimum (phase 2). (2) The case n=1, d>1 is essentially the same problem setting analyzed there. (3) The final paragraph in Sec. 6 (starting after L278), discussing relationships to two-layer linear neural networks, was the setting studied in Lewkowycz, et al. (In fact, the u, v notation is identical.)


**Summary Of The Paper:**

This paper studies gradient descent optimization of matrix factorization (various forms, including the most generic case) under large learning rate. The authors prove convergence results that apply when the learning rate is larger than 2/L and up to ~4/L, and also show that optimization with large learning rates leads to dynamic balancing between the two matrix factors in the factorization, an effect that does not arise when using small learning rates. In proving some of the theorems, the authors observe the optimization can have two phases: one in which gradient descent is driven to search for flat regions, and a second phase near the global minimum where convergence occurs towards a balanced solution for the factorization problem.

**Summary Of The Review:**

I think this paper derives useful theoretical results on large learning rate optimization of matrix factorization that will be of interest to the community.

---

> ### Author Response · Authors · 2021-11-22
> **Response to Reviewer C6VY**
>
> Thank you for the positive and encouraging comments.
>
> The catapult mechanism in Lewkowycz et al. (2020) indeed has interesting resonance with our theoretical findings, and thanks to the reviewer's helpful suggestion, we expanded our discussion in the revised version to better reflect the connections and acknowledge their seminal contribution. Here are detailed responses:
>
> Lewkowycz et al. (2020) showed that in the NTK limit, allowing a constant large learning rate $\eta \in (2/L, 4/L)$ leads to first blowing-up and then convergent behavior of GD. Theoretical analysis is provided explicitly on linear two-layer networks with atomic data ($x = 1$ and $y = 0$). They also corroborate with experiments and demonstrate empirically that using large learning rate $\eta \in (2/L, 4/L)$ improves generalization performance. In our paper, we focus on matrix factorization problems, providing rigorous theoretical analysis. In the following, we elaborate on the connection between our theoretical findings to Lewkowycz et al. (2020).
>
> >(1) the upper stability limit h~4/L for gradient descent, and a catapult phase that occurs between h> 2/L and < 4/L that involves search (phase 1) & subsequent convergence to a flatter minimum (phase 2).
>
> We agree that a searching-to-convergence transition appears in both our work and Lewkowycz et al. (2020). However, the algorithmic behaviors are not the same. In fact, in our searching phase (phase 1), the objective function does not exhibit the blow-up phenomenon. In our convergence phase (phase 2), the analysis relies on a detailed state space partition (see Line 202) due to nonconvex nature of the problem, while the analysis in Lewkowycz et al. (2020) is akin to monotone convergence in a convex problem. We elaborate this connection in the revised manuscript in Line 172 - 177.
>
> >(2) The case n=1, d>1 is essentially the same problem setting analyzed there.
>
> We comment in Line 149 - 152 in the revised manuscript that overparameterized scalar factorization is also considered in Lewkowycz et al. (2020), which is formulated as using linear two-layer neural networks for regression. Nonetheless, we remark that the analysis is very different, since the infinite-width limit NTK perspective (equivalent to $d \to \infty$) is adopted in Lewkowycz et al. (2020). Our theory holds for any $d > 1$.
>
> >(3) The final paragraph in Sec. 6 (starting after L278), discussing relationships to two-layer linear neural networks, was the setting studied in Lewkowycz, et al. (In fact, the u, v notation is identical.)
>
> Despite the same $u, v$ notation, the difference is nontrivial. The work of Lewkowycz et al. (2020) considers both a simplified model and a full model. The former assumes single atomic input-output pairs, i.e., $x = 1$ and $y = 0$. The latter assumes $u, v \in \mathbb{R}^d$ with sufficiently large $d$. In contrast, we do not require these assumptions, and the discussion in section 6 can be viewed as a sensible generalization. We comment on this connection in Line 293 - 294 in the revised manuscript.

---

### Author Response · Authors · 2021-11-22
**A brief summary of our rebuttal and revision**

First of all, we sincerely thank all reviewers for your valuable time reviewing our submission.

In order to clarify misunderstandings, address important concerns of the reviewers, and improve the presentation of our paper, we significantly revised the paper and prepared detailed, itemized responses. Here is a summary of major points (numerous minor points not listed for brevity of this summary; all line numbers refer to the revised version):

* We added more explanations on our definition of large learning rates, how they go beyond what existing theories can give, why dependence on the local Lipschitz constant of the gradient is actually an advantage, and the different ramifications of balanced and unbalanced initialization, including when $h\lessapprox 4/L$ can be attained. See our responses to Reviewer VWDy, item #2&3, and responses to Reviewer Xmww, in addition to scattered revisions in the paper.
* We added Appendix A to include a large number of additional experiments, many on matrix factorization with additional systematically sampled parameter values, including dimensions, learning rates, and initializations. All experiments support our claims, including (1) the implicit regularization effect of balancing starts to manifest when $h$ goes beyond what traditional optimization theory can analyze, i.e. starting in the large learning rate regime (we already proved this, but it's nice to have visuals); (2) larger $h$ gives tighter bound on balancing (we already proved this, but it's nice to have visuals); (3) even though our convergence proofs for large learning rate only apply to cases in Sec.3&4, for the most general cases (Sec.5), the large learning regime still exists and GD still converges, with balancing, at least in all numerical experiments (given convergence, however, balancing is proved for all cases). The rest of newly added experiments are on additional problems of matrix sensing and matrix completion, and they demonstrate that balancing induced by large learning rate is an effect not specific to matrix factorization theoretically studied here.
* We elaborated on our connection and difference with Lewkowycz et al. (2020). Details are in Lines 149 - 152, Lines 172 - 177, and Lines 293 - 294.
* We modified our statement on our contributions to clarify that our convergence results hold for the two special cases (Section 3 and 4) while our balancing results hold for all the general matrix factorization in Line 52-58.

We sincerely hope our revision and rebuttal can better clarify the significance of this work, in terms of both providing quantitative understanding of a new mechanism (large learning rate induced balancing), which is well aligned with the important question of when one gets to a flatter minimum, and developing new tools to address highly nontrivial theoretical challenges. Thank you for working on our paper and helping us improve it.

---

### Decision · Program_Chairs · 2022-01-20

**Decision:**

Accept (Poster)

**Comment:**

The paper studies gradient descent for matrix factorization with a learning rate that is large relative to the a certain notion of the scale of the problem. In particular, they show that the use of large learning rates leads to balancing between the two factors in the factorization.

The discussion between the authors and the reviewers was fruitful in dispelling some of the reviewers' doubts and at the same time improving the paper.

The paper seems to make some contribution on a relevant problem for the ICLR community. However, even in the restricted settings they consider, the problem does not appear to be completely solved. That said, I agree with the majority of the reviewers that the step forward seems enough to warrant the acceptance.

I would still encourage the authors to take into account the reviewers' comments in preparing the camera-ready version. In particular, in the internal discussion it was suggested that the presentation of the paper could be improved by clearly stating the limitations of the current approach (e.g., the assumption of convergence in Theorem 5.1, a better discussion on large vs small learning rates w.r.t. the balancing effect).